# An 83,000 year old ice core from Roosevelt Island, Ross Sea, Antarctica

James E. Lee[1], Edward J. Brook[1], Nancy A. N. Bertler[2,3], Christo Buizert[1], Troy Baisden[3,4], Thomas Blunier[5], V. Gabriela Ciobanu[5], Howard Conway[6], Dorthe Dahl-Jensen[5], Tyler J. Fudge[6], Richard Hindmarsh[7], Elizabeth D. Keller[3], Frédéric Parrenin[8], Jeffrey P. Severinghaus[9], Paul Vallelonga[5], Edwin D. Waddington[6], and Mai Winstrup[5]

[1]College of Earth, Ocean, and Atmospheric Sciences, Oregon State University, Corvallis, OR 97331, USA
[2]Antarctic Research Centre, Victoria University of Wellington, Wellington, New Zealand
[3]GNS Science, Gracefield, Lower Hutt, 5010, New Zealand
[4]Now at Faculty of Science and Engineering, University of Waikato, Hamilton, New Zealand
[5]Centre for Ice and Climate, Niels Bohr Institute, University of Copenhagen, Copenhagen, Denmark
[6]Department of Earth and Space Sciences, University of Washington, Seattle, WA 98195, USA
[7]British Antarctic Survey, Cambridge CB3 0ET, United Kingdom
[8]University Grenoble Alpes, Centre National de la Recherche Scientifique, Institut de Recherche pour le Développement, Institut des Géosciences de l'Environnement, 38000 Grenoble, France
[9]Scripps Institution of Oceanography, University of California, San Diego, La Jolla, CA 92093, USA

**Correspondence:** James E. Lee (JLee@COAS.OregonState.edu)

**Abstract.** In 2013 an ice core was recovered from Roosevelt Island, an ice dome between two submarine troughs carved by paleo-ice-streams in the Ross Sea, Antarctica. The ice core is part of the Roosevelt Island Climate Evolution (RICE) project and provides new information about the past configuration of the West Antarctic Ice Sheet and its retreat during the last deglaciation. In this work we present the RICE17 chronology, which establishes the depth-age relationship for the top 754 m of the 763 m core. RICE17 is a composite chronology combining annual layer interpretations for 0-343 m (Winstrup et al., 2017) with new estimates for gas and ice ages based on synchronization of $CH_4$ and $\delta^{18}O_{atm}$ records to corresponding records from the WAIS Divide ice core and by modeling of the gas age-ice age difference.

Novel aspects of this work include: 1) an automated algorithm for multi-proxy stratigraphic synchronization of high-resolution gas records, 2) synchronization using centennial-scale variations in methane for pre-anthropogenic time periods (60-720 m, 1971 CE to 30 ka), a strategy applicable for future ice cores, and 3) the observation of a continuous climate record back to ~65 ka providing evidence that the Roosevelt Island Ice Dome was a constant feature throughout the last glacial period.

## 1 Introduction

The Roosevelt Island Climate Evolution (RICE) project seeks to combine geophysical measurements with climate information from a well-dated ice core to improve constraints of the glacial history of the eastern Ross Sea (Conway et al., 1999). With this motivation, the RICE project drilled and recovered a 763 m long ice core from Roosevelt Island. Here, we first present new

data sets from the RICE ice core and then describe the development of the RICE17 age-scale, a composite age-scale which combines annual layer counts (Winstrup et al., 2017) with new age constraints and spans the top 753.75 m.

The most precise chronologies for ice cores have been constructed by combining absolute age markers, for example volcanic ash layers, with annual layers counts from visual stratigraphy and from variations in chemical concentrations, stable isotope ratios of the ice, and electrical properties. Examples include the Greenlandic ice cores composite chronology (GICC05) which extends to 60 ka (Svensson et al., 2008) and the Antarctic WAIS Divide WD2014 chronology through the last 31 ka (Sigl et al., 2016). For the RICE ice core this strategy was only applicable above 165.02 m, the depth of the oldest absolute age marker, at 1251 C.E. (Winstrup et al., 2017). Annual layer interpretations were extended to 343.7 m (2649 yr BP, all ages are reported as years before present (BP), where present is defined as 1950 C.E., Winstrup et al. 2017). Below this depth, annual layers became too thin to reliably interpret the seasonal signals.

For the 343.7-754 m section of the RICE ice core, the best age constraints are from measurements of methane and the isotopic composition of molecular oxygen ($\delta^{18}O_{atm}$). Variations in these parameters are globally synchronous (Bender et al., 1994; Blunier et al., 1998) meaning that, to a first-order, different ice cores will record the same atmospheric history. We establish the gas-phase depth-age relationship by matching the new RICE records to corresponding records from other ice cores with established chronologies. From the gas age-scale, an ice-phase age-scale was obtained by modelling the ice age-gas age offset ($\Delta$age). $\Delta$age is constrained by measurements of the isotopic composition of molecular nitrogen ($\delta^{15}N$-$N_2$) and estimates of temperature based on $\delta D$ of ice. For overlapping depths the gas-based ice age estimates are in excellent agreement with the annual layer counted age estimates. A final age-scale for the entire core, named RICE17, is presented and is a composite of the annual layer-based age-scale for 0-343.7 m depth and the gas-based age-scale for deeper sections of the core.

The approach used for RICE17 is not unique but includes several refinements that improve the dating accuracy. Primarily, the RICE17 chronology benefited from the availability of high-resolution gas records from both the RICE ice core and the WAIS Divide ice core. Centennial-scale variability of methane was well-captured for the last 30 ka in both data sets. Additionally, an automated matching routine, adapted from Huybers and Wunsch (2004) and novel in the application to ice cores, simultaneously matched methane and $\delta^{18}O_{atm}$ records for ice up to 30 ka (0-720 m). The multi-proxy approach to synchronization leads to strengthened age estimates. Using an automated routine resulted in a more objective match of the RICE gas records to the reference records and an Monte Carlo based estimate of age uncertainty.

RICE17 is a continuous chronology with age monotonically increasing with depth until at least 64.6 ka. At 746.00 m, folding of the ice is observed and the age-scale is segmented and no longer continuous. The oldest ice near the bottom of the core has a minimum age of 83 ka. We use these observations, in support with measurements of total air content (TAC), to interpret that the Roosevelt Island ice dome was stable throughout the the last glacial maximum.

## 2 The Roosevelt Island ice core and glaciological history

Undertanding the stability of the West Antarctic Ice Sheet (WAIS) is important for predicting future sea level rise (Church et al., 2013; Jevrejeva et al., 2014; Golledge et al., 2014; Pollard et al., 2015; DeConto and Pollard, 2016). Much of the ice sheet is grounded below sea level with the bed deepening towards the center of the ice sheet. This configuration is thought to be

unstable and prone to rapid disintegration due to physical forces related to bouyancy (Hughes, 1973; Weertman, 1974; Schoof, 2007; Feldmann and Levermann, 2015), vulnerability to undercutting by "warm" subsurface currents (Robin and Adie, 1964; Shepherd et al., 2004), and ice cliff instability (DeConto and Pollard, 2016). Vulnerability of WAIS to future warming can be assessed by investigating how it has responded to different climate regimes in the past. Unfortunately, geologic evidence of the past size and extent of Antarctic ice sheets is spatially sparse, tends to have large chronological uncertainty, and is sometimes

contradictory (Whitehouse et al., 2012; Anderson et al., 2014; Bentley et al., 2014; Clark and Tarasov, 2014; Halberstadt et al., 2016; McKay et al., 2016).

The Ross Embayment is the largest drainage of WAIS, both in terms of area and mass loss (Halberstadt et al., 2016). Historically, two scenarios have been proposed for the configuration of WAIS in the Ross Embayment during the Last Glacial Maximum (LGM) (Stuiver et al., 1981). In the "maximum scenario," a thick and grounded ice sheet in the Ross Sea extended

to the continental shelf break (Denton et al., 1989). Details of this scenario are supported by geomorphic features including grounding-zone wedges, which form at the terminus of marine based glaciers (Shipp et al., 1999), and over-compressed diamictons, which are the result of thick overlying ice (Anderson et al., 1984, 1992). Evidence of high stands in the Transantarctic Mountains and the islands of the western Ross Sea (Denton and Marchant, 2000) as well as cosmogenic exposure dates on nunataqs in Marie Byrd Land (Stone et al., 2003) also support this idea. In an alternate scenario, Denton et al. (1989) proposed

that grounded ice in the Ross Sea was kept thin by fast-flowing ice streams. In this scenario, the retreat of WAIS during the last deglaciation may not have contributed significantly to sea level change. Studies of ice cores from Byrd Station (Steig et al., 2001) and Siple Dome (Waddington et al., 2005), glacial modeling (Parizek and Alley, 2004), and cosmogenic exposure dates from the Ohio Range (Ackert et al., 1999, 2007) all support this "minimal scenario."

Roosevelt Island is an ice dome located in the eastern Ross Sea in West Antarctica (79.36° S, 161.71° W; elev. 550 m above

sea level, Fig. 1). It is grounded on a submarine plateau (∼200 mbsl) dividing the Whales Deep and Little America Basins (Fig. 1). During the LGM these troughs were presumably occupied by the extension of the modern Bindschadler and MacAyeal ice streams (Ice Streams D and E, respectively) (Shipp et al., 1999), and Roosevelt Island would have been located along the main ice flow of WAIS.

The 763 m long RICE ice core was drilled to bedrock near the summit of Roosevelt Island. In addition to the main deep core,

a shallow core was drilled to 20 m and several snow pits were sampled to understand recent climate (Bertler et al., 2018). Local mean annual air temperature is -23.5° C and annual snow accumulation is estimated to be ∼0.22 m ice equivalent, based on annual layers in snow pits (Bertler et al., 2018). A cooler estimate of modern temperature, -27.4 ± 2.4° C, based on ERA interim data from 1979-2012 was presented in Bertler et al. (2018), but this estimate is cooler than borehole thermometry measurements and previously published estimates for Roosevelt Island (Herron and Langway, 1980; Conway et al., 1999; Martín et al., 2006)

and does not provide a good fit to the density profile in the firn model. Other estimates of recent accumulation at the RICE drill site range from 0.18 m ice per year to 0.27 m ice per year, depending on method and time period (Winstrup et al., 2017; Bertler et al., 2018; Herron and Langway, 1980; Conway et al., 1999; Kingslake et al., 2014).

## 3   New data sets from the RICE ice core

### 3.1   Methane concentrations

The RICE discrete methane record was measured at Oregon State University (OSU) following methods described by Mitchell et al. (2011, 2013) with updates described in Appendix A. A total of 702 samples were measured at 583 distinct depths between 60-753 m (Fig. 2a, e). Samples from 406 depths were measured between 60 and 670 m, dating from ∼1970 C.E. to 11.87 ka, with a mean sample spacing of 28.75 years. Between 670 and 718.13 m the record spans 11.87 to 29.9 ka; 96 samples in this interval provide age resolution of 189 years. Age resolution decreases significantly for deeper depths. The interval from 718.53 to 746.00 m corresponds to 30.1-64.6 ka with a mean resolution of 548 years. The deepest dated ice is at 752.95 m with an age of 83 ka (±2 ka).

Methane was also measured continuously with a laser spectroscopy technique (Stowasser et al., 2012; Rhodes et al., 2013) during two separate continuous flow analysis (CFA) campaigns at GNS Science (Gracefield, New Zealand) in 2013 and 2014 (Pyne et al., 2018). The CFA methane record was affected by variability of air flow to the measurement instrument and fractures that allowed drill fluid and modern air into the melt head. Exclusion of these artifacts caused significant gaps in the record, particularly at depths below 676 m (12.6 ka). We consider the CFA record to be a supplement to the more robust but lower resolution discrete data. Between 29.9-59.1 ka, the CFA record is critical for establishing age control. Below ∼746 m (64.6 ka), the CFA data are difficult to interpret because of gaps in the record, uncertainty in measurement depth, and uncertainty in methane calibration (Appendix B).

Several anomalously high discrete methane measurements appear between 44.6-50.9 ka (729.05-736.05 m) and below 64.6 ka (746 m) (Fig. 3a). In the former interval methane is enriched by ∼30 ppb compared with the WAIS Divide CFA data (Rhodes et al., 2015). This is likely due to healed fractures that include modern air (Aydin et al., 2010). All RICE samples deeper than 500 m were visually inspected for fractures as they were prepared for measurement and for drill fluid during air extraction. In fractured ice it was common to see drill fluid in the flask as the samples were melted. However, neither observation was a strong indicator that a sample would have an elevated methane concentration. None of these high concentration results were rejected.

### 3.2   Total air content

Total air content (TAC) is defined as the amount trapped air in a sample, reported here in units of $cm^3$ air at STP/g ice. TAC (Fig. 2d, h) was measured at OSU as part of the methane concentration measurement following methodology of Mitchell et al. (2015) and updates from Edwards et al. (in prep). TAC is influenced by accumulation and temperature, total summer insolation

(Raynaud et al., 2007), thermal gradients in the firn from multi-annual climate trends, and surface air pressure (Martinerie et al., 1992; Raynaud and Whillans, 1982).

Preparation of ice samples causes loss of air from bubbles which intersect the surface of the sample, an effect known as the cut-bubble effect (Martinerie et al., 1990). The amount of air lost due to the cut-bubble effect depends on the geometry of the sample and the bubble size. For bubble size, we assume that the bubble diameter at the bubble-close-off depth is the same as at the bubble-close-off depth at WAIS Divide and that the diameter shrinks exponentially to zero at the bubble-clathrate-transition depth (763 m). Ice samples were cut to uniform shapes to limit variability of TAC related to the cut-bubble effect so that TAC can be directly compared between samples. However, inclusion of some fractures was often unavoidable and their contribution to gas loss is potentially large. TAC data were rejected when gas loss was believed to greatly impact the results, such as in samples with visible fractures or samples which consisted of multiple pieces. Of the 706 samples measured at OSU for TAC, 165 results were rejected based upon visual inspection of the sample. Many of these came from the 670-752.95 m (11.7-83 ka) interval where only 58 of 177 TAC measurements are considered reliable. Reproducibility of replicate TAC measurements is 0.7% after application of a correction for the cut-bubble effect (reproducibility of 0.6% without correction). Estimate of uncertainty of TAC measurements is 28% with the largest contributions to the uncertainty from the shape of the sample and the assumed bubble diameter used for the cut-bubble correction.

The TAC record from the RICE ice core without the cut-bubble correction appears remarkably consistent (age weighted mean=0.1182 cm$^3$ air-STP/g ice, age weighted standard deviation=0.0023 cm$^3$ air-STP/g ice, n=410). After application of the cut-bubble correction, the TAC record shows a trend to higher values at more recent ages (Fig. 3d). This trend reflects the decrease of bubble size with depth and therefore a larger cut-bubble correction for shallower samples. Although this trend has a physical explanation, it is not statistically significant due to the uncertainty of the cut-bubble correction. We advise caution when interpreting the absolute values of TAC from the RICE ice core due to the potential artifacts caused by gas loss through fractures and the large uncertainty in the cut-bubble correction.

### 3.3 $\delta^{18}O_{atm}$ and $\delta^{15}$N-N$_2$

$\delta^{18}O_{atm}$ and $\delta^{15}$N-N$_2$ were measured on samples adjacent to the discrete methane samples (Fig. 2b, f and c, g, respectively). Analysis was conducted at Scripps Institution of Oceanography following Petrenko et al. (2006) and Severinghaus et al. (2009). Pooled standard deviation of replicate measurements is 0.006‰ for $\delta^{18}O_{atm}$ and 0.0027‰ for $\delta^{15}$N-N$_2$ (both scales are relative to modern atmospheric composition).

Variations of $\delta^{18}O_{atm}$ are primarily caused by changes in location and intensity of low-latitude rainfall that affect the $\delta^{18}O$ of leaf water used in photosynthesis (Severinghaus et al., 2009; Landais et al., 2007, 2010; Seltzer et al., 2017) and changes in seawater $\delta^{18}O$ caused by ice sheets on glacial cycles (Horibe et al., 1985; Bender et al., 1985, 1994; Sowers et al., 1993). Importantly, these variations are well known in independently dated ice cores and the atmosphere is well mixed on the relevant timescales, so $\delta^{18}O_{atm}$ variability may be used as a chronostratigraphic marker (Bender et al., 1994). These variations are well sampled in the RICE record until ∼64.6 ka (746.00 m), beyond which the chronology is no longer continuous.

Atmospheric $N_2$ is isotopically stable over very long time-scales (Hattori, 1983; Sowers et al., 1989; Schwander, 1989). Variability of $\delta^{15}N$-$N_2$ in the ice core records primarily reflects changes in gravitational fractionation and thermal fractionation within the firn due to changes in surface temperature and accumulation rate (Schwander, 1989; Sowers et al., 1989). In Antarctica, temperature changes gradually and the effect of thermal separation is minimal.

## 4    Strategy for developing the RICE17 chronology

In this section, we describe how the RICE17 age-scale is developed. We start with age constraints in the gas-phase based on well recorded variations in $\delta^{18}O_{atm}$ and methane. We then discuss how the gas age-scale is used to estimate the gas-phase ice-phase age offset and thus the ice age-scale. Finally, we compare the ice age-scale to the age-scale derived from annual layer interpretations.

All age control points (ACPs), additional control points from the synchronization routine (floating control points, discussed below), and gas and ice ages interpolated to more closely spaced depths are provided in the supplementary material.

### 4.1    An automated matching algorithm for synchronizing ice core records: 0-30 ka

An automated matching algorithm was adapted from Huybers and Wunsch (2004) to synchronize between gas records. Prior ACPs, shown as open triangles in Fig. 3 and Fig. 5, all correspond to well defined increases or decreases of either methane or $\delta^{18}O_{atm}$. We assume that the uncertainty (2 standard deviations) for an ACP corresponds to the time elapsed between 25% and 75% of the change in either methane or $\delta^{18}O_{atm}$ (Table 1). The goal of the routine is to iteratively adjust the interpolation between ACPs to improve a goodness-of-fit parameter by following steps 1-9 described below.

"Goodness-of-fit" is calculated as the $\chi^2$ value comparing the normalized methane record from the RICE ice core to expectant values interpolated from the WAIS Divide ice core plus the analogous $\chi^2$ value comparing the normalized $\delta^{18}O_{atm}$ records. The algorithm can accept additional parameters for synchronization, such as $CO_2$ or $N_2O$, if available. In this analysis we normalized both the methane and $\delta^{18}O_{atm}$ records to have the same mean (5) and variance (1) in order to equally weight their $\chi^2$ values and their importance to the synchronization.

A realization starts by randomly perturbing the age of ACPs within their prescribed uncertainty (Table 1) to define an initial depth-age scale. The perturbed ACPs remain fixed throughout the realization. ACPs are given the subscript $k$, where $k$=1 is the youngest/shallowest ACP. The records are then broken into $N$ subsections which are distributed between ACPs. Subsections are designated with a subscript $i$, where $i$=1,2,3,...N. Floating control points (FCPs) are defined as the bounding depths/ages of these subsections, which will include the initial ACPs. Prior to any perturbations, the durations of the subsections ($\Delta t$) are approximately the same. We have chosen $\Delta t$ to roughly match the recurrence interval of variations of methane in the reference record and so that each subsection contains about five methane and five $\delta^{18}O_{atm}$ samples. The following steps are repeated to optimize goodness-of-fit:

1. Random scaling factors ($p_i$), which perturb the durations of the subsections, are drawn for each subsection from a normal distribution of $\mu$=1 and $\sigma_1$=0.25.

$$\Delta t'_i = \Delta t_i \cdot p_i$$

2. Because random perturbations will change the length of time between the initial ACPs, we apply a second scaling so the perturbed chronology remains consistent with the prior ACPs. In this case, $ACP_k$ and $ACP_{k+1}$ are the nearest gas ACPs bounding subsection $i$, respectively. $ACP'$ is the perturbed age of the gas age constraint after step 1, and $\Delta t_i^*$ is the duration of the subsection after the second scaling:

$$\Delta t_i^* = \Delta t'_i \cdot \left( \frac{ACP_{k+1} - ACP_k}{ACP'_{k+1} - ACP'_k} \right)$$

3. Mean "accumulation rate" of each subsection ($\overline{A}_i$) is calculated following Nye (1963), which assumes that the thickness of annual layers ($\overline{\lambda}_i$) is the product of their original thickness and their relative depth ($\frac{\overline{z}_i}{H}$):

$$\overline{A}_i = \overline{\lambda}_i \cdot \left(1 - \frac{\overline{z}_i}{H}\right)^{-1}$$

where $\overline{z}_i$ is the mid-depth of the subsection $i$, H is the thickness of the ice sheet, and both $\overline{z}_i$ and H are in ice equivalent units.

Because the assumptions about thinning from Nye (1963) is too simple to describe the flow conditions at Roosevelt Island, we do not consider this result to be representative of the true accumulation history. The assumption is necessary for the purposes of interpolating age versus depth to ensure that annual layer thickness decreases smoothly between FCPs. An accumulation history that we believe is more accurate is calculated below with an alternative method, using a dynamic firn model (Section 4.4).

4. A perturbed chronology is only accepted if:

    (a) Intervals between FCPs ($\Delta t_i^*$) are within a factor of 10 of the initial durations,

    $$\frac{1}{10} \cdot \frac{(t_{end} - t_1)}{N} < \Delta t_i^* < 10 \cdot \frac{(t_{end} - t_1)}{N}$$

    (b) Mean "accumulation rates" are realistic ($\overline{A}_i > 0$ cm ice eq per yr, $\overline{A}_{max} < 75$ cm ice eq per yr), and

    (c) Mean "accumulation rates" of adjacent subsections are within a factor of 2 of each other.

    These conditions provide loose restrictions for continuity in the depth-age relationship although they may not be physically realistic for the site.

5. Ages for all RICE sample depths are interpolated from the perturbed FCPs (Step 3), assuming piece-wise constant accumulation between FCPs and a linearly decreasing thinning function (Nye, 1963).

6. Goodness-of-fit is calculated on the perturbed age-scale (goodness-of-fit = $\chi_{CH4} + \chi_{\delta 18Oatm}$).

7. When a perturbation improves the goodness-of-fit, that iteration becomes the base for subsequent perturbations.

8. The cycle is repeated until 20 sequential perturbations fail to improve the goodness-of-fit in step 6.

9. Starting with the FCPs from step 8, the above steps (1-8) are repeated 13 times, reducing the size of perturbations in step 1 from $\sigma_1 = \frac{1}{4}$ to $\sigma_{13} = \frac{1}{128}$.

A Monte Carlo analysis repeats these steps (Steps 1-9) 1000 times, initiated from a different prior depth-age relationship by randomly perturbing ACPs within their age uncertainty (Table 1). Parameters used for the synchronization are given in Table 2 for the 60.05-670.13 m and 670.13-719.30 m intervals. Code for the synchronization routine is provided as supplementary material.

We choose the best realization for gas age constraints for the RICE17 gas age-scale. The best age estimate (realization) is not necessarily the most frequent age estimate. Fig. 5d shows an example from sample depth 621.28 m where there is a large difference between these two age estimates. Large differences can occur because the prior age estimate (i.e. the age estimate based only on prior ACPs) differs by a large amount from the "true" age of that sample and because the goodness-of-fit parameter considers the fit over the whole record. For depth 621.28 m most realizations resulted in an age estimate of 9200 yr BP, similar to its prior age estimate of 9,240 yr BP, but the best realization estimated the age to be 9012 yr BP. There are two possible reasons for this type of result: 1) this realization is the best because it managed to push the age of this sample by >200 years towards younger ages while not significantly changing the ages of nearby sections which already fit well, or 2) no significant improvement in the goodness-of-fit was found by adjusting the age of this depth, and the goodness-of-fit was dictated by other sections of the record.

Uncertainty in gas age constraints is calculated as the root-mean-square error of FCP ages from the 1000 Monte Carlo realizations (Fig. 4d), although we acknowledge that this can overly simplify the empirical distribution of possible ages for a given depth (Fig. 5). Assessing uncertainty in this way integrates two types of error. The first is the ability to exactly match timing of two features. This uncertainty is determined by how well features are resolved in the records, measurement error, and the limited degrees of freedom in synchronization (i.e. the synchronization routine assumes constant accumulation during each subsections in a piece-wise manner, although the true accumulation history is more likely to smoothly vary). The second type of error is analogous to deciding which feature in the reference record is the correct match. For example, the methane peak centered at 459.05 m (4675 yr BP) is, in some realizations, matched to a peak at ∼4550 yr BP in the WAIS Divide record, providing two distinct age possibilities (Fig. 5c).

### 4.2 Extending the chronology with visually matched gas age control points: 30-83 ka

The more limited resolution of the RICE methane record below 719.3 m (30.66 ka) prevented use of the automated synchronization routine. Gas age control points were visually chosen between 719.30 m (30.66 ka) and 746.00 m (65.6 ka) and between 746.00 and 752.95 m (∼83 ka) (Table 1). Between 719.30 (30.66 ka) and 746.00 m (65.6 ka), ACPs were visually chosen by comparing the RICE discrete and CFA methane records and the $\delta^{18}O_{atm}$ record to the WAIS Divide methane (Rhodes et al., 2015) and $\delta^{18}O_{atm}$ records (Buizert et al., 2015b).

The WAIS Divide records and the WD2014 chronology end at 67.8 ka. For the deeper (older) section of the RICE ice core, between 746.00-752.95 m, we build target records from records of methane and $\delta^{18}O_{atm}$ from the NGRIP ice core

(Baumgartner et al., 2014; Landais et al., 2007, 2010). Using the NGRIP ice core is internally consistent with the WD2014 chronology, which is tied to a modified GICC05modelext age-scale between 30-67.8 ka (Buizert et al., 2015b). Because NGRIP is in the northern hemisphere the methane concentration is higher than that in the RICE record. To account for this interpolar difference we scale the NGRIP target methane record (Baumgartner et al., 2014) to Antarctic values using an empirical least

squares fit between WAIS and NGRIP records between 55-67.8 ka.

Buizert et al. (2015b) found that the annual layer counted portion of the GICC05modelext chronology (0-60 ka) (Andersen et al., 2006; Svensson et al., 2008) is systematically younger than ages of corresponding features found in the U/Th absolute dated Hulu speleothem record (Reimer et al., 2013; Southon et al., 2012). A fit to Hulu ages was optimized by scaling the GICC05modelext ages linearly by 1.0063. This suggests that on average 6.3 out of every 1000 annual layers were not counted.

For our NGRIP-based target records we adjust the NGRIP age-scale by adopting the scaling of Buizert et al. (2015b). Older ages in the GICC05modelext chronology are derived from the ss09sea Dansgaard-Johnsen model (Johnsen et al., 2001; NGRIP Community Members, 2004). We have added a constant 378 years (0.0063*60,000) for depths older than 60 ka in the target GICC05modelext ages to make this section continuous with the adjusted annual layer counted section.

Synchronization requires well-sampled identifiable features in both records. We estimate age uncertainty for visually matched

ACP's (ACP's older than 29.9 ka) as the larger of the sample spacing of the two records being synchronized, following methods of Brook et al. (2005). Gas age uncertainty is plotted in Fig. 4d. The largest uncertainty is ∼1500-1700 years and occurs between 41.7-47.6 ka, where the age-scale is very compressed and the RICE records are poorly sampled.

### 4.3    Age control points in the disturbed ice: 746-753 m

Continuity of the RICE ice core appears to end at 746.00 m below surface, where a discontinuity of 20‰ is observed in the $\delta$D

record (746.00-746.10 m) (Fig. 6b) and a 0.35‰ change is observed between sequential $\delta^{18}O_{atm}$ samples (745.81 and 746.20 m) (Fig. 6c). The continuous chronology described in the previous section dates this ice to the end of DO-18 (∼64.6 ka). This age is supported by very negative $\delta$D values and a trend to more enriched values at shallower depths which is consistent with warming trends observed in other Antarctic ice cores at this time period (Buizert et al., 2015b; EPICA Community Members, 2006; Petit et al., 1999; Parrenin et al., 2013) (Fig. 6f).

Figure 6 presents our best effort to date the ice in the disturbed section, between 746.00 and 752.95 m. The left panel of Fig. 6 shows the data as a function of depth, color-coded by age. The right panel shows the reconstructed time history of these variables, color-coded by depth. Below, we discuss how the ages are assigned to different depth sections (divided by vertical red lines in the left side of Fig. 6), starting with the shallowest portion. Dating folded ice remains challenging, and the solution we found may not be unique.

Immediately below the discontinuity, between 746.10-747.85 m, methane and $\delta^{18}O_{atm}$ values best match our target records during DO-20 between 74.5-77.3 ka (Fig. 6), although the RICE methane record appears ∼30 ppb too high. Dating of this section to DO-20 implies that either 9.4 ka of climate history is missing from the RICE ice core because climate was not recorded or that 9.4 ka is compressed into ∼10 cm of ice. Extremely thin layers can be explained by ice flow or by an absence

of accumulation, but the latter scenario would cause a collapse of the firn and $\delta^{15}$N-$N_2$ values approaching 0‰ which are not observed.

A cluster of samples between 747.85-750.46 m are $\sim$0.16‰ more enriched in $\delta^{18}O_{atm}$ than was observed in the shallower section dated to DO-20 (Fig. 6c). In the NGRIP, EDML, and Siple Dome ice cores, a long-term trend in $\delta^{18}O_{atm}$ towards more enriched values is observed from 80 to 65 ka (Capron et al., 2010; Severinghaus et al., 2009; Landais et al., 2007) (Fig. 6g). The enriched values between 748.34-750.46 m most likely indicate that this ice is younger than the adjacent shallower depths and the stratigraphic order is reversed. This depth range is best matched to DO-19 between 71.8-74.3 ka (Fig. 6). Below 750.46 m, methane and $\delta^{18}O_{atm}$ return to values that best match DO-20.

We note that two significant measurement gaps exist in the $\delta$D record, at 746.89-747.07 m and at 750.00-750.25 m (gray bars in Fig. 6b). A 12‰ shift in $\delta$D accompanies both of these gaps. The second measurement gap (750.46-750.56 m) is at nearly the same depth which separates the samples dating to DO-19 from the grouping immediately below dated to DO-20. The discontinuity of the $\delta$D record at these gaps may signify another hiatus in the climate record or highly contorted layers that are typically found around folds (Cunningham and Waddington, 1990; Alley et al., 1997; Thorsteinsson and Waddington, 2002; Waddington et al., 2001).

Below 750.46 meters trends in methane and $\delta^{18}O_{atm}$ indicate that the age of ice increases with depth until at least 753 m ($\sim$83 ka). Age of this depth is constrained by a measured $\delta^{18}O_{atm}$ value of -0.175‰. Such a depleted value is rare and only occurs during periods of high sea-level and small ice sheets, the most recent time period prior to the Holocene being MIS-5a (80-85 ka) (Severinghaus et al., 2009; Capron et al., 2010; Petit et al., 1999; Landais et al., 2007; Kawamura et al. , 2007). Negative values were observed at two other depths below 753 m (Fig. 2f), but the stratigraphic order of these depths is difficult to assess.

## 4.4    Firn modelling to determine $\Delta$age

Air transport in the firn causes an age difference between ice and air trappend in the ice, commonly referred to as $\Delta$age. Firn densification models are typically used to simulate past $\Delta$age (Schwander et al., 1997; Goujon et al., 2003). Input parameters (for example, temperature, accumulation rate, and surface density) are normally the larges sources of uncertainty. Using $\delta^{15}$N-$N_2$ as a proxy for past firn column thickness and assuming $\delta$D records past site temperature faithfully, firn densification models can be run in an inverse mode to estimate both past $\Delta$age and accumulation rates (Buizert et al., 2014, 2015b), and is the approach we employ here.

We use a dynamic version of the Herron-Langway model (Herron and Langway, 1980), which was also used for construction of the WD2014 chronology (Buizert et al., 2015b). The model simulates firn compaction rates and vertical heat diffusion and advection. The model domain is the full 763 m ice column at 0.5 m resolution; a time step of 1 year is used. The model simulates both gravitational enrichment of $\delta^{15}$N-$N_2$ and fractionation in the presence of thermal gradients. The model is forced with a temperature history derived from the $\delta$D record assuming a constant isotope sensitivity of 6‰·K$^{-1}$ (Brook et al., 2005). In conjunction with an assumed geothermal heat flux of 78 mW·m$^{-2}$, this provides a good fit to the measured borehole temperature profile (Clemens-Sewall et al., Unpublished). This sensitivity is similar to that observed for West Antarctica (Stenni

et al., 2017; Cuffey et al., 2016; Masson-Delmotte et al., 2008) but somewhat higher than that obtained by comparing the RICE isotope record to ERA interim data (Bertler et al., 2018). However, lower sensitivities decreased the fit of the modeled borehole temperature profile. The dependency of $\Delta$age on the assumed isotope sensitivity was explored in a model sensitivity test (Appendix C), and is incorporated into the $\Delta$age uncertainty estimates. The model further assumes a constant ice thickness,

a constant 2 m convective zone similar to other high accumulation dome sites with low mean wind speeds (Kawamura et al., 2006; Landais et al., 2006), and surface firn density of $400 \, kg \cdot m^{-3}$ to match the modern surface firn density.

In a first iteration, we assume a prior ice age-scale for the temperature history by adding a constant 150 years to the gas chronology (Section 4.1 and 4.2). A new $\Delta$age solution is then calculated using the dynamic firn densification model. The $\Delta$age solution (and thereby the ice age-scale) is refined iteratively until it no longer changes appreciably (consistent within $\sim$1

year).

The climate at Roosevelt Island, with high accumulation and relatively warm temperatures, results in small $\Delta$age values and consequently relatively low absolute uncertainty in $\Delta$age compared to most Antarctic sites. Modern $\Delta$age (estimated at 60 m depth, the shallowest measurement of $\delta^{15}$N-N$_2$) is 146 years with a reconstructed lock-in depth (LID) of 48 m, consistent with the modern density profile (Bertler et al., 2018; Winstrup et al., 2017). Holocene $\Delta$age is small, ranging between 140 to 182

years. During the last glacial period simulated $\Delta$age values fluctuate between $\sim$150-350 years.

Uncertainties in past $\Delta$age include the uncertainty in model inputs as well as the model itself. The uncertainty of the Herron-Langway model is conservatively estimated to be 20%, based on differences between firn models (Lundin et al., 2017). We assessed the uncertainty due to model inputs using a steady-state Herron-Langway model that approximates the dynamic version but requires less computational time (Appendix C). In a sensitivity test, we randomly perturb the parameterizations

of temperature and LID and assumptions of convective-zone thickness, surface firn density, and geothermal heat flux and recalculate the $\Delta$age solution (model parameters and their base values and ranges used in the sensitivity test are provided in Table A1). A total of 6000 iterations were run. The sensitivity tests include a wide range of temperature histories to account for the possibility that some variations in $\delta$D were caused by non-thermal effects such as variability in precipitation seasonality, moisture sources and pathways, and post-depositional vapor exchange. Model sensitivity is reported as the root-mean-square-

error of $\Delta$age calculations for each depth. Combined $\Delta$age uncertainty, provided in the included age-scale, is the root-sum-squares of the model uncertainty and the model sensitivity.

## 4.5    Comparison of gas-derived and layer counted chronologies: 0-2,649 yrs BP

The gas-derived ice age-scale provides a chronology from 60 to 753 m depth (Fig. 2) that is independent of but overlaps the annual layer counted section (0-343.7 m, Winstrup et al. 2017). Figure 7 shows both chronologies and differences between

them (positive values indicate that the annual layer counts are older than the gas-derived ages). The gas-derived ice age-scale agrees well with the annual layer counted age-scale, within 33 years, with similar trends in the implied annual layer thickness (Fig. 7c). Differences between chronologies can result from error in the synchronization of gas records, calculation of $\Delta$age for either the RICE or WAIS Divide ice cores, interpolation between ACPs, or annual layer counts in either core (because the WAIS Divide age-scale is used as a reference for the gas age-scale).

The average age difference at depths of FCPs is -1 years (n=18, implied age from gas-derived ice age-scale being older than the layer counted chronology). Root-mean-square of the age difference is 17.3 years. Discrete points in the gas-derived ice age-scale can also be compared to the age of 67 volcanic peaks identified in the RICE ice core and correlated to peaks identified in the WAIS Divide ice core (open red circles in Fig. 7, Winstrup et al. 2017). Compared to these volcanic peaks, the root-mean-square ice age difference of the gas-derived ice age-scale is 13.6 years from their WD2014 ages. Good agreement between the two approaches gives confidence in the methodology used for the deeper section of the RICE17 chronology. The two largest differences occur at 89.72 m (243 yr BP) and at 169.11 m (757 yr BP) and are +30 years and -33 years, respectively (Fig. 7b). These offsets are similar in magnitude to the individual uncertainties in calculating $\Delta$age or in synchronizing the gas records. The small age differences between the two ice chronologies also indicates that our approach to calculating uncertainty is likely conservative.

The RICE17 time scale transitions between the annual layer counted and gas-derived age-scales at 343.7 m, the deepest/oldest FCP for which annual layers were identifiable (Section 4.1). The age difference at this depth is 3 years, with the gas-derived ice age implying an older age than the annual layer counted chronology. This age difference is much less than the respective age uncertainties of 45 years and 111 years (2-$\sigma$) for the annual layer counts and gas-derived ice age-scale, respectively. Because of the good agreement between the two ice age-scales, no correction is made for the 3 year offset and the annual layer counted age is used.

## 5 Results and key observations from the RICE17 chronology and gas data sets

### 5.1 Implied accumulation history of Roosevelt Island

From 65 to 32 ka, the low-variability of $\delta^{15}$N-$N_2$ ranging from 0.20 to 0.24‰ and the cooling trend observed for Antartica for this time period suggest a decreasing trend in accumulation (Fig. 3).

While $\delta^{15}$N-$N_2$ values appear steady from 65 to 32 ka and during the Holocene (Fig. 3c), they exhibit large variability between 32 and 10 ka (Fig. 8a). $\delta^{15}$N-$N_2$ generally trends to heavier values beginning at 32 ka until 14.71 ka, indicating a growing firn column. An inflection point is observed at 25.3 ka which is interpreted as an acceleration of firn thickening concluding in a peak of 0.293‰ at 21.8 ka. After a short depletion the steep trend resumed with a second $\delta^{15}$N-$N_2$ maximum of 0.326‰ reached at 15.7 ka. Following this maximum, $\delta^{15}$N-$N_2$ abruptly decreased from 0.308‰ to 0.220‰ at 14.71 ka, which corresponds with the beginning of the Antarctic Cold Reversal (ACR). At 12.38 ka, after the ACR, $\delta^{15}$N-$N_2$ partially recovers to pre-ACR values with an abrupt increase to 0.260‰.

Between 25.3-32 ka, accumulation is estimated to be ~10 cm ice equivalent per year (Fig. 8e) and the increasing firn thickness is largely attributable to decreasing temperature. After 25.3 ka accumulation starts to increase and by the first $\delta^{15}$N-$N_2$ maximum (21.8 ka), accumulation had increased to ~17 cm ice equivalent per year. This feature is not apparent in other ice cores from the Ross Sea region, but those records tend to be difficult to interpret because of chronological uncertainties, such as is the case for Taylor Dome (Baggenstos et al., 2017), or because of unexplained jumps in $\delta^{15}$N-$N_2$, such as is the case for Siple Dome (Severinghaus et al., 2009). By the second maximum (15.7 ka) accumulation increased to ~25 cm ice equivalent

per year, similar to accumulation rates observed during the Holocene. An accumulation peak at the end of glacial terminations is consistent with evidence from trimlines in interior WAIS (Ackert et al., 2013) and is also observed in the accumulation histories from WAIS Divide and Siple Dome (WAIS Divide Project Members, 2013; Waddington et al., 2005), but to a smaller degree. The early accumulation peak at 21.8 ka is unique to the RICE ice core.

A large, rapid depletion in the $\delta^{15}$N-N$_2$ is observed at 14.71 ka with low values lasting until 12.38 ka. Analysis of the lead-lag relationship between $\delta^{15}$N-N$_2$ and methane has been used to infer near-synchronous climate changes throughout the tropics and northern hemisphere (Rosen et al., 2014). These climate events are believed to originate in the northern hemisphere and propagate to the southern hemisphere (Blunier et al., 1998; Blunier and Brook, 2001; Buizert et al., 2015a; Buizert et al. , 2018). Curiously, the abrupt decrease in $\delta^{15}$N-N$_2$ at 14.71 ka (0.088 permil, 683.70-681.80 m) is observed in samples deeper

than the increase in methane marking the onset of the Bølling-Allerød (14.66-14.52 ka, 683.13-682.52 m) meaning that this climate event unambiguously precedes the Northern Hemisphere event (Fig. 9). Where as Rosen et al. (2014) specifically considered the thermal component of $\delta^{15}$N-N$_2$, the thermal component is considered to be negligible in Antarctic cores because air temperature only changes gradually. At Roosevelt Island, this period of low $\delta^{15}$N-N$_2$ values is interpreted to represent shallow firn thickness which in the firn model is caused by a large reduction of snow accumulation (<10 cm/yr, Fig. 8e). Low

accumulation during this period is consistent with thin annual layers interpreted from the age-depth relationship (Fig. 3d); 0.3-0.6 cm/yr annual layer thickness compared with 1.6-3.2 cm/yr between 10.09-11.01 ka (642.75-661.07 m). Following the ACR, the modeled accumulation rate fully recovers to the ∼25 cm per year observed before the ACR.

We propose three potential explanations for the thin annual layers and low accumulation rate observed during the ACR, with the last explanation being our preferred. 1) A large accumulation gradient is observed across the Roosevelt Island ice dome

(Winstrup et al., 2017) which implies that a period of low accumulation at the RICE site may be the result of changes to the geometry of the Roosevelt Island ice dome. However, ice divide migration typically occurs over long timescales. 2) Accelerated ice flow may also cause thin annual layers and could potentially even affect layers within the firn. This flow could be the result of the ice streams which surrounded and buttressed Roosevelt Island ice dome suddenly being removed (Ackert et al., 1999, 2013; Halberstadt et al., 2016). The timing of the low accumulation interval is similar to when dust records from the Taylor

Dome ice core (Aarons et al., 2016), a site located in the Transantarctic Mountains near the Ross Sea (Fig. 1), show an increase in dust diameter and a change in radiogenic isotope composition which implies that the dust source changes to a local source. Aarons et al. (2016) interpret this source is newly exposed land created by the withdrawal of Ross Ice Shelf. The timing of the low-accumulation period is also similar to some estimates of the timing of retreat of the WAIS based on sediment facies succession and radiocarbon dating (minimum date of 8.6 ka, McKay et al. 2016) and ice-sheet modeling (Golledge et al., 2014;

McKay et al., 2016). Although annual layer thickness immediately above the ACR section is observed to be nearly five times as thick as the ACR section, this large change in thickness, if solely the result of changes in ice flow, would require an unrealistic change in the thickness of the dome or its ice flow which is not supported by TAC measurements or ice-flow modeling.

3) Our preferred explanation is that an interval of low accumulation between 12.38 and 14.71 ka resulted in a shallow firn structure and depleted $\delta^{15}$N-N$_2$ values. In recent times, moisture arriving at Roosevelt Island is frequently related to enhanced

cyclonic air flow in the Ross Sea and a strong Amundsen Sea Low (Tuohy et al., 2015; Emanuelsson et al., 2018). This

period of low accumulation may indicate a changed atmospheric structure in the South Pacific where southward air flow is blocked by a persistent low pressure zone north of the Ross Sea such as observed in more recent periods in the accumulation record from RICE (Bertler et al., 2018; Emanuelsson et al., 2015) and potentially related to the past ice shelf extent. Such a pronounced minimum in accumulation is not observed in the Siple Dome ice core, where annual layers are observed to thicken during the ACR (Brook et al., 2005; Waddington et al., 2005). If non-thermal effects influenced the RICE $\delta$D record, which is interpreted as temperature in our firn model, then the magnitude of accumulation change during this period may not be as large as reconstructed from the firn model. While not currently available, measurements of d$_{excess}$, dust particle size distributions, and dust geochemistry may be helpful in explaining the temperature and accumulation history of RICE.

## 5.2 Implications for climate and ice sheet history

Early reconstructions of the Ross Ice sheet during the LGM were based on glacial geological constraints from the western margin of the embayment. Denton and Hughes (2002) describe a maximum scenario in which thick ice in the Ross Embayment overrode both Siple Dome and Roosevelt Island. However, more recent observations and model experiments indicate a "fast and thin" scenario in which Siple Dome was not overrun by the interior ice sheet during the LGM (Waddington et al., 2005; Price et al., 2007) and although the Ross ice streams likely slowed down during the LGM, they remained active, maintaining a low elevation profile of the ice sheet in the Ross Sea (Parizek and Alley, 2004).

Our results further support the fast and thin scenario, and add a key new constraint on ice thickness and thinning in the Eastern Ross Sea. Specifically, our results suggest that ice deposited on Roosevelt Island originated as accumulation local to the drilling site which may not be true if WAIS was thick during the LGM and overrode Roosevelt Island. If remnants of WAIS were stranded on Roosevelt Island, this would likely result in a hiatus in the gas and ice chronology, in values of $\delta$D or TAC characteristic of more continental precipitation or much higher elevations, or discontinuities in the $\delta^{15}$N-N$_2$ record indicating a much different firn structure.

While the RICE ice core chronology does exhibit an abrupt shift in $\delta^{15}$N-N$_2$ and TAC during the ACR 3), no discontinuity in $\delta$D was observed meaning that it is unlikely that any of this ice originated as part of WAIS. At least one age discontinuity (at 64.6 ka) as well as an age reversal was observed deeper in the core, but these depths do not coincide with the timing of the retreat of WAIS in the Ross Sea (McKay et al., 2016). The largest discontinuity, at 746.00-746.10 m, is accompanied a 20‰ change in $\delta$D. Methane and $\delta^{18}$O$_{atm}$ indicate that this discontinuity represents a 9.4 ka age gap. Over the same age range, the EDML ice core records a similar change in $\delta$D (Fig. 6f) indicating that this change in $\delta$D is explained by Antarctic climate patterns alone and without invoking large changes in ice sheet configuration. A possible second discontinuity is observed at 747.00 m depth at a small gap in measurements of 2.7 cm. Ice immediately below 747.00 m is interpreted as being warmer, meaning that this is probably not derived from somewhere upstream in WAIS or from a higher elevation (Fig. 6b). Additionally, no dramatic or sudden change in $\delta^{15}$N-N$_2$ or TAC was observed in association with either of these discontinuities (Fig. 6). We conclude that it is highly unlikely that the accumulation site of the RICE ice core changed during the deglaciation and that Roosevelt Island ice dome probably remained independent of an advanced WAIS during the LGM. A similar conclusion was

reached about Siple Dome during the LGM (Waddington et al., 2005; Nereson et al., 1998; Price et al., 2007; Parizek and Alley, 2004).

Geomorphological features on the Ross Sea bed do provide evidence of grounded ice north of Roosevelt Island during the LGM (Shipp et al., 1999; Anderson et al., 1984, 1992, 2014; Halberstadt et al., 2016). If these features were formed by an extended WAIS, it would imply that ice flowed around Roosevelt Island and Siple Dome and therefore was limited in thickness. These conditions would indicate that ice streams were active throughout the last glacial period. Alternatively, these geomorphic features may be the result of ice from a different origin. Price et al. (2007) proposed that during the LGM, an ice dome may have existed on Mary Byrd Land. In this scenario, thick, grounded ice could exist north of Roosevelt Island without flowing over or around the Roosevelt Island sea rise. The RICE records can not distinguish between these scenarios.

## 6 Conclusions

We present the RICE17 gas and ice chronologies for the RICE ice core. These timescales date the gas and ice records for the last $\sim$83 ka. Between 0-30 ka an automated synchronization routine is used to identify gas age control points that best match the RICE methane and $\delta^{18}O_{atm}$ records to the respective records from the WAIS Divide ice core (WAIS Divide Project Members, 2013; Buizert et al., 2015b). This technique requires few prior constraints, accommodates simultaneous synchronization of multiple parameters, and allows assessment of age uncertainty. Unique in this approach is the use of centennial-scale variability of methane for chronostratigraphic matching for ages older than the last $\sim$2,000 years. Synchronization between ice cores for the time period between 30 and 83 ka (719-753 m) was accomplished by manually choosing tie-points. Below 753 m the ice could not be dated with the currently available data. The RICE17 ice age-scale is based on annual layer counts between -62 and 2649 yr BP (0-343.7 m) and for depths below 343.7 m, a separate ice age-scale was derived from the synchronized gas age-scale by adding $\Delta$age estimated from a firn model.

A key contribution from the development of the RICE17 age-scale is evidence of active ice streams in the Eastern Ross Sea during the last glacial cycle. This is supported by the continuous age-scale and continuous records of climate from RICE which imply that the Roosevelt Island ice dome remained stable and independent of WAIS for at least the last 64.6 ka and likely for the last 83 ka.

The RICE ice core provides records of climate, with precise dating, for a scarcely sampled region of Antarctica. Future work will investigate regional climate of the Eastern Ross Sea in comparison to climate records from other sites to better understand spatial patterns around Antarctica, such as during the ACR when Roosevelt Island experienced an interval of particularly low accumulation, and to study the glacial retreat of WAIS in the Ross Sea.

*Code and data availability.* The following will be made available on public archives: RICE prior age control points, RICE final age control points, RICE17 age-scale interpolated at higher resolution, RICE $CH_4$, $\delta^{18}O$-$O_2$, and $\delta^{15}N$-$N_2$ data, and code for the gas synchronization routine.

## Appendix A: Methane Measurements

The methodology used at Oregon State University for measuring methane concentration in ice cores was described in Grachev et al. (2009) and Mitchell et al. (2013). Briefly, 40-60 g ice samples are trimmed and placed in glass flasks. The glass flasks are then attached to a high vacuum line and immersed in a chilled ethanol bath set to -63°C to keep the samples frozen. Since Mitchell et al. (2013), insulation has been added around the ethanol bath and above where the flasks are mounted. The added insulation reduced the temperature and water vapor content of gas in the headspace of the flasks and decreased their variability. Both can affect methane measurements by changing the pressure reading or the retention time of methane in the GC column. We also made efforts to more carefully regulate the amount of ethanol in the chilled bath and temperature of the hot water bath during melting. These steps improved stability of measurements and extraction between days.

After laboratory air has been evacuated, the flask valves are closed and the samples are melted in a hot water bath for 30 minutes to liberate air. Samples are refrozen by immersing the flasks in the cold ethanol bath. Once the samples are completely refrozen and the sample flask temperature has stabilized (approximately 1 hour after refreezing begins), methane measurement is performed by expanding sample air from the flask headspace into a gas chromatograph (GC).

Calibration measurements are made on an internal standard which is referenced to several compressed air standards externally calibrated on the NOAA scale (Dlugokencky et al., 2005). Calibration runs of systematically varying pressures are made both before and after samples are measured. The calibration curve is a least-squares linear regression between pressure and peak area for both sets of calibration runs described by a slope $m_{cal}$ and intercept $b_{cal}$.

Methane concentration ($C$) was previously calculated by comparing the sample pressure ($P_{meas}$) and the peak area ($PA_{meas}$) from GC analysis to the predicted peak area of the standard gas of equal pressure.

$$C_{raw} = C_{standard} \cdot \frac{PA_{meas} - b_{cal}}{P_{meas} \cdot m_{cal}} \tag{A1}$$

Four measurements can be made on each ice sample. If the two sets of calibration runs are taken individually, interpretation of sample concentrations typically differ by <3 ppb.

In this calculation, the sample pressure is used to quantify the number of moles of air in the sample assuming that the sample temperature remains constant and equal to standard air temperature during calibration runs. However, in a series of dry blank experiments it was determined that sample gas temperature is cooler than the standard gas temperature. We estimate the temperature difference to be 0.42% of the standard air temperature and thus the sample pressure needs to be corrected upwards by that amount. We call this the GC-thermal effect. We now apply the correction to the previous concentration calculation.

$$C_{raw} = C_{standard} \cdot \frac{PA_{meas} - b_{cal}}{P_{meas} \cdot 1.0042 \cdot m_{cal}} \tag{A2}$$

Results from typical wet blank samples, in which we add standard air over bubble free ice made from Milli-Q ultra-pure water, had shown that these samples are typically 2-3 ppb enriched compared to the standard concentration using the old method for calculating concentration. This agrees with the predicted magnitude of enrichment from the GC-thermal effect of 2.1 ppb.

Several wet blank samples are measured each day and are interspersed between ice core samples. The offset between the measured concentration of the wet blank and the known concentration of the standard gas added is called the "blank" correction.

This represents any effects during the sample analysis process which may alter the measured concentration. We bin the wet blank results over the time period for which samples were measured to establish a single blank correction value with an estimated blank correction uncertainty. RICE ice core samples were measured during two separate periods and have separate

blank correction values.

Since the OSU analytical methods use a wet extraction technique to liberate ice core air, effects of gas solubility must be accounted for. Methane is more soluble than the major components of air and is preferentially dissolved during the extraction step. This leads to a decrease in the measured concentration compared to the true ice core concentration. Mitchell et al. (2013) empirically derived a solubility correction ($S$) which we repeated several times for the RICE samples. $S$ is defined as the total

amount of methane (gas + dissolved) divided by the amount in gas phase. A solubility value of 1.0165 was used for ice core samples and 1.0079 for bubble free ice. We believe the difference in $S$ for the different ice samples results from the differences between how blank ice and ice containing air behave during melting. Specifically,

  – Bubble-free ice melts slowly in comparison to glacial ice which sometimes melts rapidly and cracks violently. This, along with bubbles rising and breaching the melt water surface, cause disturbances in the water-air interface and promotes

exchange of $CH_4$ into the melt water. This should lead to greater mixing and homogenization of air and water.

  – Bubbles released into the melt water will be at higher pressures than the overlaying air because of surface tension. The higher partial pressure of $CH_4$ in those bubbles, in comparison to the standard gas added over the bubble-free ice, will cause air to go in to solution faster.

  – Because glacial ice tends to be melted sooner than blank ice, a longer time period for liquid-gas exchange is available.

The "blank" and "solubility" corrections are applied in the following way:

$$C_{corrected} = C_{raw} \cdot S_{sample} - (C_{blank} \cdot S_{blank} - C_{standard}) \tag{A3}$$

This formula differs from that used by Mitchell et al. (2013). In Mitchell et al. (2013) no solubility correction was applied to the bubble free ice samples. The difference results in a constant offset of -7.4 ppb compared to Mitchell et al. (2013).

**Appendix B: Calibration of RICE CFA methane**

The RICE CFA methane record is qualitatively used for synchronization to the WAIS Divide methane record, and thus careful calibration was not required. Regardless, we present an ad hoc calibration of the RICE CFA methane record based on comparison to the RICE discrete methane record measured at Oregon State University (OSU). The RICE CFA methane record was measured in multiple campaigns and major adjustments to the analytical system occurred during both of those periods. Calibration of the RICE CFA record is done in a piecewise manner to reflect these changes.

Our calibration scheme accounts for instrument calibration, a concentration-based correction due to instrument sensitivity, and measurement drift, a time-dependent correction. For comparison between datasets, we subsample the CFA data at depths

where discrete measurements were made. We first apply the concentration calibration by regressing the sub-sampled CFA methane concentration against the discrete measurement. Drift was accounted for by a second regression comparing the residual of the concentration calibration against either the measurement time or sample depth, which ever provided the better correlation

to the discrete dataset. Drift was only a significant factor between 500-680 m depth. Calibration values are given in Table A2.

Uncertainty in the relationship may be caused by measurement uncertainty, which is relatively small, or the uncertainty in the depth registration of the continuous measurements. Uncertainty in the depth registration is relatively unimportant in the top $\sim$670 m of core. In this section annual layers are relatively thick and methane variability is relatively low which results in only minor uncertainty in methane concentration from an error in depth. However, for the deeper section of core,

methane concentrations vary rapidly with depth and small errors in the depth registry represents large differences in methane concentration. Because of errors in the depth registry and the sensitivity of the inferred methane concentration, we restrict our calibration scheme to the last $\sim$39.66 ka (725.63 m) ending after the inclusion of GIS-8.

## Appendix C:  Steady-State Herron-Langway Sensitivity Test

The gas age-ice age offset ($\Delta$age) for the RICE ice core was estimated with a dynamic Herron-Langway model (Buizert

et al., 2015b). The model is constrained by measurements of $\delta^{15}$N-N$_2$ (a proxy for firn thickness) and $\delta$D (a proxy for past temperature). The dynamic model also assumes a convective zone of 2 m, surface firn density of 400 kg$\cdot$m$^{-3}$, and geothermal heat flux of 78 mW$\cdot$m$^{-2}$.

A steady-state version of the Herron-Langway model mimics the dynamic version following a similar iterative methodology. A constant $\Delta$age estimate is assumed as a prior to assume an ice age scale for the temperature history. The model then calculates

a new $\Delta$age solution from the temperature and firn thickness histories. This process is repeated until iterations no longer change appreciably.

While the dynamic version is used to establish the $\Delta$age history for RICE, the steady state version has the advantage of being computationally faster and is used in a sensitivity test. In this test, we vary prior assumptions about the isotopic temperature sensitivity used to infer past temperature, the convective zone thickness, surface firn density, and assumptions about geothermal

induced temperature gradient in the firn. In a Monte Carlo approach, each parameter is given a range of acceptable values from which the steady-state Herron-Langway model calculates the $\Delta$age history (Table A1). This is repeated for 6000 realizations. Realizations are rejected if:

  – The modeled modern ice age at lock-in depth (LID) is more than 25 years different than the annual layer counted age of that depth (48.57 m, 89 yrs BP).

– The modeled modern accumulation is less than 0.15 or greater than 0.35 m ice per year.

  – The isotopic sensitivity is less than 3.2 or greater than 9.6 ‰$\cdot$K$^{-1}$, similar to the range of values observed for West Antarctica (Masson-Delmotte et al., 2008; Cuffey et al., 2016; Stenni et al., 2017). This parameter is randomly chosen from a normal distribution at the beginning of each realization and can fall outside of this range.

- The minimum temperature is less than -60°C.

- The maximum estimated $\Delta$age is less than 1000 years.

- Modeled accumulation does not exceed 1.0 m ice per year and is never negative.

$\Delta$age is calculated for each depth that $\delta^{15}$N-N$_2$ was measured. Uncertainty in $\Delta$age is assumed to be the root-mean-square-error of accepted realizations and is estimated for each sample depth.

*Author contributions.*  JEL measured CH$_4$ at OSU on RICE samples. JEL, NANB, T. Baisden, T. Blunier, VGC, EDK, and PV participated in the 2013 or 2014 continuous flow analysis campaigns. JPS measured $\delta^{15}$N-N$_2$ and $\delta^{18}$O-O$_2$ at SIO. JEL, CB, TJF, MW, EB, DDJ, FP, JPS, and EDW contributed to methods and discussion leading to the development of the RICE17 age scale. JEL, HC, TJF, RH, JPS, and
EDW contributed with discussion of glaciological interpretations of the ice core record. All authors provided valuable feedback and made helpful contributions to writing the manuscript.

*Competing interests.*  The authors declare no conflicts of interest.

*Acknowledgements.*  This work is a contribution to the Roosevelt Island Climate Evolution (RICE) Program, with contributions from USA, UK, Sweden, New Zealand, Italy, Germany, Australia, China Denmark, Germany, Italy, China, Denmark, and Australia. The US contribu-
tion was funded by grants from the US National Science Foundation Office of Polar Programs (0944021, 0837883, 0944307, 1042883 & 1643394). The NZ contribution was funded through New Zealand Ministry of Business, Innovation, and Employment grants issued through Victoria University (RDF-VUW-1103, 15-VUW-131), GNS Science (540GCT32, 540GCT12) and Antarctica New Zealand (K049). The Danish contribution was funded by the Carlsberg Foundation's North-South Climate Connections project grant. The research also received funding from the European Research Council under the European Community's Seventh Framework Programme (FP7/2007-2013) ERC
grant agreement 610055 as part of the Ice2Ice project.

We thank Micheal Rebarchik and Will Patterson for laboratory assistance at Oregon State University. We thank Ross Beaudette for laboratory assistance at Scripps Institution of Oceanography. We thank Marius Simonsen, Helle Kjær, Rebecca Pyne, Daniel Emanuelsson, Bernadette Proemse, Ross Edwards, Darcy Mandeno, and the rest of the RICE team for participation in preparing ice samples and operating instruments during the continuous melting campaigns at GNS. We thank the US Antarctic Support Contractor, Antarctica NZ, the US Air Force, the Air National Guard, and Kenn Borek Air for logistical and field support.

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

**Figures**

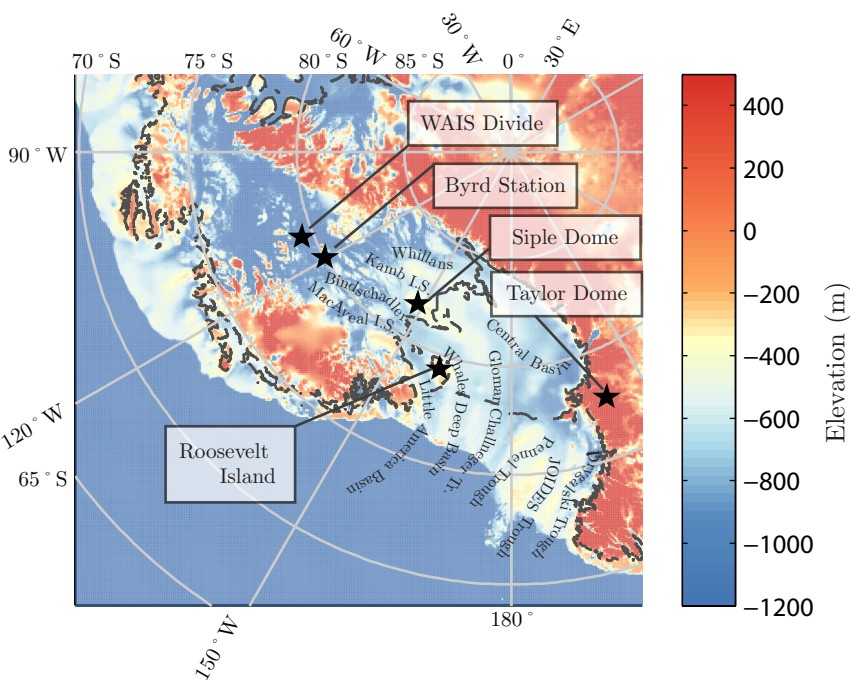

**Figure 1.** Map of bedrock elevation in the Ross Sea Sector of Antarctica (referenced to WGS84 datum) (Fretwell et al., 2013). Gray dashed lines indicate ice sheet grounding lines and ice margins. Locations of the RICE (Roosevelt Island), Siple Dome, Byrd Station, WAIS Divide, and Taylor Dome ice cores are marked with black stars.

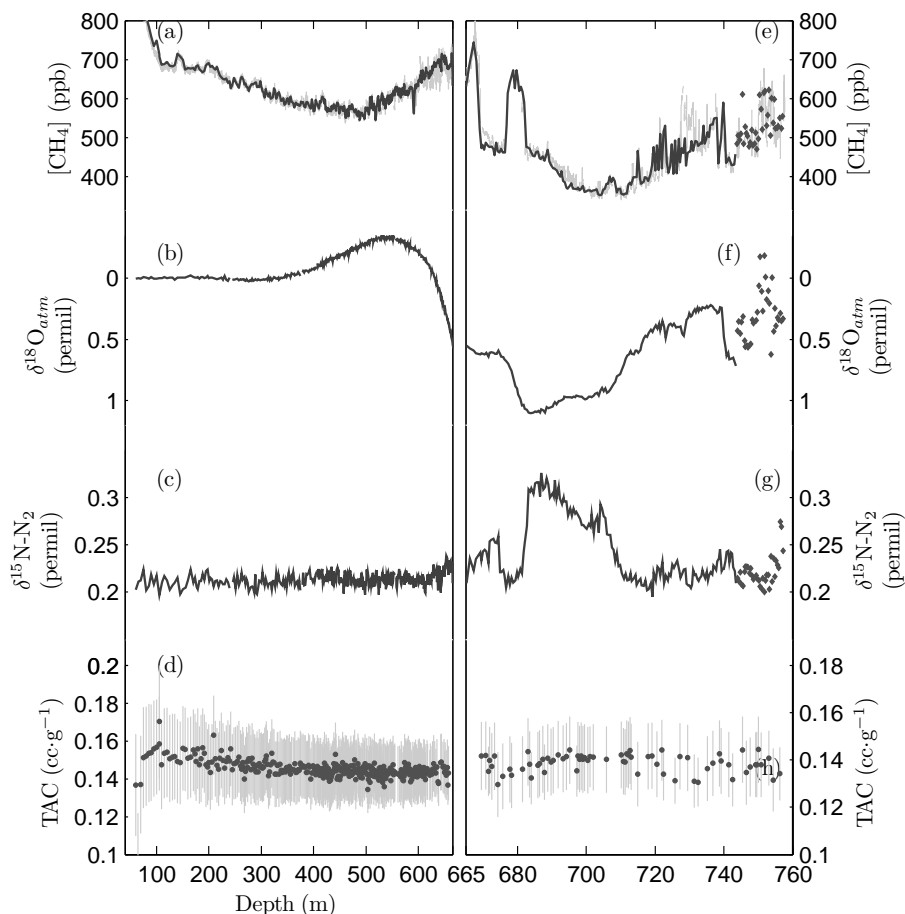

**Figure 2.** Gas data from the RICE ice core. Left panel (a-d) (60-665 m) covers the last 11.26 ka; Right panel (e-h) (665-760 m) covers measurements from 11.26-83 ka and measurements from the 9 m of ice below the dated section. (a, e) Continuous methane measurements, gray, between 0-726 m depth are calibrated to discrete methane measurements, black. Beyond 726 m depth, raw CFA methane measurements are plotted. (b, f) $\delta^{18}O_{atm}$ measurements are corrected for gravitational enrichment in the firn layer using $\delta^{15}$N-N$_2$ (c, g) . (d, h) Total air content (TAC) was measured in conjunction with discrete methane.

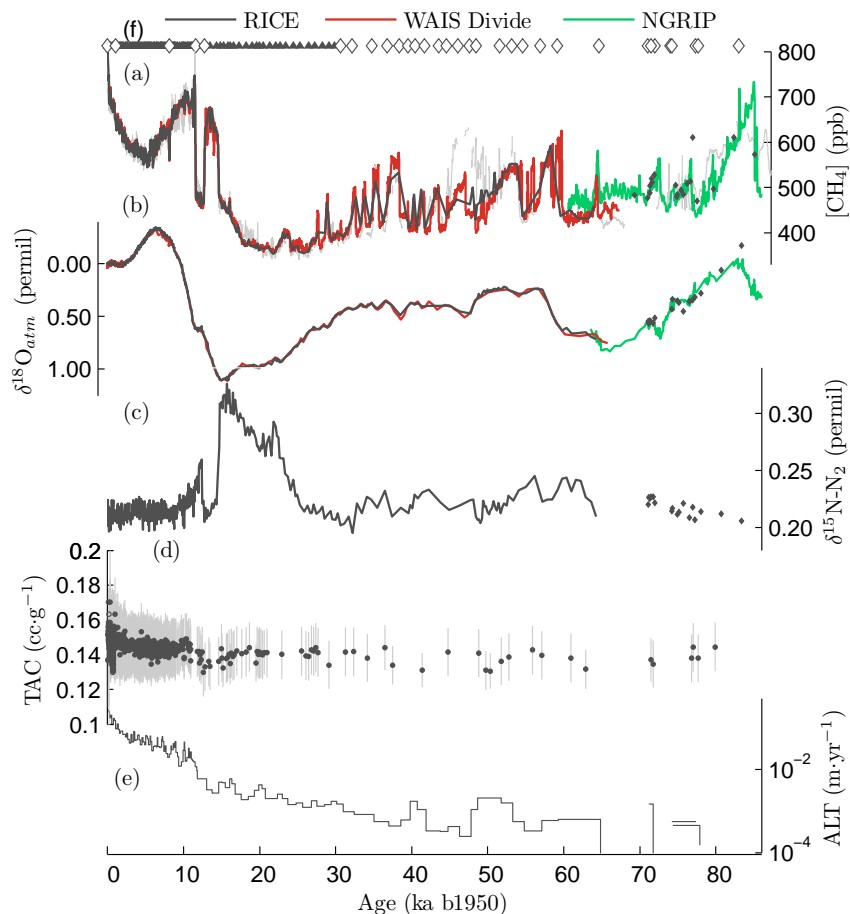

**Figure 3.** Gas data from the RICE ice core plotted on the RICE17 age scale. (a) RICE methane and (b) $\delta^{18}O_{atm}$ records (gray) shown in comparison to target records from the WAIS Divide ice core on the WD2014 age scale (Buizert et al., 2015b) (red) and NGRIP ice core (Baumgartner et al., 2014; Landais et al., 2007) on a modified GICC05modelext chronology (Wolff et al., 2010) (green). Solid triangles above panel (a) are gas age constraints from a Monte Carlo analysis, open diamonds are prior ACPs from visual matching (see text). (c) $\delta^{15}N$-$N_2$ is used in a firn densification model to estimate the ice age-gas age offset. (d) RICE TAC measurements. (e) Mean annual layer thickness calculated from gas age control points adjusted for $\Delta$age.

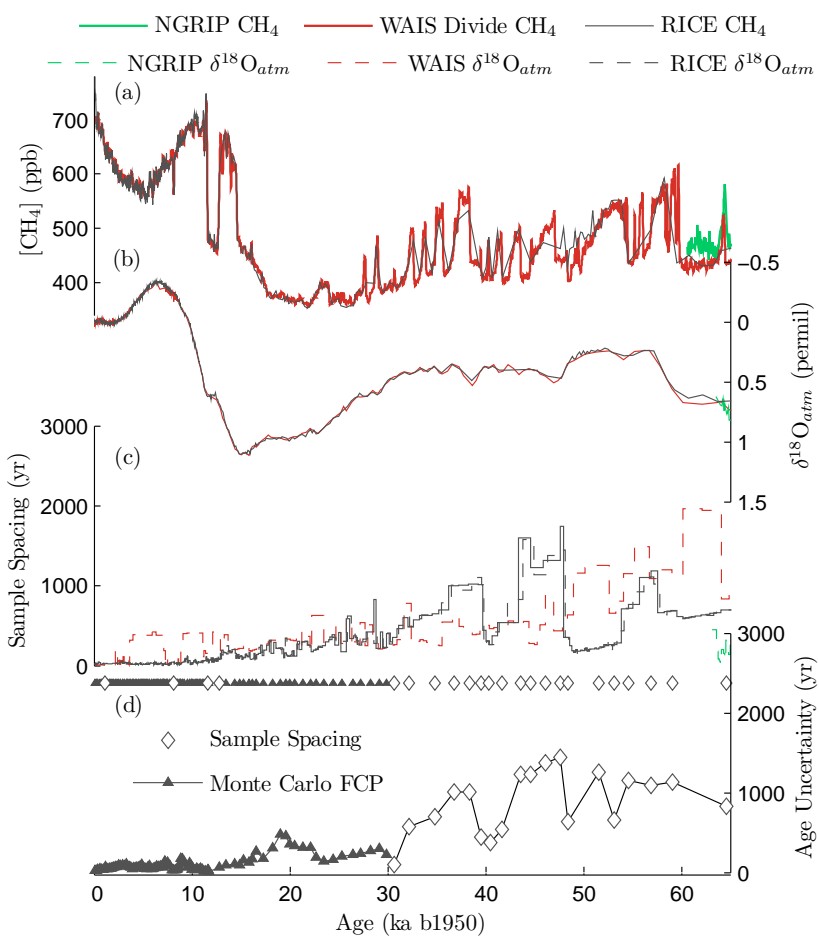

**Figure 4.** RICE ice core age uncertainty. (a) Methane and (b) $\delta^{18}O_{atm}$ records from the RICE (gray), WAIS Divide (Rhodes et al., 2015) (red), and NGRIP ice cores (Baumgartner et al., 2014) (green). (c) Sample spacing for methane (solid lines) and $\delta^{18}O_{atm}$ (dashed lines) for the various cores. (d) Gas age uncertainty, relative to WD2014, for ages determined from a Monte Carlo analysis (solid triangles) and for extended gas age control points (open diamonds).

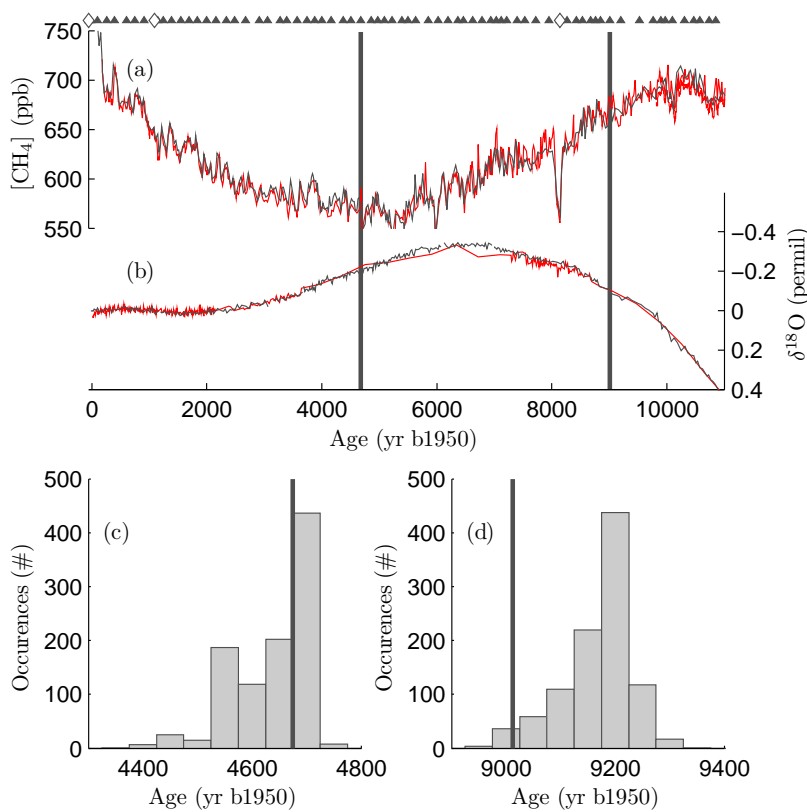

**Figure 5.** (a) RICE methane and (b) $\delta^{18}O_{atm}$ records matched to those from the WAIS Divide ice core (Rhodes et al., 2015; Buizert et al., 2015b) (red lines) for the last 11.7 ka. FCPs from Monte Carlo routine are shown as solid triangles, prior constraints are shown as open diamonds. Lower panels (c, d) show the distribution of the gas ages for two particular depths, 459.05 m (4675 yr BP) (c) and 621.28 m (9012 yr BP) (d), resulting from the Monte Carlo analysis. The final age of these depths, resulting from the best realization, are shown as vertical gray lines.

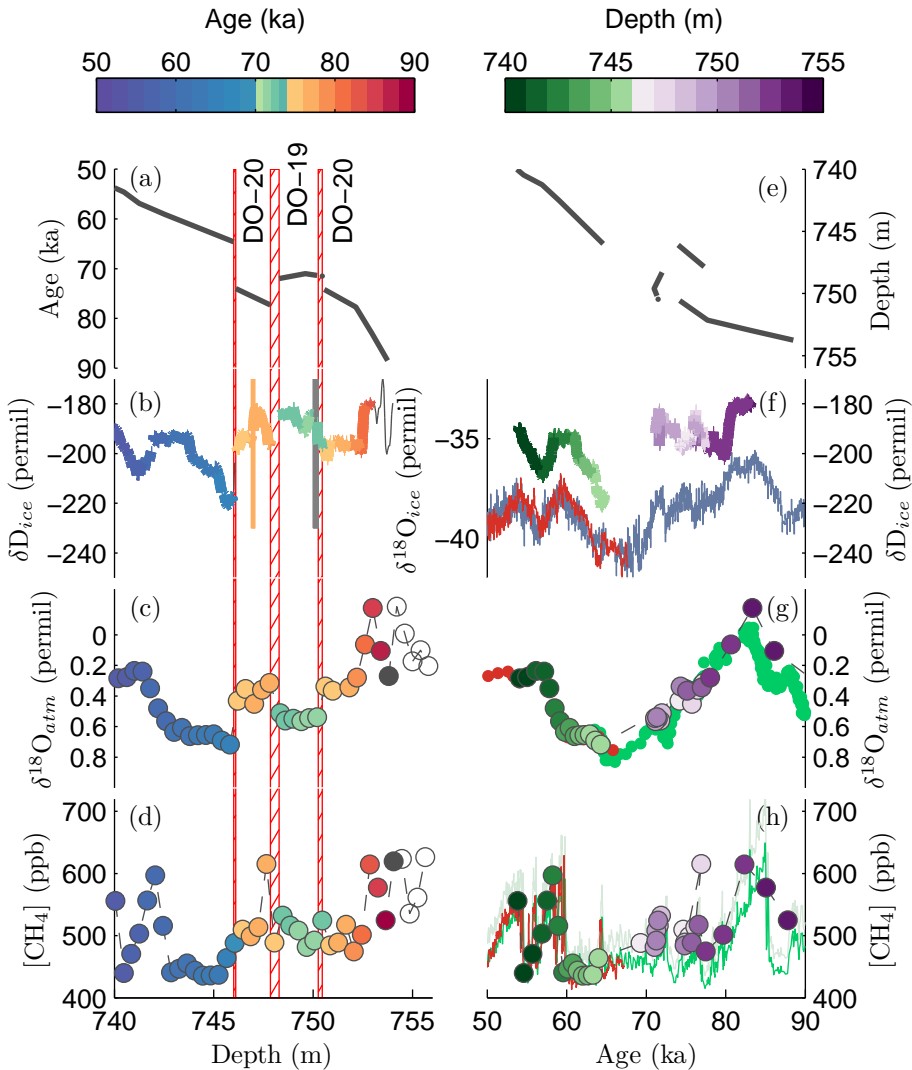

**Figure 6.** (a) Depth-age relationship from 740-756 m and evidence of an age reversal within the RICE ice core from (b) $\delta$D, (c) $\delta^{18}O_{atm}$, and (d) methane data. (b-d) Measurements are plotted against depth and color coded according to the age of the sample. Open circles represent samples for which an age could not be determined. (a-d) Red hatched bars represent discontinuities, where periods of climate history appear to be missing. Solid gray bars in (b) are measurement gaps in the $\delta$D record associated with large changes in $\delta$D. (e-h) The depth-age relationship and measurements are plotted against the age of the sample and color coded according to depth. WAIS Divide data (red), NGRIP (green), and the EDML $\delta^{18}O_{ice}$ record (blue).

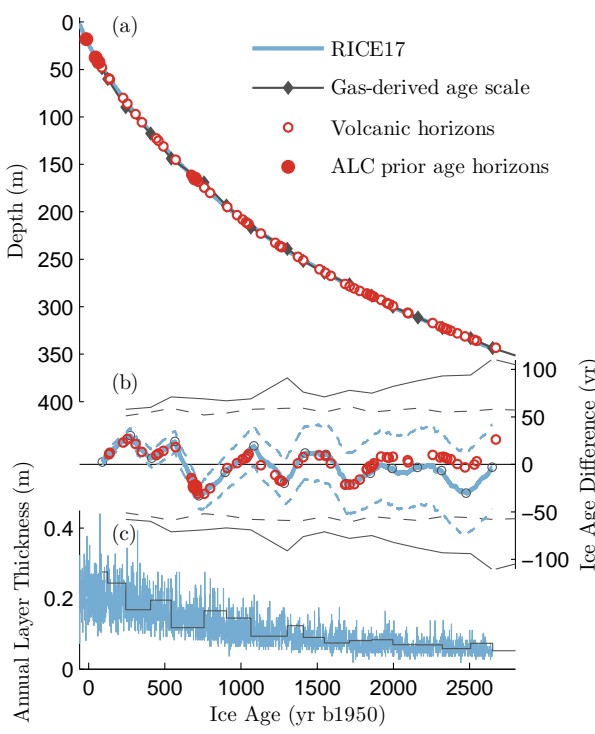

**Figure 7.** Comparison of the gas-derived (gray) and the annual layer counted ice age scale (blue, Winstrup et al. 2017). Six absolute age markers (closed red circles) were identified and an additional 67 volcanic horizons were cross correlated to volcanic horizons in the WAIS Divide core (plotted on WD2014 ice age scale, open red circles, Winstrup et al. 2017). (b) Difference in ice age between the gas-derived ice age scale and annual layer counts; positive values indicate that at the same depth the annual layer counts is older than in the gas-derived age scale. Uncertainty estimates of the gas-derived ice age scale (solid gray lines), of the $\Delta$age estimate only (dashed gray lines), and of the annual layer counted age scale (blue dashed lines). (c) Interpretations of annual layer thickness from the gas-derived ice age scale and annual layer interpretations.

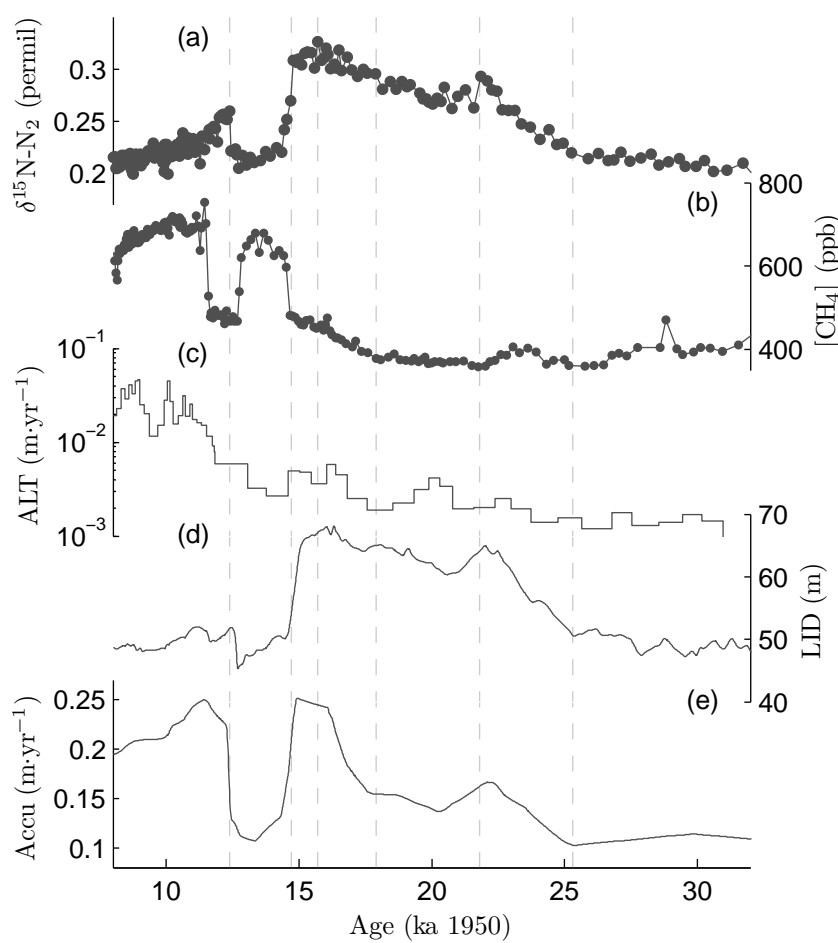

**Figure 8.** Comparison of (a) $\delta^{15}$N-N$_2$, (b) CH$_4$, (c) annual layer thickness implied from depth-age scale, (d) lock-in depth (LID), and (e) accumulation reconstructions from the RICE ice core. Lock-in depth and accumulation is calculated with a dynamic Herron-Langway firn densification model.

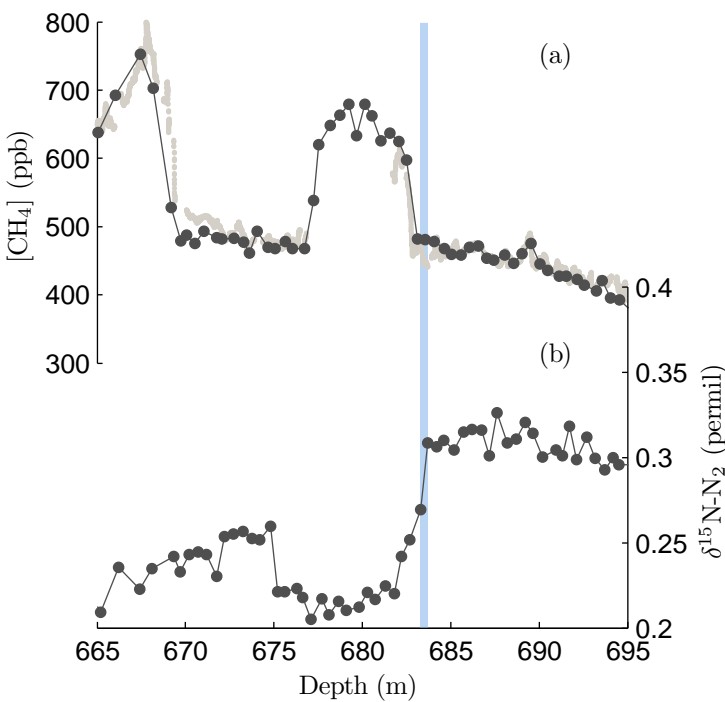

**Figure 9.** Comparison of (a) methane and (b) $\delta^{15}$N-N$_2$ from the RICE ice core plotted on depth. Vertical blue bar highlights the onset of a large decrease in $\delta^{15}$N-N$_2$.

**Tables**

**Table 1.** Prior gas age constraints and uncertainty are based on matching of features in the atmospheric history of methane and $\delta^{18}O_{atm}$. Name or description of feature are given in the notes and the primary parameter used to identify the feature is provided as the identifying variable. Age uncertainty is related to the duration of the feature.

| Depth (m) | Gas_Age (yr) | $\sigma$ (yr) | Identifying Variable | Notes |
|---|---|---|---|---|
| 48.57 | -55.4 | 7 | Modern LID | |
| 239.000 | 1092 | 45 | CH$_4$ | |
| 591.000 | 8140 | 30 | CH$_4$ | 8.2 ka event |
| 669.150 | 11580 | 30 | CH$_4$ | Younger-Dryas — Preboreal |
| 677.300 | 12780 | 57 | CH$_4$ | Bølling-Allerød — Younger Dryas |
| 719.300 | 30660 | 100 | CH$_4$ | GI 5.1 Termination |
| 720.700 | 32150 | | CH$_4$ | GI 5.2 Termination |
| 722.800 | 34780 | | CH$_4$ | GI 7 Termination |
| 723.900 | 36750 | | CH$_4$ | GI 8 Termination |
| 724.600 | 38370 | | CH$_4$ | GI 8 Onset |
| 725.310 | 39530 | | CH$_4$ | Mid-GS 9 Methane Event |
| 726.850 | 40332 | | CH$_4$ | GI 9 Onset |
| 728.090 | 41643 | | CH$_4$ | GI 10 Onset |
| 728.720 | 43544 | | CH$_4$ | GI 11 Onset |
| 729.050 | 44562 | | CH$_4$ | GI 12 Termination |
| 729.680 | 46100 | | $\delta^{18}O_{atm}$ | |
| 730.050 | 47620 | | $\delta^{18}O_{atm}$ | GI 12 Onset |
| 730.950 | 48420 | | $\delta^{18}O_{atm}$ | Mid-GS 12 Methane Event |
| 737.260 | 51570 | | $\delta^{18}O_{atm}$ | |
| 739.650 | 53115 | | $\delta^{18}O_{atm}$ | |
| 740.470 | 54595 | | $\delta^{18}O_{atm}$ | GI 14 Onset |
| 741.250 | 56885 | | $\delta^{18}O_{atm}$ | (MIS 4/3 Transition) |
| 742.550 | 59100 | | $\delta^{18}O_{atm}$ | GI 17.1a Onset (MIS 4/3 Transition) |
| 746.000 | 64600 | | $\delta D, \delta^{18}O_{atm}$, CH$_4$ | Top Depth of discontinuity |
| 746.100 | 74000 | | $\delta D, \delta^{18}O_{atm}$, CH$_4$ | Bottom Depth of discontinuity |
| 747.850 | 77300 | | $\delta D, \delta^{18}O_{atm}$, CH$_4$ | Depth of reversal in $\delta D$ |
| 748.290 | 72000 | | $\delta^{18}O_{atm}$ | First $\delta^{18}O_{atm}$ sample clearly in DO-19 grouping |
| 749.600 | 71000 | | $\delta^{18}O_{atm}$, CH$_4$ | Depth of youngest part of reversal in $\delta D$ |
| 750.460 | 71500 | | CH$_4$ | Deepest sample clearly related to reversal grouping |
| 750.560 | 74200 | | $\delta^{18}O_{atm}$, CH$_4$ | Shallowest sample related to return to DO20 values |
| 752.150 | 77700 | | $\delta^{18}O_{atm}$, CH$_4$ | Last $\delta^{18}O_{atm}$ sample clearly part of DO20 |
| 752.950 | 83000 | | $\delta^{18}O_{atm}$ | Minima in $\delta^{18}O_{atm}$, matching values observed in NGRIP and EDML for MIS 5a |
| 753.750 | 88500 | | $\delta^{18}O_{atm}$ | Enriched $\delta^{18}O_{atm}$, MIS 5b? |

**Table 2.** Model Parameters used for optimized correlation routine. Code for routine can be found in supplementary material.

| Variable | Description | 0-670 m | 670-718.13 m |
|---|---|---|---|
| **Model Parameters:** | | | |
| Runs | # of realizations in Monte Carlo Analysis | 1000 | 1000 |
| N | # of subsections | 76 | 25 |
| $\Delta t$ | Duration of subsections | 154 yrs | 726 yrs |
| k | # of refinements to perturbation | 13 | 13 |
| $n_{rep}$ | # of repetitions before moving to next refinement | 20 | 20 |
| **Perturbation Conditions:** | | | |
| $\overline{A}_{max}$ | Maximum "Accumulation" | 75 cm·yr$^{-1}$ | 75 cm·yr$^{-1}$ |
| $\overline{A}_{min}$ | Minimum "Accumulation" | 0 cm·yr$^{-1}$ | 0 cm·yr$^{-1}$ |
| $\Delta t/\Delta t_{prior}$ | Relative change in duration of subsection from prior | 10x | 10x |
| $\overline{A}_i/\overline{A}_{i-1}$ | Maximum relative change of "accumulation rate" between subsequent subsections | 2x | 2x |

**Appendix Tables**

**Table A1.** Model Parameters used for steady-state Herron-Langway model.

| Description | Mean | $\sigma$ | Distribution |
|---|---|---|---|
| $\delta^{15}$N-N$_2$ | | 0.0027‰ | normal |
| Modern temperature | -23.5°C | 3°C | normal |
| Isotope sensitivity | 6 ‰·K$^{-1}$ | 1.2 ‰·K$^{-1}$ | normal |
| Surface firn density | 400 kg·m$^{-3}$ | .05 kg·m$^{-3}$ | normal |
| Convective zone thickness | 2 m | 2 m | uniform |
| Geothermal induced temperature gradient | -0.6 K | 0.6 K | uniform |

**Table A2.** Calibration of the RICE CFA methane dataset is performed by comparison to the RICE discrete methane dataset with corrections for instrument sensitivity and instrument drift. Calibration is done for different segments of the core. Quality of the fit is described by the $R^2$ statistic comparing calibrated values of the sub-sampled CFA methane record to discrete measurements.

| Depth Range | Instr. | Drift | $R^2$ |
|---|---|---|---|
| 0-500 m | $C_{final} = C_{raw} \cdot 1.1079 + 25.0900$ | | 0.9854 |
| 500-680 m | $C_{final} = C_{raw} \cdot 0.9440 + 10.9337$ | $+ \quad t_{meas} \cdot 4.0181 \cdot 10^{-5} - 1.4013 \cdot 10^{-5}$ | 0.9239 |
| 682-726 m | $C_{final} = C_{raw} \cdot 0.8869 + 29.2036$ | | |

# An 83,000 year old ice core from Roosevelt Island, Ross Sea, Antarctica

James E. Lee[1], Edward J. Brook[1], Nancy A. N. Bertler[2,3], Christo Buizert[1], Troy Baisden[3,4], Thomas Blunier[5], V. Gabriela Ciobanu[5], Howard Conway[6], Dorthe Dahl-Jensen[5], Tyler J. Fudge[6], Richard Hindmarsh[7], Elizabeth D. Keller[3], Frédéric Parrenin[8], Jeffrey P. Severinghaus[9], Paul Vallelonga[5], Edwin D. Waddington[6], and Mai Winstrup[5]

[1]College of Earth, Ocean, and Atmospheric Sciences, Oregon State University, Corvallis, OR 97331, USA
[2]Antarctic Research Centre, Victoria University of Wellington, Wellington, New Zealand
[3]GNS Science, Gracefield, Lower Hutt, 5010, New Zealand
[4]Now at Faculty of Science and Engineering, University of Waikato, Hamilton, New Zealand
[5]Centre for Ice and Climate, Niels Bohr Institute, University of Copenhagen, Copenhagen, Denmark
[6]Department of Earth and Space Sciences, University of Washington, Seattle, WA 98195, USA
[7]British Antarctic Survey, Cambridge CB3 0ET, United Kingdom
[8]University Grenoble Alpes, Centre National de la Recherche Scientifique, Institut de Recherche pour le Développement, Institut des Géosciences de l'Environnement, 38000 Grenoble, France
[9]Scripps Institution of Oceanography, University of California, San Diego, La Jolla, CA 92093, USA

**Correspondence:** James E. Lee (JLee@COAS.OregonState.edu)

**Abstract.** In 2013 an ice core was recovered from Roosevelt Island, an ice dome between two submarine troughs carved by paleo-ice-streams in the Ross Sea, Antarctica. The ice core is part of the Roosevelt Island Climate Evolution (RICE) project and provides new information about the past configuration of the West Antarctic Ice Sheet and its retreat during the last deglaciation. In this work we present the RICE17 chronology, which establishes the depth-age relationship for the top 754 m of the 763 m core. RICE17 is a composite chronology combining annual layer interpretations for 0-343 m (Winstrup et al., 2017) with new estimates for gas and ice ages based on synchronization of $CH_4$ and $\delta^{18}O_{atm}$ records to corresponding records from the WAIS Divide ice core and by modeling of the gas age-ice age difference.

Novel aspects of this work include: 1) an automated algorithm for multi-proxy stratigraphic synchronization of high-resolution gas records, 2) synchronization using centennial-scale variations in methane for pre-anthropogenic time periods (60-720 m, 1971 CE to 30 ka), a strategy applicable for future ice cores, and 3) the observation of a continuous climate record back to ~65 ka providing evidence that the Roosevelt Island Ice Dome was a constant feature throughout the last glacial period.

## 1 Introduction

The Roosevelt Island Climate Evolution (RICE) project seeks to combine geophysical measurements with climate information from a well-dated ice core to improve constraints of the glacial history of the eastern Ross Sea (Conway et al., 1999). With this motivation, the RICE project drilled and recovered a 763 m long ice core from Roosevelt Island. Here, we first present new

data sets from the RICE ice core and then describe the development of the RICE17 age-scale, a composite age-scale which combines annual layer counts (Winstrup et al., 2017) with new age constraints and spans the top 753.75 m ~~764~~.

The most precise chronologies for ice cores have been constructed by combining absolute age markers, for example ~~such as~~ volcanic ash layers, with annual layers counts from visual stratigraphy and from variations in chemical concentrations, stable

isotope ratios of the ice, and electrical properties. Examples include the Greenlandic ice cores composite chronology (GICC05) which extends to 60 ka (Svensson et al., 2008) and the Antarctic WAIS Divide WD2014 chronology through the last 31 ka (Sigl et al., 2016). For the RICE ice core this strategy was only applicable above 165.02 m, the depth of the oldest absolute age marker, at 1251 C.E. (Winstrup et al., 2017). Annual layer interpretations were extended to 343.7 m (2649 yr BP, all ages are reported as years before present (BP), where present is defined as 1950 C.E., Winstrup et al. 2017). Below this depth, annual

layers became too thin to reliably interpret the seasonal signals.

For the 343.7-754 m section of the RICE ice core, the best age constraints are from measurements of methane and the isotopic composition of molecular oxygen ($\delta^{18}O_{atm}$). Variations in these parameters are globally synchronous (Bender et al., 1994; Blunier et al., 1998) meaning that, to a first-order, different ice cores will record the same atmospheric history. We establish the gas-phase depth-age relationship by matching the new RICE records to corresponding records from other ice

cores with established chronologies. From the gas age-scale, an ice-phase age-scale was obtained by modelling the ice age-gas age offset ($\Delta$age). $\Delta$age is constrained by measurements of the isotopic composition of molecular nitrogen ($\delta^{15}$N-N$_2$) and estimates of temperature based on $\delta$D of ice. For overlapping depths the gas-based ice age estimates are in excellent agreement with the annual layer counted age estimates. A final age-scale for the entire core, named RICE17, is presented and is a composite of the annual layer-based age-scale for 0-343.7 m depth and the gas-based age-scale for deeper sections of the

core.

The approach used for RICE17 is not unique but includes several refinements that improve the dating accuracy. Primarily, the RICE17 chronology benefited from the availability of high-resolution gas records from both the RICE ice core and the WAIS Divide ice core. Centennial-scale variability of methane was well-captured for the last 30 ka in both data sets. Additionally, an automated matching routine, adapted from Huybers and Wunsch (2004) and novel in the application to ice cores, simulta-

neously matched methane and $\delta^{18}O_{atm}$ records for ice up to 30 ka (0-720 m). The multi-proxy approach to synchronization leads to strengthened age estimates. Using an automated routine resulted in a more objective match of the RICE gas records to the reference records and an Monte Carlo based estimate of age uncertainty.

RICE17 is a continuous chronology with age monotonically increasing with depth until at least 64.6 ka. At 746.00 m, folding of the ice is observed and the age-scale is segmented and no longer continuous. The oldest ice near the bottom of the core has

a minimum age of 83 ka. We use these observations, in support with measurements of total air content (TAC), to interpret that the Roosevelt Island ice dome was stable throughout the the last glacial maximum.

## 2 The Roosevelt Island ice core and glaciological history

Undertanding the stability of the West Antarctic Ice Sheet (WAIS) is important for predicting future sea level rise (Church et al., 2013; Jevrejeva et al., 2014; Golledge et al., 2014; Pollard et al., 2015; DeConto and Pollard, 2016). Much of the ice sheet is grounded below sea level with the bed deepening towards the center of the ice sheet. This configuration is thought to be
unstable and prone to rapid disintegration due to physical forces related to bouyancy (Hughes, 1973; Weertman, 1974; Schoof, 2007; Feldmann and Levermann, 2015), vulnerability to undercutting by "warm" subsurface currents (Robin and Adie, 1964; Shepherd et al., 2004), and ice cliff instability (DeConto and Pollard, 2016). Vulnerability of WAIS to future warming can be assessed by investigating how it has responded to different climate regimes in the past. Unfortunately, geologic evidence of the past size and extent of Antarctic ice sheets is spatially sparse, tends to have large chronological uncertainty, and is sometimes
contradictory (Whitehouse et al., 2012; Anderson et al., 2014; Bentley et al., 2014; Clark and Tarasov, 2014; Halberstadt et al., 2016; McKay et al., 2016).

The Ross Embayment is the largest drainage of WAIS, both in terms of area and mass loss (Halberstadt et al., 2016). Historically, two scenarios have been proposed for the configuration of WAIS in the Ross Embayment during the Last Glacial Maximum (LGM) (Stuiver et al., 1981). In the "maximum scenario," a thick and grounded ice sheet in the Ross Sea extended
to the continental shelf break (Denton et al., 1989). Details of this scenario are supported by geomorphic features including grounding-zone wedges, which form at the terminus of marine based glaciers (Shipp et al., 1999), and over-compressed diamictons, which are the result of thick overlying ice (Anderson et al., 1984, 1992). Evidence of high stands in the Transantarctic Mountains and the islands of the western Ross Sea (Denton and Marchant, 2000) as well as cosmogenic exposure dates on nunataqs in Marie Byrd Land (Stone et al., 2003) also support this idea. In an alternate scenario, Denton et al. (1989) proposed
that grounded ice in the Ross Sea was kept thin by fast-flowing ice streams. In this scenario, the retreat of WAIS during the last deglaciation may not have contributed significantly to sea level change. Studies of ice cores from Byrd Station (Steig et al., 2001) and Siple Dome (Waddington et al., 2005), glacial modeling (Parizek and Alley, 2004), and cosmogenic exposure dates from the Ohio Range (Ackert et al., 1999, 2007) all support this "minimal scenario."

Roosevelt Island is an ice dome located in the eastern Ross Sea in West Antarctica (79.36° S, 161.71° W; elev. 550 m above
sea level, Fig. 1). It is grounded on a submarine plateau (∼200 mbsl) dividing the Whales Deep and Little America Basins (Fig. 1). During the LGM these troughs were presumably occupied by the extension of the modern Bindschadler and MacAyeal ice streams (Ice Streams D and E, respectively) (Shipp et al., 1999), and Roosevelt Island would have been located along the main ice flow of WAIS.

The 763 m long RICE ice core was drilled to bedrock near the summit of Roosevelt Island. In addition to the main deep core,
a shallow core was drilled to 20 m and several snow pits were sampled to understand recent climate (Bertler et al., 2018). Local mean annual air temperature is -23.5° C and annual snow accumulation is estimated to be ∼0.22 m ice equivalent, based on annual layers in snow pits (Bertler et al., 2018). A cooler estimate of modern temperature, -27.4 ± 2.4° C, based on ERA interim data from 1979-2012 was presented in Bertler et al. (2018), but this estimate is cooler than borehole thermometry measurements and previously published estimates for Roosevelt Island (Herron and Langway, 1980; Conway et al., 1999; Martín et al., 2006)

and does not provide a good fit to the density profile in the firn model. Other estimates of recent accumulation at the RICE drill site range from 0.18 m ice per year to 0.27 m ice per year, depending on method and time period (Winstrup et al., 2017; Bertler et al., 2018; Herron and Langway, 1980; Conway et al., 1999; Kingslake et al., 2014).

# 3 New data sets from the RICE ice core

## 3.1 Methane concentrations

The RICE discrete methane record was measured at Oregon State University (OSU) following methods described by Mitchell et al. (2011, 2013) with updates described in Appendix A. A total of 702 samples were measured at 583 distinct depths between 60-753 m (Fig. 2a, e). Samples from 406 depths were measured between 60 and 670 m, dating from ~1970 C.E. to 11.87 ka, with a mean sample spacing of 28.75 years. Between 670 and 718.13 m the record spans 11.87 to 29.9 ka; 96 samples in this interval provide age resolution of 189 years. Age resolution decreases significantly for deeper depths. The interval from 718.53 to 746.00 m corresponds to 30.1-64.6 ka with a mean resolution of 548 years. The deepest dated ice is at 752.95 m with an age of 83 ka (±2 ka).

Methane was also measured continuously with a laser spectroscopy technique (Stowasser et al., 2012; Rhodes et al., 2013) during two separate continuous flow analysis (CFA) campaigns at GNS Science (Gracefield, New Zealand) in 2013 and 2014 (Pyne et al., 2018). The CFA methane record was affected by variability of air flow to the measurement instrument and fractures that allowed drill fluid and modern air into the melt head. Exclusion of these artifacts caused significant gaps in the record, particularly at depths below 676 m (12.6 ka). We consider the CFA record to be a supplement to the more robust but lower resolution discrete data. Between 29.9-59.1 ka, the CFA record is critical for establishing age control. Below ~746 m (64.6 ka), the CFA data are difficult to interpret because of gaps in the record, uncertainty in measurement depth, and uncertainty in methane calibration (Appendix B).

Several anomalously high discrete methane measurements appear between 44.6-50.9 ka (729.05-736.05 m) and below 64.6 ka (746 m) (Fig. 3a). In the former interval methane is enriched by ~30 ppb compared with the WAIS Divide CFA data (Rhodes et al., 2015). This is likely due to healed fractures that include modern air (Aydin et al., 2010). All RICE samples deeper than 500 m were visually inspected for fractures as they were prepared for measurement and for drill fluid during air extraction. In fractured ice it was common to see drill fluid in the flask as the samples were melted. However, neither observation was a strong indicator that a sample would have an elevated methane concentration. None of these high concentration results were rejected.

## 3.2 Total air content

Total air content (TAC) is defined as the amount trapped air in a sample, reported here in units of $cm^3$ air at STP/g ice. TAC (Fig. 2d, h) was measured at OSU as part of the methane concentration measurement following methodology of Mitchell et al. (2015) and updates from Edwards et al. (in prep). TAC is influenced by accumulation and temperature, total summer insolation

(Raynaud et al., 2007), thermal gradients in the firn from multi-annual climate trends, and surface air pressure (Martinerie et al., 1992; Raynaud and Whillans, 1982).

Preparation of ice samples causes loss of air from bubbles which intersect the surface of the sample, an effect known as the cut-bubble effect (Martinerie et al., 1990). The amount of air lost due to the cut-bubble effect depends on the geometry of the sample and the bubble size. For bubble size, we assume that the bubble diameter at the bubble-close-off depth is the same as at the bubble-close-off depth at WAIS Divide and that the diameter shrinks exponentially to zero at the bubble-clathrate-transition depth (763 m). Ice samples were cut to uniform shapes to limit variability of TAC related to the cut-bubble effect so that TAC can be directly compared between samples. However, inclusion of some fractures was often unavoidable and their contribution to gas loss is potentially large. TAC data were rejected when gas loss was believed to greatly impact the results, such as in samples with visible fractures or samples which consisted of multiple pieces. Of the 706 samples measured at OSU for TAC, 165 results were rejected based upon visual inspection of the sample. Many of these came from the 670-752.95 m (11.7-83 ka) interval where only 58 of 177 TAC measurements are considered reliable. Reproducibility of replicate TAC measurements is 0.7% after application of a correction for the cut-bubble effect (reproducibility of 0.6% without correction). Estimate of uncertainty of TAC measurements is 28% with the largest contributions to the uncertainty from the shape of the sample and the assumed bubble diameter used for the cut-bubble correction.

The TAC record from the RICE ice core without the cut-bubble correction appears remarkably consistent (age weighted mean=0.1182 cm$^3$ air-STP/g ice, age weighted standard deviation=0.0023 cm$^3$ air-STP/g ice, n=410). After application of the cut-bubble correction, the TAC record shows a trend to higher values at more recent ages (Fig. 3d). This trend reflects the decrease of bubble size with depth and therefore a larger cut-bubble correction for shallower samples. Although this trend has a physical explanation, it is not statistically significant due to the uncertainty of the cut-bubble correction. We advise caution when interpreting the absolute values of TAC from the RICE ice core due to the potential artifacts caused by gas loss through fractures and the large uncertainty in the cut-bubble correction.

## 3.3   $\delta^{18}O_{atm}$ and $\delta^{15}$N-N$_2$

$\delta^{18}O_{atm}$ and $\delta^{15}$N-N$_2$ were measured on samples adjacent to the discrete methane samples (Fig. 2b, f and c, g, respectively). Analysis was conducted at Scripps Institution of Oceanography following Petrenko et al. (2006) and Severinghaus et al. (2009). Pooled standard deviation of replicate measurements is 0.006‰ for $\delta^{18}O_{atm}$ and 0.0027‰ for $\delta^{15}$N-N$_2$ (both scales are relative to modern atmospheric composition).

Variations of $\delta^{18}O_{atm}$ are primarily caused by changes in location and intensity of low-latitude rainfall that affect the $\delta^{18}$O of leaf water used in photosynthesis (Severinghaus et al., 2009; Landais et al., 2007, 2010; Seltzer et al., 2017) and changes in seawater $\delta^{18}$O caused by ice sheets on glacial cycles (Horibe et al., 1985; Bender et al., 1985, 1994; Sowers et al., 1993). Importantly, these variations are well known in independently dated ice cores and the atmosphere is well mixed on the relevant timescales, so $\delta^{18}O_{atm}$ variability may be used as a chronostratigraphic marker (Bender et al., 1994). These variations are well sampled in the RICE record until ∼64.6 ka (746.00 m), beyond which the chronology is no longer continuous.

Atmospheric $N_2$ is isotopically stable over very long time-scales (Hattori, 1983; Sowers et al., 1989; Schwander, 1989). Variability of $\delta^{15}N$-$N_2$ in the ice core records primarily reflects changes in gravitational fractionation and thermal fractionation within the firn due to changes in surface temperature and accumulation rate (Schwander, 1989; Sowers et al., 1989). In Antarctica, temperature changes gradually and the effect of thermal separation is minimal.

## 4   Strategy for developing the RICE17 chronology

In this section, we describe how the RICE17 age-scale is developed. We start with age constraints in the gas-phase based on well recorded variations in $\delta^{18}O_{atm}$ and methane. We then discuss how the gas age-scale is used to estimate the gas-phase ice-phase age offset and thus the ice age-scale. Finally, we compare the ice age-scale to the age-scale derived from annual layer interpretations.

All age control points (ACPs), additional control points from the synchronization routine (floating control points, discussed below), and gas and ice ages interpolated to more closely spaced depths are provided in the supplementary material.

### 4.1   An automated matching algorithm for synchronizing ice core records: 0-30 ka

An automated matching algorithm was adapted from Huybers and Wunsch (2004) to synchronize between gas records. Prior ACPs, shown as open triangles in Fig. 3 and Fig. 5, all correspond to well defined increases or decreases of either methane or $\delta^{18}O_{atm}$. We assume that the uncertainty (2 standard deviations) for an ACP corresponds to the time elapsed between 25% and 75% of the change in either methane or $\delta^{18}O_{atm}$ (Table 1). The goal of the routine is to iteratively adjust the interpolation between ACPs to improve a goodness-of-fit parameter by following steps 1-9 described below.

"Goodness-of-fit" is calculated as the $\chi^2$ value comparing the normalized methane record from the RICE ice core to expectant values interpolated from the WAIS Divide ice core plus the analogous $\chi^2$ value comparing the normalized $\delta^{18}O_{atm}$ records. The algorithm can accept additional parameters for synchronization, such as $CO_2$ or $N_2O$, if available. In this analysis we normalized both the methane and $\delta^{18}O_{atm}$ records to have the same mean (5) and variance (1) in order to equally weight their $\chi^2$ values and their importance to the synchronization.

A realization starts by randomly perturbing the age of ACPs within their prescribed uncertainty (Table 1) to define an initial depth-age scale. The perturbed ACPs remain fixed throughout the realization. ACPs are given the subscript $k$, where $k$=1 is the youngest/shallowest ACP. The records are then broken into $N$ subsections which are distributed between ACPs. Subsections are designated with a subscript $i$, where $i$=1,2,3,...N. Floating control points (FCPs) are defined as the bounding depths/ages of these subsections, which will include the initial ACPs. Prior to any perturbations, the durations of the subsections ($\Delta t$) are approximately the same. We have chosen $\Delta t$ to roughly match the recurrence interval of variations of methane in the reference record and so that each subsection contains about five methane and five $\delta^{18}O_{atm}$ samples. The following steps are repeated to optimize goodness-of-fit:

1. Random scaling factors ($p_i$), which perturb the durations of the subsections, are drawn for each subsection from a normal distribution of $\mu$=1 and $\sigma_1$=0.25.

$$\Delta t'_i = \Delta t_i \cdot p_i$$

2. Because random perturbations will change the length of time between the initial ACPs, we apply a second scaling so the perturbed chronology remains consistent with the prior ACPs. In this case, $ACP_k$ and $ACP_{k+1}$ are the nearest gas ACPs bounding subsection $i$, respectively. $ACP'$ is the perturbed age of the gas age constraint after step 1, and $\Delta t^*_i$ is the duration of the subsection after the second scaling:

$$\Delta t^*_i = \Delta t'_i \cdot \left(\frac{ACP_{k+1} - ACP_k}{ACP'_{k+1} - ACP'_k}\right)$$

3. Mean "accumulation rate" of each subsection ($\overline{A}_i$) is calculated following Nye (1963), which assumes that the thickness of annual layers ($\overline{\lambda}_i$) is the product of their original thickness and their relative depth ($\frac{\overline{z}_i}{H}$):

$$\overline{A}_i = \overline{\lambda}_i \cdot \left(1 - \frac{\overline{z}_i}{H}\right)^{-1}$$

where $\overline{z}_i$ is the mid-depth of the subsection $i$, H is the thickness of the ice sheet, and both $\overline{z}_i$ and H are in ice equivalent units.

Because the assumptions about thinning from Nye (1963) is too simple to describe the flow conditions at Roosevelt Island, we do not consider this result to be representative of the true accumulation history. The assumption is necessary for the purposes of interpolating age versus depth to ensure that annual layer thickness decreases smoothly between FCPs. An accumulation history that we believe is more accurate is calculated below with an alternative method, using a dynamic firn model (Section 4.4).

4. A perturbed chronology is only accepted if:

    (a) Intervals between FCPs ($\Delta t^*_i$) are within a factor of 10 of the initial durations,
    $$\frac{1}{10} \cdot \frac{(t_{end} - t_1)}{N} < \Delta t^*_i < 10 \cdot \frac{(t_{end} - t_1)}{N}$$

    (b) Mean "accumulation rates" are realistic ($\overline{A}_i > 0$ cm ice eq per yr, $\overline{A}_{max} < 75$ cm ice eq per yr), and

    (c) Mean "accumulation rates" of adjacent subsections are within a factor of 2 of each other.

These conditions provide loose restrictions for continuity in the depth-age relationship although they may not be physically realistic for the site.

5. Ages for all RICE sample depths are interpolated from the perturbed FCPs (Step 3), assuming piece-wise constant accumulation between FCPs and a linearly decreasing thinning function (Nye, 1963).

6. Goodness-of-fit is calculated on the perturbed age-scale (goodness-of-fit = $\chi_{CH4} + \chi_{\delta 18Oatm}$).

7. When a perturbation improves the goodness-of-fit, that iteration becomes the base for subsequent perturbations.

8. The cycle is repeated until 20 sequential perturbations fail to improve the goodness-of-fit in step 6.

9. Starting with the FCPs from step 8, the above steps (1-8) are repeated 13 times, reducing the size of perturbations in step 1 from $\sigma_1 = \frac{1}{4}$ to $\sigma_{13} = \frac{1}{128}$.

A Monte Carlo analysis repeats these steps (Steps 1-9) 1000 times, initiated from a different prior depth-age relationship by randomly perturbing ACPs within their age uncertainty (Table 1). Parameters used for the synchronization are given in Table 2 for the 60.05-670.13 m and 670.13-719.30 m intervals. Code for the synchronization routine is provided as supplementary material.

We choose the best realization for gas age constraints for the RICE17 gas age-scale. The best age estimate (realization) is not necessarily the most frequent age estimate. Fig. 5d shows an example from sample depth 621.28 m where there is a large difference between these two age estimates. Large differences can occur because the prior age estimate (i.e. the age estimate based only on prior ACPs) differs by a large amount from the "true" age of that sample and because the goodness-of-fit parameter considers the fit over the whole record. For depth 621.28 m most realizations resulted in an age estimate of 9200 yr BP, similar to its prior age estimate of 9,240 yr BP, but the best realization estimated the age to be 9012 yr BP. There are two possible reasons for this type of result: 1) this realization is the best because it managed to push the age of this sample by >200 years towards younger ages while not significantly changing the ages of nearby sections which already fit well, or 2) no significant improvement in the goodness-of-fit was found by adjusting the age of this depth, and the goodness-of-fit was dictated by other sections of the record.

Uncertainty in gas age constraints is calculated as the root-mean-square error of FCP ages from the 1000 Monte Carlo realizations (Fig. 4d), although we acknowledge that this can overly simplify the empirical distribution of possible ages for a given depth (Fig. 5). Assessing uncertainty in this way integrates two types of error. The first is the ability to exactly match timing of two features. This uncertainty is determined by how well features are resolved in the records, measurement error, and the limited degrees of freedom in synchronization (i.e. the synchronization routine assumes constant accumulation during each subsections in a piece-wise manner, although the true accumulation history is more likely to smoothly vary). The second type of error is analogous to deciding which feature in the reference record is the correct match. For example, the methane peak centered at 459.05 m (4675 yr BP) is, in some realizations, matched to a peak at ∼4550 yr BP in the WAIS Divide record, providing two distinct age possibilities (Fig. 5c).

## 4.2 Extending the chronology with visually matched gas age control points: 30-83 ka

The more limited resolution of the RICE methane record below 719.3 m (30.66 ka) prevented use of the automated synchronization routine. Gas age control points were visually chosen between 719.30 m (30.66 ka) and 746.00 m (65.6 ka) and between 746.00 and 752.95 m (∼83 ka) (Table 1). Between 719.30 (30.66 ka) and 746.00 m (65.6 ka), ACPs were visually chosen by comparing the RICE discrete and CFA methane records and the $\delta^{18}O_{atm}$ record to the WAIS Divide methane (Rhodes et al., 2015) and $\delta^{18}O_{atm}$ records (Buizert et al., 2015b).

The WAIS Divide records and the WD2014 chronology end at 67.8 ka. For the deeper (older) section of the RICE ice core, between 746.00-752.95 m, we build target records from records of methane and $\delta^{18}O_{atm}$ from the NGRIP ice core

(Baumgartner et al., 2014; Landais et al., 2007, 2010). Using the NGRIP ice core is internally consistent with the WD2014 chronology, which is tied to a modified GICC05modelext age-scale between 30-67.8 ka (Buizert et al., 2015b). Because NGRIP is in the northern hemisphere the methane concentration is higher than that in the RICE record. To account for this interpolar difference we scale the NGRIP target methane record (Baumgartner et al., 2014) to Antarctic values using an empirical least
squares fit between WAIS and NGRIP records between 55-67.8 ka.

Buizert et al. (2015b) found that the annual layer counted portion of the GICC05modelext chronology (0-60 ka) (Andersen et al., 2006; Svensson et al., 2008) is systematically younger than ages of corresponding features found in the U/Th absolute dated Hulu speleothem record (Reimer et al., 2013; Southon et al., 2012). A fit to Hulu ages was optimized by scaling the GICC05modelext ages linearly by 1.0063. This suggests that on average 6.3 out of every 1000 annual layers were not counted.
For our NGRIP-based target records we adjust the NGRIP age-scale by adopting the scaling of Buizert et al. (2015b). Older ages in the GICC05modelext chronology are derived from the ss09sea Dansgaard-Johnsen model (Johnsen et al., 2001; NGRIP Community Members, 2004). We have added a constant 378 years (0.0063*60,000) for depths older than 60 ka in the target GICC05modelext ages to make this section continuous with the adjusted annual layer counted section.

Synchronization requires well-sampled identifiable features in both records. We estimate age uncertainty for visually matched
ACP's (ACP's older than 29.9 ka) as the larger of the sample spacing of the two records being synchronized, following methods of Brook et al. (2005). Gas age uncertainty is plotted in Fig. 4d. The largest uncertainty is ∼1500-1700 years and occurs between 41.7-47.6 ka, where the age-scale is very compressed and the RICE records are poorly sampled.

## 4.3 Age control points in the disturbed ice: 746-753 m

Continuity of the RICE ice core appears to end at 746.00 m below surface, where a discontinuity of 20‰ is observed in the $\delta$D
record (746.00-746.10 m) (Fig. 6b) and a 0.35‰ change is observed between sequential $\delta^{18}O_{atm}$ samples (745.81 and 746.20 m) (Fig. 6c). The continuous chronology described in the previous section dates this ice to the end of DO-18 (∼64.6 ka). This age is supported by very negative $\delta$D values and a trend to more enriched values at shallower depths which is consistent with warming trends observed in other Antarctic ice cores at this time period (Buizert et al., 2015b; EPICA Community Members, 2006; Petit et al., 1999; Parrenin et al., 2013) (Fig. 6f).
Figure 6 presents our best effort to date the ice in the disturbed section, between 746.00 and 752.95 m. The left panel of Fig. 6 shows the data as a function of depth, color-coded by age. The right panel shows the reconstructed time history of these variables, color-coded by depth. Below, we discuss how the ages are assigned to different depth sections (divided by vertical red lines in the left side of Fig. 6), starting with the shallowest portion. Dating folded ice remains challenging, and the solution we found may not be unique.
Immediately below the discontinuity, between 746.10-747.85 m, methane and $\delta^{18}O_{atm}$ values best match our target records during DO-20 between 74.5-77.3 ka (Fig. 6), although the RICE methane record appears ∼30 ppb too high. Dating of this section to DO-20 implies that either 9.4 ka of climate history is missing from the RICE ice core because climate was not recorded or that 9.4 ka is compressed into ∼10 cm of ice. Extremely thin layers can be explained by ice flow or by an absence

of accumulation, but the latter scenario would cause a collapse of the firn and $\delta^{15}$N-N$_2$ values approaching 0‰ which are not observed.

A cluster of samples between 747.85-750.46 m are ∼0.16‰ more enriched in $\delta^{18}$O$_{atm}$ than was observed in the shallower section dated to DO-20 (Fig. 6c). In the NGRIP, EDML, and Siple Dome ice cores, a long-term trend in $\delta^{18}$O$_{atm}$ towards more enriched values is observed from 80 to 65 ka (Capron et al., 2010; Severinghaus et al., 2009; Landais et al., 2007) (Fig. 6g). The enriched values between 748.34-750.46 m most likely indicate that this ice is younger than the adjacent shallower depths and the stratigraphic order is reversed. This depth range is best matched to DO-19 between 71.8-74.3 ka (Fig. 6). Below 750.46 m, methane and $\delta^{18}$O$_{atm}$ return to values that best match DO-20.

We note that two significant measurement gaps exist in the $\delta$D record, at 746.89-747.07 m and at 750.00-750.25 m (gray bars in Fig. 6b). A 12‰ shift in $\delta$D accompanies both of these gaps. The second measurement gap (750.46-750.56 m) is at nearly the same depth which separates the samples dating to DO-19 from the grouping immediately below dated to DO-20. The discontinuity of the $\delta$D record at these gaps may signify another hiatus in the climate record or highly contorted layers that are typically found around folds (Cunningham and Waddington, 1990; Alley et al., 1997; Thorsteinsson and Waddington, 2002; Waddington et al., 2001).

Below 750.46 meters trends in methane and $\delta^{18}$O$_{atm}$ indicate that the age of ice increases with depth until at least 753 m (∼83 ka). Age of this depth is constrained by a measured $\delta^{18}$O$_{atm}$ value of -0.175‰. Such a depleted value is rare and only occurs during periods of high sea-level and small ice sheets, the most recent time period prior to the Holocene being MIS-5a (80-85 ka) (Severinghaus et al., 2009; Capron et al., 2010; Petit et al., 1999; Landais et al., 2007; Kawamura et al., 2007). Negative values were observed at two other depths below 753 m (Fig. 2f), but the stratigraphic order of these depths is difficult to assess.

## 4.4  Firn modelling to determine Δage

Air transport in the firn causes an age difference between ice and air trappend in the ice, commonly referred to as Δage. Firn densification models are typically used to simulate past Δage (Schwander et al., 1997; Goujon et al., 2003). Input parameters (for example, temperature, accumulation rate, and surface density) are normally the larges sources of uncertainty. Using $\delta^{15}$N-N$_2$ as a proxy for past firn column thickness and assuming $\delta$D records past site temperature faithfully, firn densification models can be run in an inverse mode to estimate both past Δage and accumulation rates (Buizert et al., 2014, 2015b), and is the approach we employ here.

We use a dynamic version of the Herron-Langway model (Herron and Langway, 1980), which was also used for construction of the WD2014 chronology (Buizert et al., 2015b). The model simulates firn compaction rates and vertical heat diffusion and advection. The model domain is the full 763 m ice column at 0.5 m resolution; a time step of 1 year is used. The model simulates both gravitational enrichment of $\delta^{15}$N-N$_2$ and fractionation in the presence of thermal gradients. The model is forced with a temperature history derived from the $\delta$D record assuming a constant isotope sensitivity of 6‰·K$^{-1}$ (Brook et al., 2005). In conjunction with an assumed geothermal heat flux of 78 mW·m$^{-2}$, this provides a good fit to the measured borehole temperature profile (Clemens-Sewall et al., Unpublished). This sensitivity is similar to that observed for West Antarctica (Stenni

et al., 2017; Cuffey et al., 2016; Masson-Delmotte et al., 2008) but somewhat higher than that obtained by comparing the RICE isotope record to ERA interim data (Bertler et al., 2018). However, lower sensitivities decreased the fit of the modeled borehole temperature profile. The dependency of $\Delta$age on the assumed isotope sensitivity was explored in a model sensitivity test (Appendix C), and is incorporated into the $\Delta$age uncertainty estimates. The model further assumes a constant ice thickness,

a constant 2 m convective zone similar to other high accumulation dome sites with low mean wind speeds (Kawamura et al., 2006; Landais et al., 2006), and surface firn density of $400 \, \text{kg} \cdot \text{m}^{-3}$ to match the modern surface firn density.

In a first iteration, we assume a prior ice age-scale for the temperature history by adding a constant 150 years to the gas chronology (Section 4.1 and 4.2). A new $\Delta$age solution is then calculated using the dynamic firn densification model. The $\Delta$age solution (and thereby the ice age-scale) is refined iteratively until it no longer changes appreciably (consistent within ~1

year).

The climate at Roosevelt Island, with high accumulation and relatively warm temperatures, results in small $\Delta$age values and consequently relatively low absolute uncertainty in $\Delta$age compared to most Antarctic sites. Modern $\Delta$age (estimated at 60 m depth, the shallowest measurement of $\delta^{15}$N-N$_2$) is 146 years with a reconstructed lock-in depth (LID) of 48 m, consistent with the modern density profile (Bertler et al., 2018; Winstrup et al., 2017). Holocene $\Delta$age is small, ranging between 140 to 182

years. During the last glacial period simulated $\Delta$age values fluctuate between ~150-350 years.

Uncertainties in past $\Delta$age include the uncertainty in model inputs as well as the model itself. The uncertainty of the Herron-Langway model is conservatively estimated to be 20%, based on differences between firn models (Lundin et al., 2017). We assessed the uncertainty due to model inputs using a steady-state Herron-Langway model that approximates the dynamic version but requires less computational time (Appendix C). In a sensitivity test, we randomly perturb the parameterizations

of temperature and LID and assumptions of convective-zone thickness, surface firn density, and geothermal heat flux and recalculate the $\Delta$age solution (model parameters and their base values and ranges used in the sensitivity test are provided in Table A1). A total of 6000 iterations were run. The sensitivity tests include a wide range of temperature histories to account for the possibility that some variations in $\delta$D were caused by non-thermal effects such as variability in precipitation seasonality, moisture sources and pathways, and post-depositional vapor exchange. Model sensitivity is reported as the root-mean-square-

error of $\Delta$age calculations for each depth. Combined $\Delta$age uncertainty, provided in the included age-scale, is the root-sum-squares of the model uncertainty and the model sensitivity.

### 4.5 Comparison of gas-derived and layer counted chronologies: 0-2,649 yrs BP

The gas-derived ice age-scale provides a chronology from 60 to 753 m depth (Fig. 2) that is independent of but overlaps the annual layer counted section (0-343.7 m, Winstrup et al. 2017). Figure 7 shows both chronologies and differences between

them (positive values indicate that the annual layer counts are older than the gas-derived ages). The gas-derived ice age-scale agrees well with the annual layer counted age-scale, within 33 years, with similar trends in the implied annual layer thickness (Fig. 7c). Differences between chronologies can result from error in the synchronization of gas records, calculation of $\Delta$age for either the RICE or WAIS Divide ice cores, interpolation between ACPs, or annual layer counts in either core (because the WAIS Divide age-scale is used as a reference for the gas age-scale).

The average age difference at depths of FCPs is -1 years (n=18, implied age from gas-derived ice age-scale being older than the layer counted chronology). Root-mean-square of the age difference is 17.3 years. Discrete points in the gas-derived ice age-scale can also be compared to the age of 67 volcanic peaks identified in the RICE ice core and correlated to peaks identified in the WAIS Divide ice core (open red circles in Fig. 7, Winstrup et al. 2017). Compared to these volcanic peaks, the

root-mean-square ice age difference of the gas-derived ice age-scale is 13.6 years from their WD2014 ages. Good agreement between the two approaches gives confidence in the methodology used for the deeper section of the RICE17 chronology. The two largest differences occur at 89.72 m (243 yr BP) and at 169.11 m (757 yr BP) and are +30 years and -33 years, respectively (Fig. 7b). These offsets are similar in magnitude to the individual uncertainties in calculating $\Delta$age or in synchronizing the gas records. The small age differences between the two ice chronologies also indicates that our approach to calculating uncertainty

is likely conservative.

The RICE17 time scale transitions between the annual layer counted and gas-derived age-scales at 343.7 m, the deepest/oldest FCP for which annual layers were identifiable (Section 4.1). The age difference at this depth is 3 years, with the gas-derived ice age implying an older age than the annual layer counted chronology. This age difference is much less than the respective age uncertainties of 45 years and 111 years (2-$\sigma$) for the annual layer counts and gas-derived ice age-scale,

respectively. Because of the good agreement between the two ice age-scales, no correction is made for the 3 year offset and the annual layer counted age is used.

## 5    Results and key observations from the RICE17 chronology and gas data sets

### 5.1    Implied accumulation history of Roosevelt Island

From 65 to 32 ka, the low-variability of $\delta^{15}$N-N$_2$ ranging from 0.20 to 0.24‰ and the cooling trend observed for Antartica for

this time period suggest a decreasing trend in accumulation (Fig. 3).

While $\delta^{15}$N-N$_2$ values appear steady from 65 to 32 ka and during the Holocene (Fig. 3c), they exhibit large variability between 32 and 10 ka (Fig. 8a). $\delta^{15}$N-N$_2$ generally trends to heavier values beginning at 32 ka until 14.71 ka, indicating a growing firn column. An inflection point is observed at 25.3 ka which is interpreted as an acceleration of firn thickening concluding in a peak of 0.293‰ at 21.8 ka. After a short depletion the steep trend resumed with a second $\delta^{15}$N-N$_2$ maximum

of 0.326‰ reached at 15.7 ka. Following this maximum, $\delta^{15}$N-N$_2$ abruptly decreased from 0.308‰ to 0.220‰ at 14.71 ka, which corresponds with the beginning of the Antarctic Cold Reversal (ACR). At 12.38 ka, after the ACR, $\delta^{15}$N-N$_2$ partially recovers to pre-ACR values with an abrupt increase to 0.260‰.

Between 25.3-32 ka, accumulation is estimated to be ∼10 cm ice equivalent per year (Fig. 8e) and the increasing firn thickness is largely attributable to decreasing temperature. After 25.3 ka accumulation starts to increase and by the first $\delta^{15}$N-

N$_2$ maximum (21.8 ka), accumulation had increased to ∼17 cm ice equivalent per year. This feature is not apparent in other ice cores from the Ross Sea region, but those records tend to be difficult to interpret because of chronological uncertainties, such as is the case for Taylor Dome (Baggenstos et al., 2017), or because of unexplained jumps in $\delta^{15}$N-N$_2$, such as is the case for Siple Dome (Severinghaus et al., 2009). By the second maximum (15.7 ka) accumulation increased to ∼25 cm ice equivalent

per year, similar to accumulation rates observed during the Holocene. An accumulation peak at the end of glacial terminations is consistent with evidence from trimlines in interior WAIS (Ackert et al., 2013) and is also observed in the accumulation histories from WAIS Divide and Siple Dome (WAIS Divide Project Members, 2013; Waddington et al., 2005), but to a smaller degree. The early accumulation peak at 21.8 ka is unique to the RICE ice core.

A large, rapid depletion in the $\delta^{15}$N-N$_2$ is observed at 14.71 ka with low values lasting until 12.38 ka. Analysis of the lead-lag relationship between $\delta^{15}$N-N$_2$ and methane has been used to infer near-synchronous climate changes throughout the tropics and northern hemisphere (Rosen et al., 2014). These climate events are believed to originate in the northern hemisphere and propagate to the southern hemisphere (Blunier et al., 1998; Blunier and Brook, 2001; Buizert et al., 2015a; Buizert et al., 2018). Curiously, the abrupt decrease in $\delta^{15}$N-N$_2$ at ~~12.38~~ 14.71 ka (0.088 permil, 683.70-681.80 m) is observed in samples

deeper than the increase in methane marking the onset of the Bølling-Allerød (14.66-14.52 ka, 683.13-682.52 m) meaning that this climate event unambiguously precedes the Northern Hemisphere event (Fig. 9). Where as Rosen et al. (2014) specifically considered the thermal component of $\delta^{15}$N-N$_2$, the thermal component is considered to be negligible in Antarctic cores because air temperature only changes gradually. At Roosevelt Island, this period of low $\delta^{15}$N-N$_2$ values is interpreted to represent shallow firn thickness which in the firn model is caused by a large reduction of snow accumulation (<10 cm/yr, Fig. 8e). Low

accumulation during this period is consistent with thin annual layers interpreted from the age-depth relationship (Fig. 3d); 0.3-0.6 cm/yr annual layer thickness compared with 1.6-3.2 cm/yr between 10.09-11.01 ka (642.75-661.07 m). Following the ACR, the modeled accumulation rate fully recovers to the ∼25 cm per year observed before the ACR.

    We propose three potential explanations for the thin annual layers and low accumulation rate observed during the ACR, with the last explanation being our preferred. 1) A large accumulation gradient is observed across the Roosevelt Island ice

dome (Winstrup et al., 2017) which implies that a period of low accumulation at the RICE site may be the result of changes to the geometry of the Roosevelt Island ice dome. However, ice divide migration typically occurs over long timescales. 2) Accelerated ice flow may also cause thin annual layers and could potentially even affect layers within the firn. This flow could be the result of the ice streams which surrounded and buttressed Roosevelt Island ice dome suddenly being removed (Ackert et al., 1999, 2013; Halberstadt et al., 2016). The timing of the low accumulation interval is similar to when dust records from

the ~~an ice core from~~ Taylor Dome ice core (Aarons et al., 2016), a site located in the Transantarctic Mountains near the Ross Sea (Fig. 1), show an increase in dust diameter and a change in radiogenic isotope composition which implies that the dust source changes to a local source. Aarons et al. (2016) interpret this source is newly exposed land created by the withdrawal of ~~imply that the spatial extent of the~~ Ross Ice Shelf ~~withdrew~~ . The timing of the low-accumulation period is also ~~and is~~ similar to some estimates of the timing of retreat of the WAIS based on sediment facies succession and radiocarbon dating (minimum

date of 8.6 ka, McKay et al. 2016) and ice-sheet modeling (Golledge et al., 2014; McKay et al., 2016). Although annual layer thickness immediately above the ACR section is observed to be nearly five times as thick as the ACR section, this large change in thickness, if solely the result of changes in ice flow, would require an unrealistic change in the thickness of the dome or its ice flow which is not supported by TAC measurements or ice-flow modeling.

    3) Our preferred explanation is that an interval of low accumulation between 12.38 and 14.71 ka resulted in a shallow firn

structure and depleted $\delta^{15}$N-N$_2$ values. In recent times, moisture arriving at Roosevelt Island is frequently related to enhanced

cyclonic air flow in the Ross Sea and a strong Amundsen Sea Low (Tuohy et al., 2015; Emanuelsson et al., 2018). This period of low accumulation may indicate a changed atmospheric structure in the South Pacific where southward air flow is blocked by a persistent low pressure zone north of the Ross Sea such as observed in more recent periods in the accumulation record from RICE (Bertler et al., 2018; Emanuelsson et al., 2015) and potentially related to the past ice shelf extent. Such a

pronounced minimum in accumulation is not observed in the Siple Dome ice core, where annual layers are observed to thicken during the ACR (Brook et al., 2005; Waddington et al., 2005). If non-thermal effects influenced the RICE $\delta$D record, which is interpreted as temperature in our firn model, then the magnitude of accumulation change during this period may not be as large as reconstructed from the firn model. While not currently available, measurements of $d_{excess}$, dust particle size distributions, and dust geochemistry may be helpful in explaining the temperature and accumulation history of RICE.

## 5.2   Implications for climate and ice sheet history

Early reconstructions of the Ross Ice sheet during the LGM were based on glacial geological constraints from the western margin of the embayment. Denton and Hughes (2002) describe a maximum scenario in which thick ice in the Ross Embayment overrode both Siple Dome and Roosevelt Island. However, more recent observations and model experiments indicate a "fast and thin" scenario in which Siple Dome was not overrun by the interior ice sheet during the LGM (Waddington et al., 2005;

Price et al., 2007) and although the Ross ice streams likely slowed down during the LGM, they remained active, maintaining a low elevation profile of the ice sheet in the Ross Sea (Parizek and Alley, 2004).

Our results further support the fast and thin scenario, and add a key new constraint on ice thickness and thinning in the Eastern Ross Sea. Specifically, our results suggest that ice deposited on Roosevelt Island originated as accumulation local to the drilling site which may not be true if WAIS was thick during the LGM and overrode Roosevelt Island. If remnants of WAIS

were stranded on Roosevelt Island, this would likely result in a hiatus in the gas and ice chronology, in values of $\delta$D or TAC characteristic of more continental precipitation or much higher elevations, or discontinuities in the $\delta^{15}$N-$N_2$ record indicating a much different firn structure.

While the RICE ice core chronology does exhibit an abrupt shift in $\delta^{15}$N-$N_2$ and TAC during the ACR 3), no discontinuity in $\delta$D was observed meaning that it is unlikely that any of this ice originated as part of WAIS. At least one age discontinuity

(at 64.6 ka) as well as an age reversal was observed deeper in the core, but these depths do not coincide with the timing of the retreat of WAIS in the Ross Sea (McKay et al., 2016). The largest discontinuity, at 746.00-746.10 m, is accompanied a 20‰ change in $\delta$D. Methane and $\delta^{18}$O$_{atm}$ indicate that this discontinuity represents a 9.4 ka age gap. Over the same age range, the EDML ice core records a similar change in $\delta$D (Fig. 6f) indicating that this change in $\delta$D is explained by Antarctic climate patterns alone and without invoking large changes in ice sheet configuration. A possible second discontinuity is observed at

747.00 m depth at a small gap in measurements of 2.7 cm. Ice immediately below 747.00 m is interpreted as being warmer, meaning that this is probably not derived from somewhere upstream in WAIS or from a higher elevation (Fig. 6b). Additionally, no dramatic or sudden change in $\delta^{15}$N-$N_2$ or TAC was observed in association with either of these discontinuities (Fig. 6). We conclude that it is highly unlikely that the accumulation site of the RICE ice core changed during the deglaciation and that Roosevelt Island ice dome probably remained independent of an advanced WAIS during the LGM. A similar conclusion was

reached about Siple Dome during the LGM (Waddington et al., 2005; Nereson et al., 1998; Price et al., 2007; Parizek and Alley, 2004).

Geomorphological features on the Ross Sea bed do provide evidence of grounded ice north of Roosevelt Island during the LGM (Shipp et al., 1999; Anderson et al., 1984, 1992, 2014; Halberstadt et al., 2016). If these features were formed by an extended WAIS, it would imply that ice flowed around Roosevelt Island and Siple Dome and therefore was limited in thickness. These conditions would indicate that ice streams were active throughout the last glacial period. Alternatively, these geomorphic features may be the result of ice from a different origin. Price et al. (2007) proposed that during the LGM, an ice dome may have existed on Mary Byrd Land. In this scenario, thick, grounded ice could exist north of Roosevelt Island without flowing over or around the Roosevelt Island sea rise. The RICE records can not distinguish between these scenarios.

## 6 Conclusions

We present the RICE17 gas and ice chronologies for the RICE ice core. These timescales date the gas and ice records for the last ∼83 ka. Between 0-30 ka an automated synchronization routine is used to identify gas age control points that best match the RICE methane and $\delta^{18}O_{atm}$ records to the respective records from the WAIS Divide ice core (WAIS Divide Project Members, 2013; Buizert et al., 2015b). This technique requires few prior constraints, accommodates simultaneous synchronization of multiple parameters, and allows assessment of age uncertainty. Unique in this approach is the use of centennial-scale variability of methane for chronostratigraphic matching for ages older than the last ∼2,000 years. Synchronization between ice cores for the time period between 30 and 83 ka (719-753 m) was accomplished by manually choosing tie-points. Below 753 m the ice could not be dated with the currently available data. The RICE17 ice age-scale is based on annual layer counts between -62 and 2649 yr BP (0-343.7 m) and for depths below 343.7 m, a separate ice age-scale was derived from the synchronized gas age-scale by adding Δage estimated from a firn model.

A key contribution from the development of the RICE17 age-scale is evidence of active ice streams in the Eastern Ross Sea during the last glacial cycle. This is supported by the continuous age-scale and continuous records of climate from RICE which imply that the Roosevelt Island ice dome remained stable and independent of WAIS for at least the last 64.6 ka and likely for the last 83 ka.

The RICE ice core provides records of climate, with precise dating, for a scarcely sampled region of Antarctica. Future work will investigate regional climate of the Eastern Ross Sea in comparison to climate records from other sites to better understand spatial patterns around Antarctica, such as during the ACR when Roosevelt Island experienced an interval of particularly low accumulation, and to study the glacial retreat of WAIS in the Ross Sea.

*Code and data availability.* The following will be made available on public archives: RICE prior age control points, RICE final age control points, RICE17 age-scale interpolated at higher resolution, RICE $CH_4$, $\delta^{18}O$-$O_2$, and $\delta^{15}N$-$N_2$ data, and code for the gas synchronization routine.

## Appendix A: Methane Measurements

The methodology used at Oregon State University for measuring methane concentration in ice cores was described in Grachev et al. (2009) and Mitchell et al. (2013). Briefly, 40-60 g ice samples are trimmed and placed in glass flasks. The glass flasks are then attached to a high vacuum line and immersed in a chilled ethanol bath set to -63°C to keep the samples frozen. Since Mitchell et al. (2013), insulation has been added around the ethanol bath and above where the flasks are mounted. The added insulation reduced the temperature and water vapor content of gas in the headspace of the flasks and decreased their variability. Both can affect methane measurements by changing the pressure reading or the retention time of methane in the GC column. We also made efforts to more carefully regulate the amount of ethanol in the chilled bath and temperature of the hot water bath during melting. These steps improved stability of measurements and extraction between days.

After laboratory air has been evacuated, the flask valves are closed and the samples are melted in a hot water bath for 30 minutes to liberate air. Samples are refrozen by immersing the flasks in the cold ethanol bath. Once the samples are completely refrozen and the sample flask temperature has stabilized (approximately 1 hour after refreezing begins), methane measurement is performed by expanding sample air from the flask headspace into a gas chromatograph (GC).

Calibration measurements are made on an internal standard which is referenced to several compressed air standards externally calibrated on the NOAA scale (Dlugokencky et al., 2005). Calibration runs of systematically varying pressures are made both before and after samples are measured. The calibration curve is a least-squares linear regression between pressure and peak area for both sets of calibration runs described by a slope $m_{cal}$ and intercept $b_{cal}$.

Methane concentration ($C$) was previously calculated by comparing the sample pressure ($P_{meas}$) and the peak area ($PA_{meas}$) from GC analysis to the predicted peak area of the standard gas of equal pressure.

$$C_{raw} = C_{standard} \cdot \frac{PA_{meas} - b_{cal}}{P_{meas} \cdot m_{cal}} \tag{A1}$$

Four measurements can be made on each ice sample. If the two sets of calibration runs are taken individually, interpretation of sample concentrations typically differ by <3 ppb.

In this calculation, the sample pressure is used to quantify the number of moles of air in the sample assuming that the sample temperature remains constant and equal to standard air temperature during calibration runs. However, in a series of dry blank experiments it was determined that sample gas temperature is cooler than the standard gas temperature. We estimate the temperature difference to be 0.42% of the standard air temperature and thus the sample pressure needs to be corrected upwards by that amount. We call this the GC-thermal effect. We now apply the correction to the previous concentration calculation.

$$C_{raw} = C_{standard} \cdot \frac{PA_{meas} - b_{cal}}{P_{meas} \cdot 1.0042 \cdot m_{cal}} \tag{A2}$$

Results from typical wet blank samples, in which we add standard air over bubble free ice made from Milli-Q ultra-pure water, had shown that these samples are typically 2-3 ppb enriched compared to the standard concentration using the old method for calculating concentration. This agrees with the predicted magnitude of enrichment from the GC-thermal effect of 2.1 ppb.

Several wet blank samples are measured each day and are interspersed between ice core samples. The offset between the measured concentration of the wet blank and the known concentration of the standard gas added is called the "blank" correction.

This represents any effects during the sample analysis process which may alter the measured concentration. We bin the wet blank results over the time period for which samples were measured to establish a single blank correction value with an estimated blank correction uncertainty. RICE ice core samples were measured during two separate periods and have separate blank correction values.

Since the OSU analytical methods use a wet extraction technique to liberate ice core air, effects of gas solubility must be accounted for. Methane is more soluble than the major components of air and is preferentially dissolved during the extraction step. This leads to a decrease in the measured concentration compared to the true ice core concentration. Mitchell et al. (2013) empirically derived a solubility correction ($S$) which we repeated several times for the RICE samples. $S$ is defined as the total amount of methane (gas + dissolved) divided by the amount in gas phase. A solubility value of 1.0165 was used for ice core samples and 1.0079 for bubble free ice. We believe the difference in $S$ for the different ice samples results from the differences between how blank ice and ice containing air behave during melting. Specifically,

– Bubble-free ice melts slowly in comparison to glacial ice which sometimes melts rapidly and cracks violently. This, along with bubbles rising and breaching the melt water surface, cause disturbances in the water-air interface and promotes exchange of $CH_4$ into the melt water. This should lead to greater mixing and homogenization of air and water.

– Bubbles released into the melt water will be at higher pressures than the overlaying air because of surface tension. The higher partial pressure of $CH_4$ in those bubbles, in comparison to the standard gas added over the bubble-free ice, will cause air to go in to solution faster.

– Because glacial ice tends to be melted sooner than blank ice, a longer time period for liquid-gas exchange is available.

The "blank" and "solubility" corrections are applied in the following way:

$$C_{corrected} = C_{raw} \cdot S_{sample} - (C_{blank} \cdot S_{blank} - C_{standard}) \tag{A3}$$

This formula differs from that used by Mitchell et al. (2013). In Mitchell et al. (2013) no solubility correction was applied to the bubble free ice samples. The difference results in a constant offset of -7.4 ppb compared to Mitchell et al. (2013).

**Appendix B: Calibration of RICE CFA methane**

The RICE CFA methane record is qualitatively used for synchronization to the WAIS Divide methane record, and thus careful calibration was not required. Regardless, we present an ad hoc calibration of the RICE CFA methane record based on comparison to the RICE discrete methane record measured at Oregon State University (OSU). The RICE CFA methane record was measured in multiple campaigns and major adjustments to the analytical system occurred during both of those periods. Calibration of the RICE CFA record is done in a piecewise manner to reflect these changes.

Our calibration scheme accounts for instrument calibration, a concentration-based correction due to instrument sensitivity, and measurement drift, a time-dependent correction. For comparison between datasets, we subsample the CFA data at depths

where discrete measurements were made. We first apply the concentration calibration by regressing the sub-sampled CFA methane concentration against the discrete measurement. Drift was accounted for by a second regression comparing the residual of the concentration calibration against either the measurement time or sample depth, which ever provided the better correlation to the discrete dataset. Drift was only a significant factor between 500-680 m depth. Calibration values are given in Table A2.

Uncertainty in the relationship may be caused by measurement uncertainty, which is relatively small, or the uncertainty in the depth registration of the continuous measurements. Uncertainty in the depth registration is relatively unimportant in the top ∼670 m of core. In this section annual layers are relatively thick and methane variability is relatively low which results in only minor uncertainty in methane concentration from an error in depth. However, for the deeper section of core, methane concentrations vary rapidly with depth and small errors in the depth registry represents large differences in methane concentration. Because of errors in the depth registry and the sensitivity of the inferred methane concentration, we restrict our calibration scheme to the last ∼39.66 ka (725.63 m) ending after the inclusion of GIS-8.

## Appendix C: Steady-State Herron-Langway Sensitivity Test

The gas age-ice age offset (Δage) for the RICE ice core was estimated with a dynamic Herron-Langway model (Buizert et al., 2015b). The model is constrained by measurements of $\delta^{15}$N-N$_2$ (a proxy for firn thickness) and $\delta$D (a proxy for past temperature). The dynamic model also assumes a convective zone of 2 m, surface firn density of 400 kg·m$^{-3}$, and geothermal heat flux of 78 mW·m$^{-2}$.

A steady-state version of the Herron-Langway model mimics the dynamic version following a similar iterative methodology. A constant Δage estimate is assumed as a prior to assume an ice age scale for the temperature history. The model then calculates a new Δage solution from the temperature and firn thickness histories. This process is repeated until iterations no longer change appreciably.

While the dynamic version is used to establish the Δage history for RICE, the steady state version has the advantage of being computationally faster and is used in a sensitivity test. In this test, we vary prior assumptions about the isotopic temperature sensitivity used to infer past temperature, the convective zone thickness, surface firn density, and assumptions about geothermal induced temperature gradient in the firn. In a Monte Carlo approach, each parameter is given a range of acceptable values from which the steady-state Herron-Langway model calculates the Δage history (Table A1). This is repeated for 6000 realizations. Realizations are rejected if:

– The modeled modern ice age at lock-in depth (LID) is more than 25 years different than the annual layer counted age of that depth (48.57 m, 89 yrs BP).

– The modeled modern accumulation is less than 0.15 or greater than 0.35 m ice per year.

– The isotopic sensitivity is less than 3.2 or greater than 9.6 ‰·K$^{-1}$, similar to the range of values observed for West Antarctica (Masson-Delmotte et al., 2008; Cuffey et al., 2016; Stenni et al., 2017). This parameter is randomly chosen from a normal distribution at the beginning of each realization and can fall outside of this range.

- The minimum temperature is less than -60°C.

- The maximum estimated $\Delta$age is less than 1000 years.

- Modeled accumulation does not exceed 1.0 m ice per year and is never negative.

$\Delta$age is calculated for each depth that $\delta^{15}$N-N$_2$ was measured. Uncertainty in $\Delta$age is assumed to be the root-mean-square-error of accepted realizations and is estimated for each sample depth.

*Author contributions.* JEL measured CH$_4$ at OSU on RICE samples. JEL, NANB, T. Baisden, T. Blunier, VGC, EDK, and PV participated in the 2013 or 2014 continuous flow analysis campaigns. JPS measured $\delta^{15}$N-N$_2$ and $\delta^{18}$O-O$_2$ at SIO. JEL, CB, TJF, MW, EB, DDJ, FP, JPS, and EDW contributed to methods and discussion leading to the development of the RICE17 age scale. JEL, HC, TJF, RH, JPS, and
EDW contributed with discussion of glaciological interpretations of the ice core record. All authors provided valuable feedback and made helpful contributions to writing the manuscript.

*Competing interests.* The authors declare no conflicts of interest.

*Acknowledgements.* This work is a contribution to the Roosevelt Island Climate Evolution (RICE) Program, with contributions from USA, UK, Sweden, New Zealand, Italy, Germany, Australia, China Denmark, Germany, Italy, China, Denmark, and Australia. The US contribu-
tion was funded by grants from the US National Science Foundation Office of Polar Programs (0944021, 0837883, 0944307, 1042883 & 1643394). The NZ contribution was funded through New Zealand Ministry of Business, Innovation, and Employment grants issued through Victoria University (RDF-VUW-1103, 15-VUW-131), GNS Science (540GCT32, 540GCT12) and Antarctica New Zealand (K049). The Danish contribution was funded by the Carlsberg Foundation's North-South Climate Connections project grant. The research also received funding from the European Research Council under the European Community's Seventh Framework Programme (FP7/2007-2013) ERC
grant agreement 610055 as part of the Ice2Ice project.

     We thank Micheal Rebarchik and Will Patterson for laboratory assistance at Oregon State University. We thank Ross Beaudette for laboratory assistance at Scripps Institution of Oceanography. We thank Marius Simonsen, Helle Kjær, Rebecca Pyne, Daniel Emanuelsson, Bernadette Proemse, Ross Edwards, Darcy Mandeno, and the rest of the RICE team for participation in preparing ice samples and operating instruments during the continuous melting campaigns at GNS. We thank the US Antarctic Support Contractor, Antarctica NZ, the US Air Force, the Air National Guard, and Kenn Borek Air for logistical and field support.

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

**Figures**

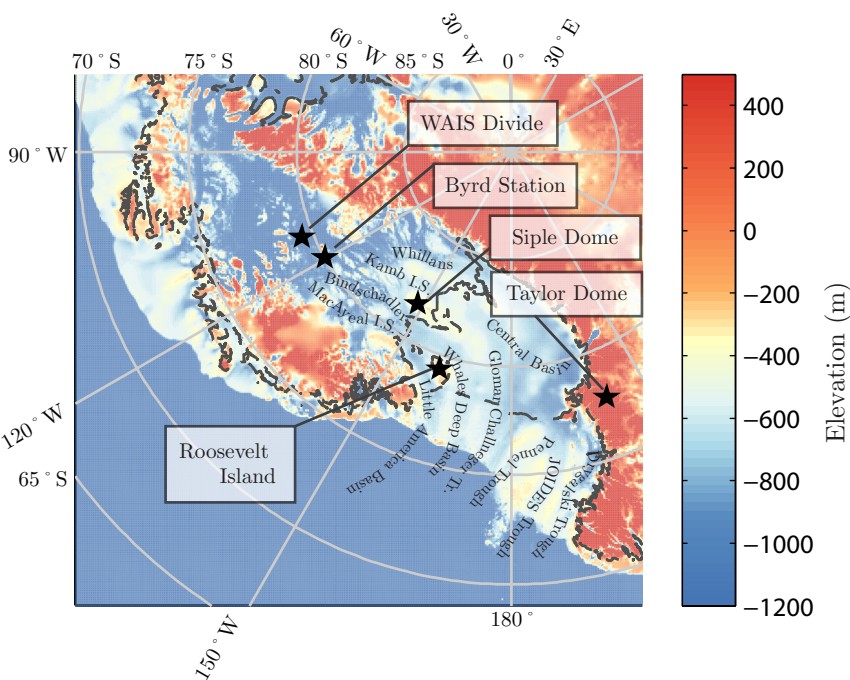

**Figure 1.** Map of bedrock elevation in the Ross Sea Sector of Antarctica (referenced to WGS84 datum) (Fretwell et al., 2013). Gray dashed lines indicate ice sheet grounding lines and ice margins. Locations of the RICE (Roosevelt Island), Siple Dome, Byrd Station, WAIS Divide, and Taylor Dome ice cores are marked with black stars.

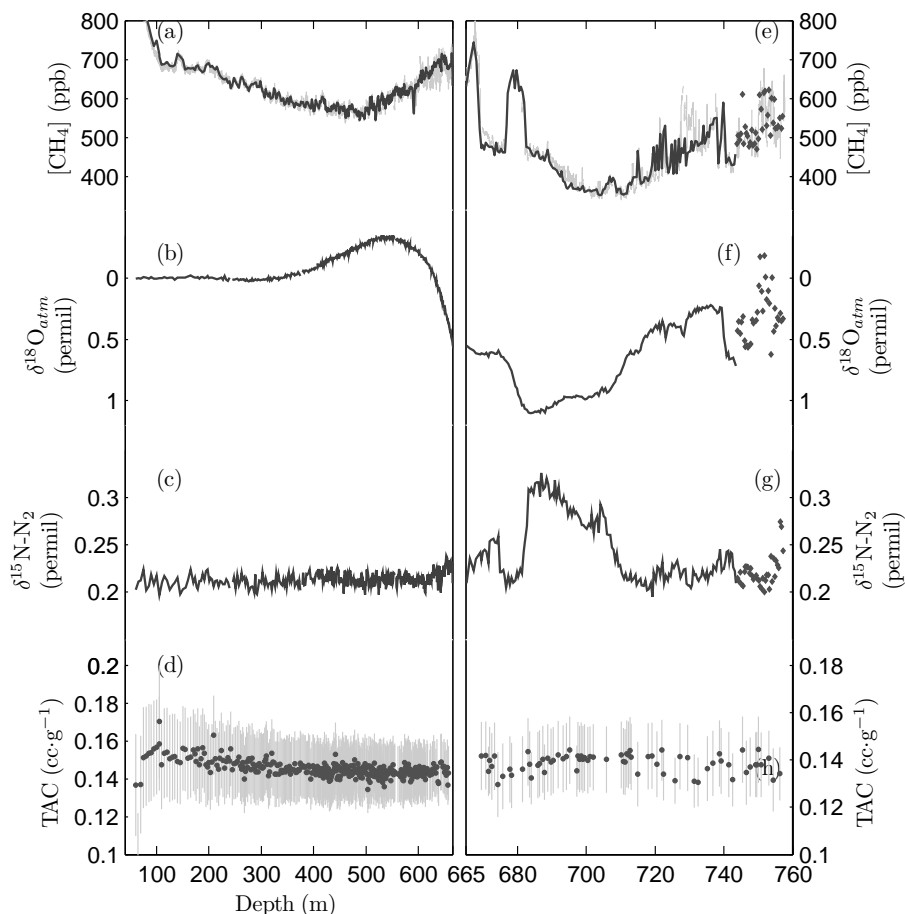

**Figure 2.** Gas data from the RICE ice core. Left panel (a-d) (60-665 m) covers the last 11.26 ka; Right panel (e-h) (665-760 m) covers measurements from 11.26-83 ka and measurements from the 9 m of ice below the dated section. (a, e) Continuous methane measurements, gray, between 0-726 m depth are calibrated to discrete methane measurements, black. Beyond 726 m depth, raw CFA methane measurements are plotted. (b, f) $\delta^{18}O_{atm}$ measurements are corrected for gravitational enrichment in the firn layer using $\delta^{15}$N-N$_2$ (c, g) . (d, h) Total air content (TAC) was measured in conjunction with discrete methane.

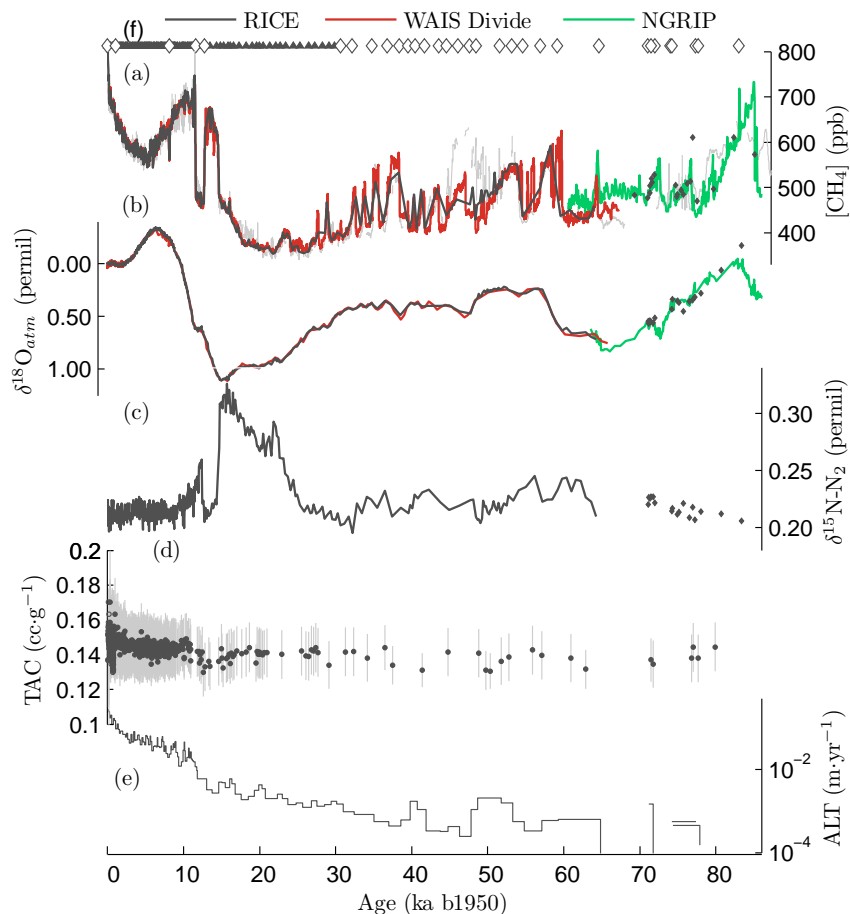

**Figure 3.** Gas data from the RICE ice core plotted on the RICE17 age scale. (a) RICE methane and (b) $\delta^{18}O_{atm}$ records (gray) shown in comparison to target records from the WAIS Divide ice core on the WD2014 age scale (Buizert et al., 2015b) (red) and NGRIP ice core (Baumgartner et al., 2014; Landais et al., 2007) on a modified GICC05modelext chronology (Wolff et al., 2010) (green). Solid triangles above panel (a) are gas age constraints from a Monte Carlo analysis, open diamonds are prior ACPs from visual matching (see text). (c) $\delta^{15}N$-$N_2$ is used in a firn densification model to estimate the ice age-gas age offset. (d) RICE TAC measurements. (e) Mean annual layer thickness calculated from gas age control points adjusted for $\Delta$age.

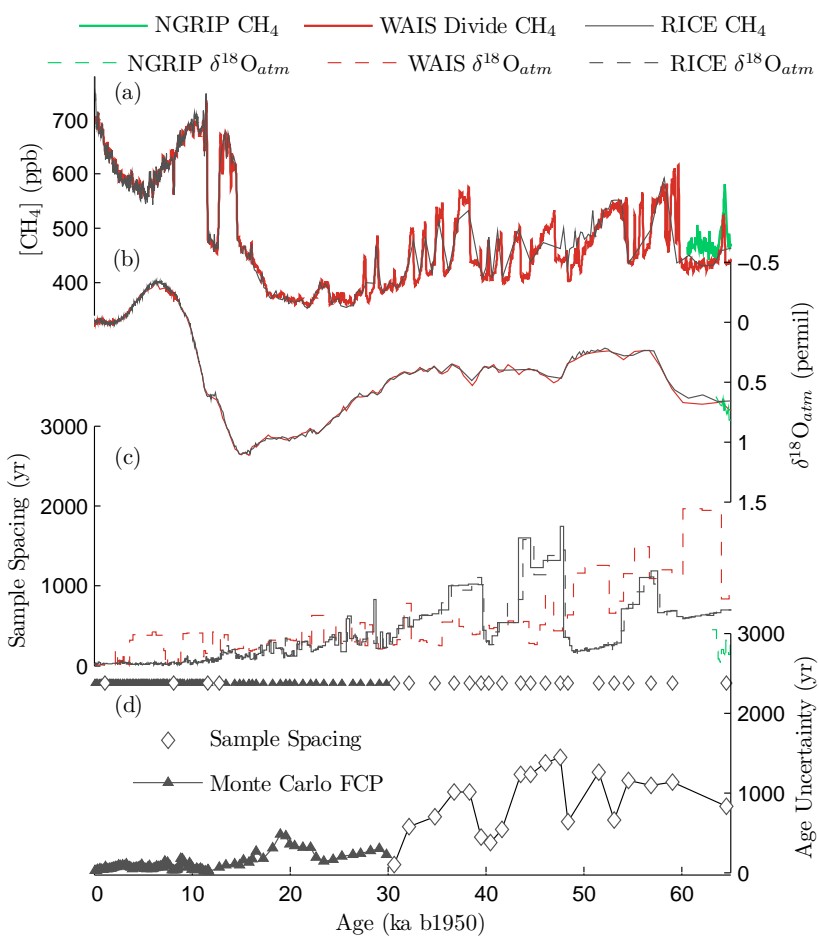

**Figure 4.** RICE ice core age uncertainty. (a) Methane and (b) $\delta^{18}O_{atm}$ records from the RICE (gray), WAIS Divide (Rhodes et al., 2015) (red), and NGRIP ice cores (Baumgartner et al., 2014) (green). (c) Sample spacing for methane (solid lines) and $\delta^{18}O_{atm}$ (dashed lines) for the various cores. (d) Gas age uncertainty, relative to WD2014, for ages determined from a Monte Carlo analysis (solid triangles) and for extended gas age control points (open diamonds).

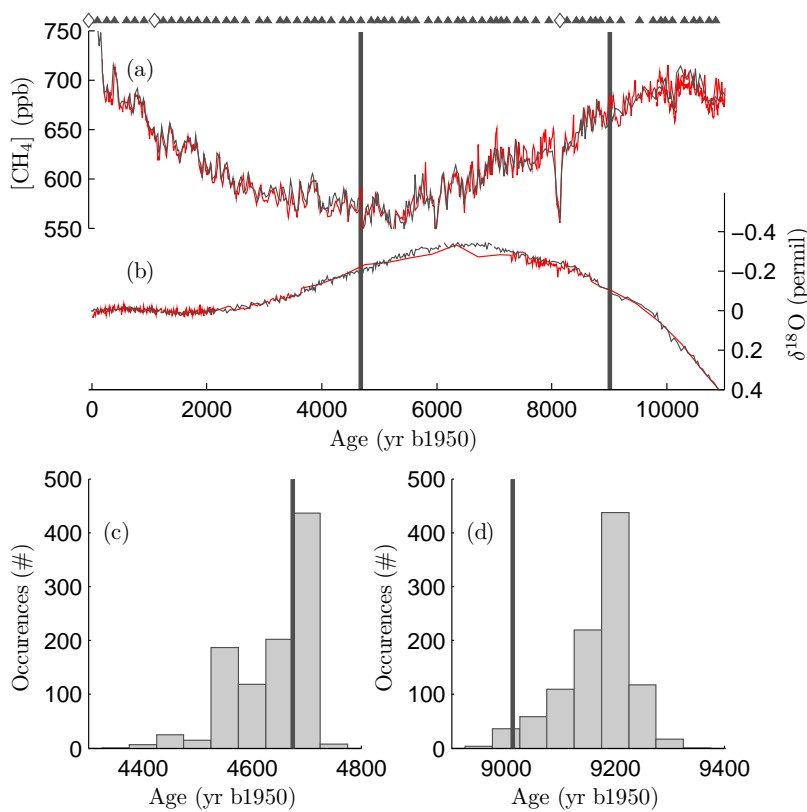

**Figure 5.** (a) RICE methane and (b) $\delta^{18}O_{atm}$ records matched to those from the WAIS Divide ice core (Rhodes et al., 2015; Buizert et al., 2015b) (red lines) for the last 11.7 ka. FCPs from Monte Carlo routine are shown as solid triangles, prior constraints are shown as open diamonds. Lower panels (c, d) show the distribution of the gas ages for two particular depths, 459.05 m (4675 yr BP) (c) and 621.28 m (9012 yr BP) (d), resulting from the Monte Carlo analysis. The final age of these depths, resulting from the best realization, are shown as vertical gray lines.

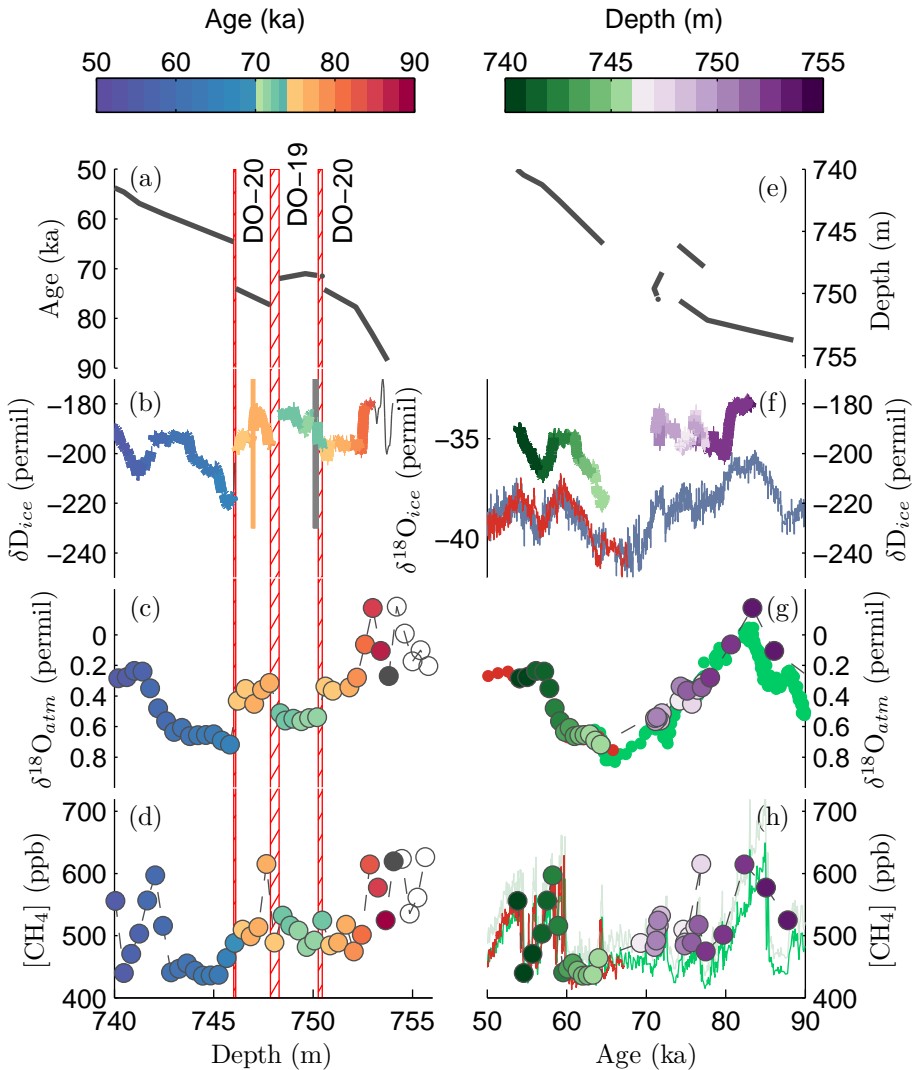

**Figure 6.** (a) Depth-age relationship from 740-756 m and evidence of an age reversal within the RICE ice core from (b) $\delta$D, (c) $\delta^{18}O_{atm}$, and (d) methane data. (b-d) Measurements are plotted against depth and color coded according to the age of the sample. Open circles represent samples for which an age could not be determined. (a-d) Red hatched bars represent discontinuities, where periods of climate history appear to be missing. Solid gray bars in (b) are measurement gaps in the $\delta$D record associated with large changes in $\delta$D. (e-h) The depth-age relationship and measurements are plotted against the age of the sample and color coded according to depth. WAIS Divide data (red), NGRIP (green), and the EDML $\delta^{18}O_{ice}$ record (blue).

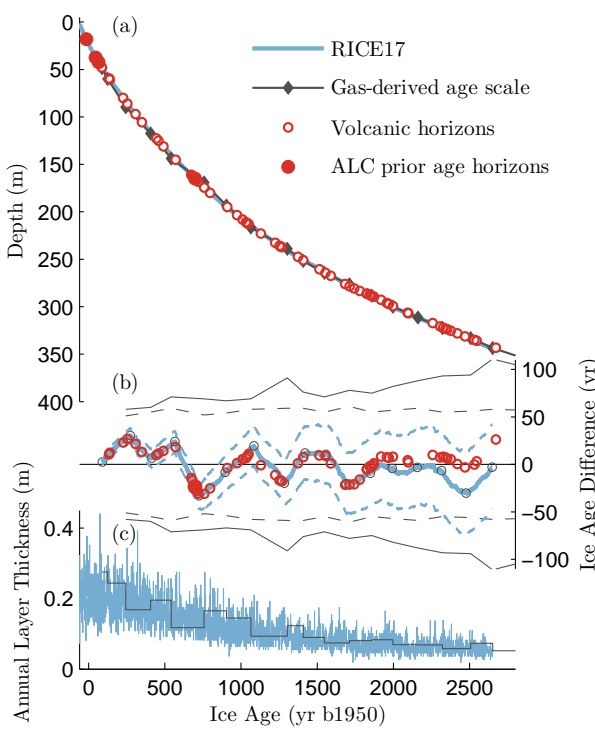

**Figure 7.** Comparison of the gas-derived (gray) and the annual layer counted ice age scale (blue, Winstrup et al. 2017). Six absolute age markers (closed red circles) were identified and an additional 67 volcanic horizons were cross correlated to volcanic horizons in the WAIS Divide core (plotted on WD2014 ice age scale, open red circles, Winstrup et al. 2017). (b) Difference in ice age between the gas-derived ice age scale and annual layer counts; positive values indicate that at the same depth the annual layer counts is older than in the gas-derived age scale. Uncertainty estimates of the gas-derived ice age scale (solid gray lines), of the Δage estimate only (dashed gray lines), and of the annual layer counted age scale (blue dashed lines). (c) Interpretations of annual layer thickness from the gas-derived ice age scale and annual layer interpretations.

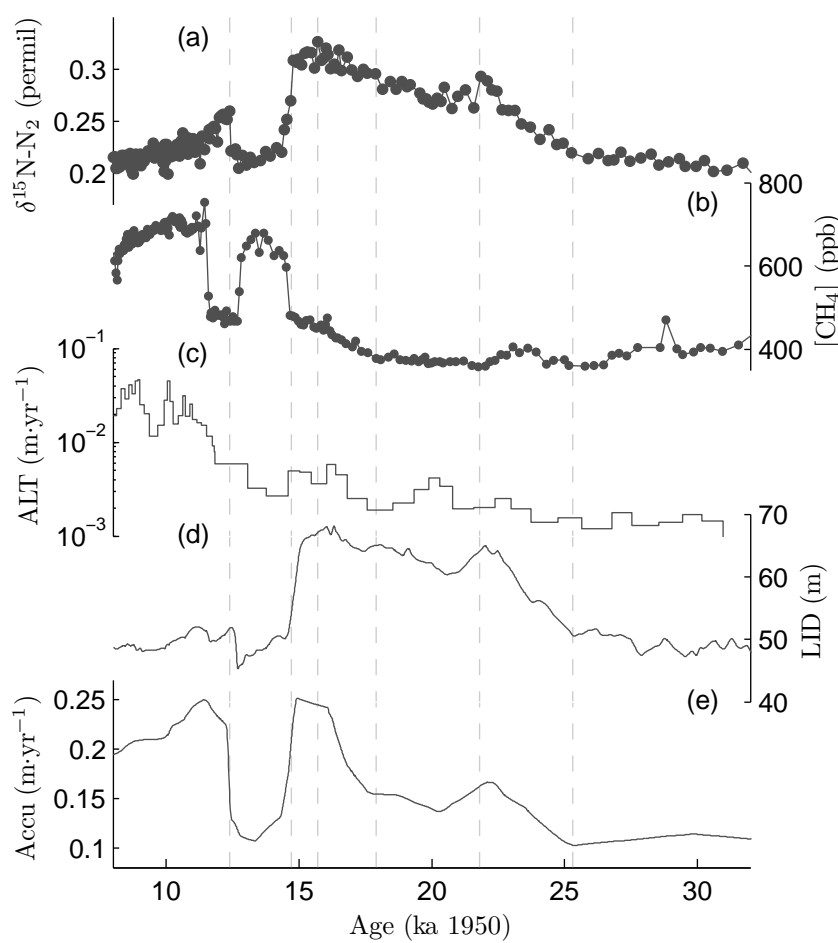

**Figure 8.** Comparison of (a) $\delta^{15}$N-N$_2$, (b) CH$_4$, (c) annual layer thickness implied from depth-age scale, (d) lock-in depth (LID), and (e) accumulation reconstructions from the RICE ice core. Lock-in depth and accumulation is calculated with a dynamic Herron-Langway firn densification model.

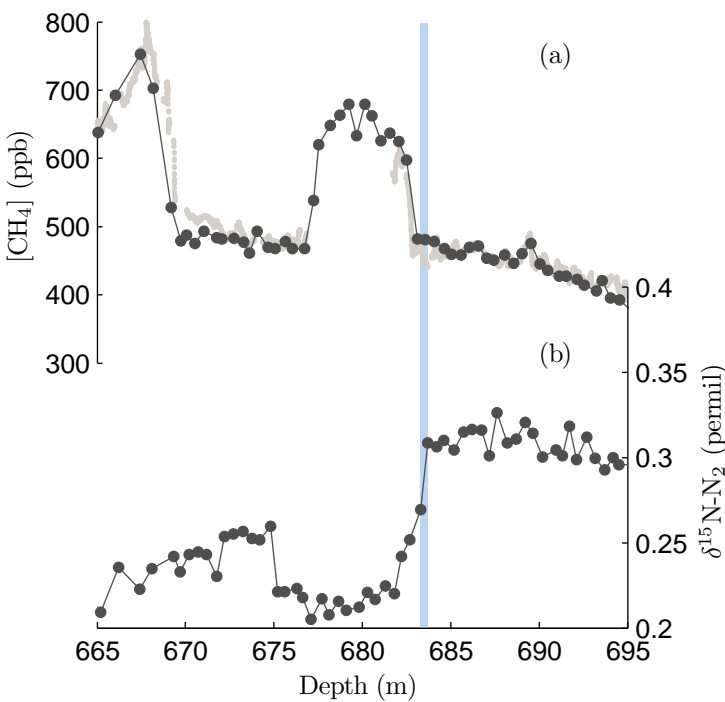

**Figure 9.** Comparison of (a) methane and (b) $\delta^{15}$N-N$_2$ from the RICE ice core plotted on depth. Vertical blue bar highlights the onset of a large decrease in $\delta^{15}$N-N$_2$.

**Tables**

**Table 1.** Prior gas age constraints and uncertainty are based on matching of features in the atmospheric history of methane and $\delta^{18}O_{atm}$. Name or description of feature are given in the notes and the primary parameter used to identify the feature is provided as the identifying variable. Age uncertainty is related to the duration of the feature.

| Depth (m) | Gas_Age (yr) | $\sigma$ (yr) | Identifying Variable | Notes |
|---|---|---|---|---|
| 48.57 | -55.4 | 7 | Modern LID | |
| 239.000 | 1092 | 45 | $CH_4$ | |
| 591.000 | 8140 | 30 | $CH_4$ | 8.2 ka event |
| 669.150 | 11580 | 30 | $CH_4$ | Younger-Dryas — Preboreal |
| 677.300 | 12780 | 57 | $CH_4$ | Bølling-Allerød — Younger Dryas |
| 719.300 | 30660 | 100 | $CH_4$ | GI 5.1 Termination |
| 720.700 | 32150 | | $CH_4$ | GI 5.2 Termination |
| 722.800 | 34780 | | $CH_4$ | GI 7 Termination |
| 723.900 | 36750 | | $CH_4$ | GI 8 Termination |
| 724.600 | 38370 | | $CH_4$ | GI 8 Onset |
| 725.310 | 39530 | | $CH_4$ | Mid-GS 9 Methane Event |
| 726.850 | 40332 | | $CH_4$ | GI 9 Onset |
| 728.090 | 41643 | | $CH_4$ | GI 10 Onset |
| 728.720 | 43544 | | $CH_4$ | GI 11 Onset |
| 729.050 | 44562 | | $CH_4$ | GI 12 Termination |
| 729.680 | 46100 | | $\delta^{18}O_{atm}$ | |
| 730.050 | 47620 | | $\delta^{18}O_{atm}$ | GI 12 Onset |
| 730.950 | 48420 | | $\delta^{18}O_{atm}$ | Mid-GS 12 Methane Event |
| 737.260 | 51570 | | $\delta^{18}O_{atm}$ | |
| 739.650 | 53115 | | $\delta^{18}O_{atm}$ | |
| 740.470 | 54595 | | $\delta^{18}O_{atm}$ | GI 14 Onset |
| 741.250 | 56885 | | $\delta^{18}O_{atm}$ | (MIS 4/3 Transition) |
| 742.550 | 59100 | | $\delta^{18}O_{atm}$ | GI 17.1a Onset (MIS 4/3 Transition) |
| 746.000 | 64600 | | $\delta D, \delta^{18}O_{atm}, CH_4$ | Top Depth of discontinuity |
| 746.100 | 74000 | | $\delta D, \delta^{18}O_{atm}, CH_4$ | Bottom Depth of discontinuity |
| 747.850 | 77300 | | $\delta D, \delta^{18}O_{atm}, CH_4$ | Depth of reversal in $\delta D$ |
| 748.290 | 72000 | | $\delta^{18}O_{atm}$ | First $\delta^{18}O_{atm}$ sample clearly in DO-19 grouping |
| 749.600 | 71000 | | $\delta^{18}O_{atm}, CH_4$ | Depth of youngest part of reversal in $\delta D$ |
| 750.460 | 71500 | | $CH_4$ | Deepest sample clearly related to reversal grouping |
| 750.560 | 74200 | | $\delta^{18}O_{atm}, CH_4$ | Shallowest sample related to return to DO20 values |
| 752.150 | 77700 | | $\delta^{18}O_{atm}, CH_4$ | Last $\delta^{18}O_{atm}$ sample clearly part of DO20 |
| 752.950 | 83000 | | $\delta^{18}O_{atm}$ | Minima in $\delta^{18}O_{atm}$, matching values observed in NGRIP and EDML for MIS 5a |
| 753.750 | 88500 | | $\delta^{18}O_{atm}$ | Enriched $\delta^{18}O_{atm}$, MIS 5b? |

**Table 2.** Model Parameters used for optimized correlation routine. Code for routine can be found in supplementary material.

| Variable | Description | 0-670 m | 670-718.13 m |
|---|---|---|---|
| **Model Parameters:** | | | |
| Runs | # of realizations in Monte Carlo Analysis | 1000 | 1000 |
| N | # of subsections | 76 | 25 |
| $\Delta t$ | Duration of subsections | 154 yrs | 726 yrs |
| k | # of refinements to perturbation | 13 | 13 |
| $n_{rep}$ | # of repetitions before moving to next refinement | 20 | 20 |
| **Perturbation Conditions:** | | | |
| $\overline{A}_{max}$ | Maximum "Accumulation" | 75 cm·yr$^{-1}$ | 75 cm·yr$^{-1}$ |
| $\overline{A}_{min}$ | Minimum "Accumulation" | 0 cm·yr$^{-1}$ | 0 cm·yr$^{-1}$ |
| $\Delta t/\Delta t_{prior}$ | Relative change in duration of subsection from prior | 10x | 10x |
| $\overline{A}_i/\overline{A}_{i-1}$ | Maximum relative change of "accumulation rate" between subsequent subsections | 2x | 2x |

**Appendix Tables**

**Table A1.** Model Parameters used for steady-state Herron-Langway model.

| Description | Mean | $\sigma$ | Distribution |
|---|---|---|---|
| $\delta^{15}$N-N$_2$ | | 0.0027‰ | normal |
| Modern temperature | -23.5°C | 3°C | normal |
| Isotope sensitivity | 6 ‰·K$^{-1}$ | 1.2 ‰·K$^{-1}$ | normal |
| Surface firn density | 400 kg·m$^{-3}$ | .05 kg·m$^{-3}$ | normal |
| Convective zone thickness | 2 m | 2 m | uniform |
| Geothermal induced temperature gradient | -0.6 K | 0.6 K | uniform |

**Table A2.** Calibration of the RICE CFA methane dataset is performed by comparison to the RICE discrete methane dataset with corrections for instrument sensitivity and instrument drift. Calibration is done for different segments of the core. Quality of the fit is described by the $R^2$ statistic comparing calibrated values of the sub-sampled CFA methane record to discrete measurements.

| Depth Range | Instr. | Drift | $R^2$ |
|---|---|---|---|
| 0-500 m | $C_{final} = C_{raw} \cdot 1.1079 + 25.0900$ | | 0.9854 |
| 500-680 m | $C_{final} = C_{raw} \cdot 0.9440 + 10.9337$ $+$ $t_{meas} \cdot 4.0181 \cdot 10^{-5} - 1.4013 \cdot 10^{-5}$ | | 0.9239 |
| 682-726 m | $C_{final} = C_{raw} \cdot 0.8869 + 29.2036$ | | |