# Peer review of "An 83,000 year old ice core from Roosevelt Island, Ross Sea, Antarctica"

_Climate of the Past, 2018_

## Referee Comment (RC1) · Anonymous Referee #1 · 4 Sep 2018

Review of 'An 83,000 year old ice core from Roosevelt Island, Ross Sea, Antarctica by Lee et al.

The paper by Lee et al. presents an age scale for the Roosevelt Island Climate Evolution project (RICE) ice core. Roosevelt Island is located in the Ross Sea, a primary outlet region for the West Antarctic Ice Sheet (WAIS).

Most of the paper is devoted to deriving the RICE age scale (RICE17), including novel use of centennial-scale variations in methane to synchronise the record with other ice cores. Following presentation of the age scale, the paper draws some conclusions on the glaciological history of the Roosevelt Island Ice Dome. Specifically, the authors argue that the WAIS did not override Roosevelt Island at any time during the past 65kyr (i.e. the island remained independent of an advanced WAIS during the glacial). The authors also comment on Holocene variability in atmospheric methane. Specifically, that centennial-scale variations in methane are present throughout the Holocene, casting doubt on previous work suggesting that similar centennial-scale variations in methane during that late Holocene have an anthropogenic origin.

I think the article should eventually be suitable for publication in Climate of the Past. In general the age scale work is good but the broader conclusions appear less mature. However some substantial work is needed to address the following points.

Major comments.
1)      There is no doubt that the RICE ice core contains important information on climate and glaciological history in the Ross Sea region. The RICE age scale is essential to decoding this information and in my assessment the authors have done a good job with the age scale and this part clearly merits publication following some revision and restructuring. On the critical side, the sections on glaciological history of the Roosevelt Island and on methane variability are in my assessment much less mature than the age scale work and need to be substantially strengthened or removed all together. The methane findings are described as preliminary in the abstract and Climate of the Past should in my opinion not be publishing work that the authors believe is preliminary.
2)      A further criticism is that the paper is excessively long, contains a lot of repetition and is not well structured – this makes it very tedious going for the reader. At present the paper reads more like a thesis than a journal article. Serious effort needs to be made by the author team to cut out information that is redundant to the results and conclusions presented.
3)      P6L1—4: An example figure is needed showing the straticounter annual layer selection.
4)      Section 5.1: There is no doubt that the centennial-scale methane variability is an interesting and important observation. However, in my view it should be the subject of a stand-alone paper, in which one would

like to see detailed comparison of the various records and labelling of the methane trends that have been attributed to anthropogenic activity. As it stands the two paragraphs do not give a thorough treatment but still take up substantial space. If it must be included then I would suggest to scale back the section, certainly not so much introductory material is needed (it's not until near the final lines of the section that the RICE results are even referred to).

5) Section 5.2: The first paragraphs appear to describe a thickening of the firn column going in to the LGM (25.3 to 21.8 ka) and an increase in accumulation rate. I find it surprising that accumulation rate would increase through the LGM and this observation merits some discussion. I note that the accumulation rate declines during the ACR as one would expect under cooler conditions.

6) The reconstruction and discussion of RICE accumulation history depends strongly on the questionable assumption that dD is a faithful recorder of temperature across the deglaciation. The potential for non-thermal effects on the dD record is critical and should be made earlier on in the paper (currently it is not until P14L25—30).

Technical comments.

1) Abstract line 4. Clim. Past should not be in the business of publishing 'preliminary observations'. See major comments on whether these should be presented at all.

2) Intro first para: The main focus of the paper is timescale development and the introduction should direct the reader to that subject from the start. Marine ice sheet stability does not come up again in the paper so does not need to be described here. Remove the para and I'd suggest replace with some sentences on importance of timescale development.

3) Intro second para: Here two scenarios are put forward for glaciological history in the Ross Sea region. The later discussion should more clearly refer back to the scenario which is supported by the new results. Since this glaciological history is not the primary focus of the paper I would suggest to move the paragraph to the end of the introduction.

4) P2L35: No need to pers. comm. a co-author.

5) P3L14: This is the sort of information that is most relevant to the main age scale development task at hand and which belongs in the intro.

6) Section 2. Para 2 of the intro could be better fit into this section renamed something like 'Roosevelt Island ice core and glaciological history'.

7) P4L20: I don't see any points in the RICE methane curve (Fig 3a) sitting 30ppb above the WAIS curve.  The legend does not inform which methane measurements are discrete and which are the problematic CFA.

8) P6L20: I don't think pers. comm. of a co-author is needed, remove here and throughout.

9)  P6L5-15: The method used for each section of the core is repeated in the abstract, in line P3L5-20 and later again in the results. That's far too

much repetition and testing of the readers patience. Its essential to revise the structure to avoid this repetition.

10) P6L18: Also repeats earlier material in Intro.

11) P6L30: '35% to 75% of the.. relevant variable": please clarify what is meant here.

12) Fig 5d): Please explain to the reader why there is a large difference between the 'best realization', judged in terms of the goodness of fit, and the number of occurrences of a particular fit.

13) Section 4.1: it would help the reader if this section referred right at the start to Fig 5.

14) P9L3: I think its now the 4th time I read this.

15) P9l15—19: As someone who works with these records I find this very hard to follow. Please revise for clarity.

16) Section 4.3: Shorten it.

17) P11L7: The delta-age is established using a firn densification model, in which the modelling relies on a RICE temperature history derived from dD. The temperature history is thus integral to the development of the age scale of the ice, however the dD-based temperature reconstruction is cited as a pers. comm. I think the authors need to refer to a published temperature history or include the temperature history here... returning from coffee break... ok reading further down I see there are some more details on the assumptions in the temperature reconstruction and comparison to borehole data. Remove the pers. comm and see major comments.

18) P12L24: Good. Agreed.

19) Section 5.1: There is no doubt that this discussion of methane variability is interesting. In my view it should be the subject of a stand-alone paper, in which one would like to see detailed comparison of the various records and labelling of the methane trends that have been attributed to anthropogenic activity. As it stands the two paragraphs do not give a thorough treatment but still take up substantial space. If it must be included then I would suggest to scale back the section, certainly not so much introductory material is needed (it's not until near the final lines of the section that the RICE results are even referred to).

20) Fig 4d. Adjust the y limits so we can more easily see the age uncertainty.

21) P13L33: Include the uncertainty in the onset of the d15N change at 14.71 ka; I'm far from convinced that it significantly precedes onset of Bølling at 14.64 +- .19 ka.

22) P14L11: This interesting sentence suffers from being way too long.

23) P14L19--40: It would be more logical and much easier for the reader to follow your arguments if you set out the preferred explanation first and then explain, briefly, why some potential alternative explanations are unlikely. I don't find the preferred explanation very convincing: I don't see any quantitative data to support it, only some arm waving analogy to recent periods.

24) Section 5.3: It would help the reader to refer early in the section to the 'maximum' and 'fast and thin' Denton (1989) scenarios that were set up in the introduction.

25) P15L4: Again refer to the scenario set up in the introduction, here and elsewhere in this section.

26) P15L18: Refer to the dD record in Fig3b. Not to a pers comm!

27) P15L26: The comment about an MBL ice dome comes out of the blue and its far from obvious who it provides an alternative explanation for the continuity of the record. Clarify or drop.

28) Conclusions para 1: The fifth time we read this?

29) Many references found in the introduction do not come up again in the discussion. I'd suggest a bit more focus and continuity between the most relevant literature flowing from the intro to the discussion.

30) As a final point, it is tedious as a reviewer to have to spend so much time commenting on structure, something the author team could have worked on internally prior to submission. The age scale is important and should be presented as accessibly as possible.

---

## Referee Comment (RC2) · Anonymous Referee #2 · 12 Sep 2018

Review of the manuscript "An 83,000 year old ice core from Roosevelt Island, Ross Sea, Antarctica" by Lee et al.

General comments: This manuscript presents a suite of new gas records from an ice core drilled at Roosevelt Island, an ice rise in the Ross Sea. The primary objective is to establish its chronology by annual layer counting for relatively shallow depths and matching of gas records with existing WAIS Divide and Greenland ice core chronologies. The continuous part of the ice core extends to 65 kyr BP, suggesting that the Roosevelt Island has existed since at least this age. CH4 records show centennial-scale variability throughout the Holocene, with implications on natural vs. anthropogenic CH4 emission in pre-industrial periods. These discussions have some important implications for past climate and ice sheet variations. The dating method developed here is

a nice contribution to the ice core community.

However, the lack of water isotope records and interpreted temperature records in this manuscript makes it difficult to review the estimated annual layer thickness using a firn densification model and its effects on dating and paleoclimatic implications. I find this study is potentially an important contribution to paleoclimatic communities but do not recommend publication in its current form. The authors would need to decide if they remove some parts of the manuscript regarding annual layer thickness estimates from firn modeling (but it will make the manuscript much less attractive), or they add water isotope data and temperature estimate (I would recommend the latter for publication in CP). The discussion of anthropogenic and natural CH4 variability needs some quantitative analyses (for example comparing frequency and variability after detrending for different time periods). To my eyes, the CH4 records appear to have different centennial-scale variations in earlier and later parts of the Holocene.

Specific comments: P5, L5. Regardless of the careful trimming of the ice in the same shape, the cut-bubble effect should change (generally decreasing) with depth due to the change in bubble sizes. The cut-bubble effect thus needs to be corrected.

P16, L28. I do not understand why the temperature stability of the sample leads to the improvement in S/N of the gas chromatograph.

P17, L30. Please explain why the solubility correction factors are so different for sample and bubble-free ice?

Fig. 2c and i. The scales of the axes should be the same for the left and right panels.

Fig. 5d. Why is the vertical line drawn at about 9000 yr BP and not near 9200 yr BP (highest occurrences)?

Supplementary file "RICE17_Interpolated_Ages_20180530.txt" appears to contain two units for the ice age (probably C.E. and yrBP are switched at 343.5 m).

---

## Author Comment (AC1) · 24 Jan 2019

**Responses to Reviewer 1:**
First, Thank you for your comments. We agree with many of the points brought up in the comments and will work to incorporate suggestions of both the technical and major comments. Many of the suggestions were in regard to improving organization and conciseness of the manuscript. This requires rearranging sections and a significant number of edits to sentence structure for clarity. We will undertake these changes, but for brevity, we do not include every planned sentence level edit in our response to reviewer comments at this time.

[Figure]

**Major Comments:**

1. *There is no doubt that the RICE ice core contains important information on climate and glaciological history in the Ross Sea region. The RICE age scale is essential to decoding this information and in my assessment the authors have done a good job with the age scale and this part clearly merits publication following some revision and restructuring. On the critical side, the sections on glaciological history of the Roosevelt Island and on methane variability are in my assessment much less mature than the age scale work and need to be substantially strengthened or removed all together. The methane findings are described as preliminary in the abstract and Climate of the Past should in my opinion not be publishing work that the authors believe is preliminary.*

   We have reorganized the manuscript to focus on the chronology development and what the chronology may tell us about the glaciology of Roosevelt Island. The introduction has been re-written, we have reduced how many times we describe development of the chronology, and we have removed section 5.1 "New observations of centennial-scale variability in the Holocene methane cycle". More specifics are given in response to other comments below.

2. *A further criticism is that the paper is excessively long, contains a lot of repetition and is not well structured – this makes it very tedious going for the reader. At present the paper reads more like a thesis than a journal article. Serious effort needs to be made by the author team to cut out information that is redundant to the results and conclusions presented.*

   This comment is addressed more specifically in our response to the technical comments, below.

3. *P6L1—4: An example figure is needed showing the straticounter annual layer selection.*

We would prefer not to take up additional space in the paper with this. The depth-age relationship based on annual layer counts is already compared to the gas-based age scale in figure 7. For more information about annual layer selection, we would rather refer to Winstrup et al. (2018) (https://www.clim-past-discuss.net/cp-2017-101/) which describes in detail the data sets used for annual layer selection, the straticounter routine used to interpret count annual layers, and description of the seasonal patterns observed in those data sets. Winstrup et al. (2018) was recently revised and resubmitted so hopefully it will be fully published shortly.

4. *Section 5.1: There is no doubt that the centennial-scale methane variability is an interesting and important observation. However, in my view it should be the subject of a stand-alone paper, in which one would like to see detailed comparison of the various records and labelling of the methane trends that have been attributed to anthropogenic activity. As it stands the two paragraphs do not give a thorough treatment but still take up substantial space. If it must be included then I would suggest to scale back the section, certainly not so much introductory material is needed (it's not until near the final lines of the section that the RICE results are even referred to).*

The relevant material on centennial-scale methane variability has been removed.

5. *Section 5.2: The first paragraphs appear to describe a thickening of the firn column going in to the LGM (25.3 to 21.8 ka) and an increase in accumulation rate. I find it surprising that accumulation rate would increase through the LGM and this observation merits some discussion. I note that the accumulation rate declines during the ACR as one would expect under cooler conditions.*

This comment refers to the sentence found on p 13, L27-28 and also in Fig. 8d:

"After 25.3 ka, accumulation starts to increase and by the first $\delta^{15}$N-N$_2$ maximum (21.8 ka), accumulation had increased to $\sim$17 cm ice equivalent per year"

We agree that the early increase in accumulation is an interesting feature. It is the solution that the firn model interprets for periods of increasing $\delta^{15}$N-N$_2$ values. The alternative explanation for rising $\delta^{15}$N-N$_2$ is cooling during this time, which is not supported by $\delta$D data. The surprise expressed by the reviewer likely stems from the widely held assumption that accumulation is closely linked to site temperature – this is true in a broad sense Frieler et al. (2015), but does not apply to millennial-scale variations at (coastal) sites where synoptic systems deliver much of the snowfall Fudge et al. (2016).

To address this comment, we have added the following:

"This feature is not apparent in other ice cores from the Ross Sea region, but those records tend to be difficult to interpret because of chronological uncertainties, such as is the case for Taylor Dome, Baggenstos et al (2017), or because of unexplained jumps in $\delta^{15}$N-N$_2$, such as is the case for Siple Dome, Severinghaus et al. (2009)."

6. *The reconstruction and discussion of RICE accumulation history depends strongly on the questionable assumption that dD is a faithful recorder of temperature across the deglaciation. The potential for non-thermal effects on the dD record is critical and should be made earlier on in the paper (currently it is not until P14L25—30).*

How Delta-age is affected by uncertainties in past temperature history is included in our description of the Delta-age sensitivity experiment on page 11. This is the earliest section which describes in detail the firn models used to estimate Delta-age. Although non-thermal effects are not specifically mentioned, a wide range of temperature histories is included in the sensitivity analysis. To be more explicit, some text has been added to p11, l33:

"The sensitivity tests include a wide range of temperature histories so as to account for the possibility that some variations in $\delta$D were caused by non-thermal

effects such as variability in precipitation seasonality, moisture sources and pathways, and post-depositional vapor exchange."

**Technical comments:**

1. *Abstract line 4. Clim. Past should not be in the business of publishing 'preliminary observations'. See major comments on whether these should be presented at all.*

   This line has been removed. See response to Major comment 1, above.

2. *Intro first para: The main focus of the paper is timescale development and the introduction should direct the reader to that subject from the start. Marine ice sheet stability does not come up again in the paper so does not need to be described here. Remove the para and I'd suggest replace with some sentences on importance of timescale development.*

   Thank you for the specific suggestions on how to tighten the manuscript. We have decided to revise the content of the introduction to focus on discussion of difficulties of developing chronologies for ice cores. To accomplish this, we have removed text describing history of the Ross Sea and MISI from the introduction and shifted text from section 4 which summarizes our strategy for chronology development. This reduces the text in section 4 (preceding section 4.1) and eliminates some redundancy.

3. *Intro second para: Here two scenarios are put forward for glaciological history in the Ross Sea region. The later discussion should more clearly refer back to the scenario which is supported by the new results. Since this glaciological history is not the primary focus of the paper I would suggest to move the paragraph to the end of the introduction.*

   This paragraph will be edited and moved to section 2.

4. *P2L35: No need to pers. comm. a co-author.*

Removed reference to "personal communication".

5. *P3L14: This is the sort of information that is most relevant to the main age scale development task at hand and which belongs in the intro.*

The manuscript has been rearranged so that the introduction will focus on the chronology development. See response to technical comment 2.

6. *Section 2. Para 2 of the intro could be better fit into this section renamed something like "Roosevelt Island ice core and glaciological history".*

The manuscript has be rearranged and the description of Roosevelt Island and discussion of the significance its glaciology has been moved to Section 2. See response to technical comment 2.

7. *P4L20: I don't see any points in the RICE methane curve (Fig 3a) sitting 30 ppb above the WAIS curve. The legend does not inform which methane measurements are discrete and which are the problematic CFA.*

Problematic samples discussed on P4L20 are not highlighted in Fig 3, but can be seen in the figure. We will work to make these measurements clearer by adding the continuous $CH_4$ records (currently light gray in Fig. 3a) to the figure legend and by making the continuous $CH_4$ line darker.

8. *P6L20: I don't think pers. comm. of a co-author is needed, remove here and throughout.*

References of "personal communication" to work performed by coauthors has been removed.

9. *P6L5-15: The method used for each section of the core is repeated in the abstract, in line P3L5-20 and later again in the results. That's far too much repetition*

*and testing of the readers patience. Its essential to revise the structure to avoid this repetition.*

We will work to eliminate needless repetitions, but because the chronology development is a main contribution of the manuscript we feel that it warrants some inclusion in the abstract, introduction, and conclusion. The reviewer notes that we have also provided an overview of the chronology here (P5L5-15).

We have reduced the text in this section (section 4 "Strategy for developing the chronology"). See edits described in response to technical comment 2.

10. *P6L18: Also repeats earlier material in Intro.*

    See response to technical comment 9.

11. *P6L30: "35% to 75% of the relevant variable": please clarify what is meant here.*

    We have re-written this sentence to be clearer, as follows:

    Original text:

    "The method starts with a set of prior ACPs which all correspond to well defined variations in either methane or $\delta^{18}O_{atm}$ (Table 1). Age uncertainty of ACPs was estimated from the length of time between 25% and 75% of the observed change of relevant variable."

    Revised text:

    "Prior ACPs all correspond to well defined increases or decreases of either methane or $\delta^{18}O_{atm}$. The age uncertainty of an ACP is assumed to be related to the duration of the corresponding increase or decrease. For this analysis we assume that the uncertainty (2 standard deviations) for an ACP corresponds to the time elapsed between 25% and 75% of the change in either methane or $\delta^{18}O_{atm}$ (Table 1)."

12. *Fig 5d): Please explain to the reader why there is a large difference between the "best realization", judged in terms of the goodness of fit, and the number of occurrences of a particular fit.*

Added text to be inserted before P8,L18:

"The best age estimate (realization) is not necessarily the same as the most frequent age estimate. Fig. 5d shows an example from sample depth 621.28 m where there is a large difference between these two age estimates. Large differences can occur because the prior age estimate (i.e. the age estimate based only on prior ACPs) differs by a large amount from the "true" age of that sample and because the goodness-of-fit parameter considers the fit over the whole record. In the case of the sample at 621.28 m, most realizations resulted in an age estimate of this sample of 9200 yr BP, similar to its prior age estimate of 9,240 yr BP, but the best realization estimated the age to be 9012 yr BP. There are two possible reasons for this type of result: 1) that this realization is the best because it managed to push the age of this sample by >200 years towards younger ages while not significantly changing the ages of nearby sections which already fit well, or 2) that no significant improvement in the goodness-of-fit was found by adjusting the age of this depth, and the goodness-of-fit was dictated by other sections of the record."

13. *Section 4.1: it would help the reader if this section referred right at the start to Fig 5.*

Figure 5 is now referred to at the beginning of section 4.1 to show prior ACPs (white triangles in Fig. 5a) and a comparison between RICE and WAIS Divide $CH_4$ and $\delta^{18}O_{atm}$ on the final age scale (Fig. 5a-b).

14. *P9L3: I think its now the 4th time I read this.*

We removed this line. See response to technical comment 9.

15. *P9l15—19: As someone who works with these records I find this very hard to follow. Please revise for clarity.*

Original text (full paragraph quoted for context):

"Buizert et al. (2015) found that the annual layer counted portion of the GICC05modelext chronology (0-60 ka) (Andersen et al., 2006; Svensson et al., 2008) is systematically younger than ages of corresponding features found in the U/Th absolute dated Hulu speleothem record. A fit to Hulu ages was optimized by scaling the GICC05modelext ages linearly by 1.0063. For the target records we adopt the same scaling as Buizert et al. (2015) for the annual layer counted section of NGRIP, ending at 60 ka in the GICC05modelext chronology and equating to 60.378 ka in WD2014 (and RICE17). Ages older than 60 ka in the GICC05modelext chronology are derived from the ss09sea Dansgaard-Johnsen model (Johnsen et al., 2001; NGRIP Community Members, 2004) which is not susceptible to under counting of annual layers. This portion of GICC05modelext was stitched to the annual layer counts by subtracting a constant 705 years (Wolff et al., 2010). For this section of the target NGRIP records, a constant 378 years is added to the age from GICC05modelext starting at the depth corresponding to 60 ka in the GICC05modelext (60.378 ka in WD2014)."

Revised text (edits in red):

"Buizert et al. (2015) found that the annual-layer-counted portion of the GICC05modelext chronology (0-60 ka) (Andersen et al., 2006; Svensson et al., 2008) is systematically younger than ages of corresponding features found in the U/Th absolute dated Hulu speleothem record. A fit to Hulu ages was optimized by scaling the GICC05modelext ages linearly by 1.0063. This suggests that on average 6.3 out of every 1000 annual layers were not counted. For our NGRIP-based target records, we adjust the NGRIP age scale by adopting the scaling of Buizert et al. (2015). Older ages in the GICC05modelext chronology are derived from the ss09sea Dansgaard-Johnsen model (Johnsen et al., 2001; NGRIP Community Members, 2004). To make this section continuous with the adjusted annual layer counted section, we have added a constant 378 years (0.0063*60,000) for depths older than 60 ka in the target GICC05modelext ages."

16. *Section 4.3: Shorten it.*

   We will edit this section to shorten it.

17. *P11L7: The delta-age is established using a firn densification model, in which the modelling relies on a RICE temperature history derived from dD. The temperature history is thus integral to the development of the age scale of the ice, however the dD-based temperature reconstruction is cited as a pers. comm. I think the authors need to refer to a published temperature history or include the temperature history here... returning from coffee break... ok reading further down I see there are some more details on the assumptions in the temperature reconstruction and comparison to borehole data. Remove the pers. comm and see major comments.*

   Our estimates of Delta-age, which will me made available with the paper, are dependent on assumptions regarding the temperature history. This record will be made publicly available in a forthcoming community manuscript from the RICE project. As described in the paper we account for uncertainties in Delta-age which result from our assumptions with a Monte-Carlo approach. We include chronological uncertainties, uncertainties in the assumptions in deriving a temperature history, uncertainty in constraints due to measurement error, and uncertainties from non-temperature related effects within the $\delta$D record.

   Reference to "personal communication" will be removed.

18. *P12L24: Good. Agreed.*

19. *Section 5.1: There is no doubt that this discussion of methane variability is interesting. In my view it should be the subject of a stand-alone paper, in which one*

*would like to see detailed comparison of the various records and labelling of the methane trends that have been attributed to anthropogenic activity. As it stands the two paragraphs do not give a thorough treatment but still take up substantial space. If it must be included then I would suggest to scale back the section, certainly not so much introductory material is needed (it's not until near the final lines of the section that the RICE results are even referred to).*

This section will be removed. See response to major comment 4.

20. *Fig 4d. Adjust the y limits so we can more easily see the age uncertainty.*

    The axes in Figure 4d have been adjusted.

21. *P13L33: Include the uncertainty in the onset of the d15N change at 14.71 ka; I'm far from convinced that it significantly precedes onset of Bølling at 14.64 $\pm$ .19 ka.*

    The age of the depletion in $\delta^{15}$N-N$_2$ unambiguously precedes the onset of the Bølling, which is defined by an abrupt increase in CH$_4$, because both events are recorded in gas-phase measurements and the change in $\delta^{15}$N-N$_2$ is observed at a deeper depth than the change in CH$_4$. We have edited the text from:

    "Curiously, this abrupt decrease in $\delta^{15}$N-N$_2$ precedes the increase in CH$_4$ marking the onset of the Bølling-Allerød."

    To:

    "Curiously, this abrupt decrease in $\delta^{15}$N-N$_2$ is observed in samples deeper than the increase in CH$_4$ marking the onset of the Bølling-Allerød meaning that this climate event unambiguously precedes the Northern Hemisphere event."

    I have attached a figure which shows the $\delta^{15}$N-N$_2$ and CH$_4$ records during the deglaciation plotted versus depth.

22. *P14L11: This interesting sentence suffers from being way too long.*

See response to technical comment 23. We have shortened it.

23. *P14L19–40: It would be more logical and much easier for the reader to follow your arguments if you set out the preferred explanation first and then explain, briefly, why some potential alternative explanations are unlikely. I don't find the preferred explanation very convincing: I don't see any quantitative data to support it, only some arm waving analogy to recent periods.*

Changes to P14 L5-30 were made in accordance with the reviewer's suggestion on how to organize the discussion of possible interpretations.

24. *Section 5.3: It would help the reader to refer early in the section to the "maximum" and "fast and thin" Denton (1989) scenarios that were set up in the introduction.*

We will make this change.

25. *P15L4: Again refer to the scenario set up in the introduction, here and elsewhere in this section.*

We will make this change.

26. *P15L18: Refer to the dD record in Fig3b. Not to a pers comm!*

Reference to "personal communication" will be removed.

We would like to note that $\delta$D is not shown in Fig 3b. What is shown it the $^{18}O/^{16}O$ ratio for $O_2$.

27. *P15L26: The comment about an MBL ice dome comes out of the blue and its far from obvious who it provides an alternative explanation for the continuity of the record. Clarify or drop.*

We will change text to further incorporate the idea of a MBL ice dome. This was a hypothesis from several previous publications to explain geomorphological features in the eastern Ross Sea.

Original Text (full paragraph provided for context):

"Geomorphological features on the Ross Sea bed do provide evidence of an expansive ice sheet which extended past Roosevelt Island during the LGM (Shipp et al., 1999; Anderson et al., 1984, 1992, 2014; Halberstadt et al., 2016). The stability of the Roosevelt Island ice dome and of Siple Dome implies that at this time, WAIS flowed around these sites rather than over them. This observation implies that as WAIS grew spatially, its thickness in the Ross Sea was limited, conditions that indicate ice streams were active throughout the last glacial period. Alternatively, Price et al. (2007) proposed that the geomorphological features observed in the eastern Ross Sea may represent building of an ice dome in Marie Byrd Land. The RICE records can not distinguish between these scenarios."

Revised Text:

"Geomorphological features on the Ross Sea bed do provide evidence of grounded ice north of Roosevelt Island during the LGM (Shipp et al., 1999; Anderson et al., 1984, 1992, 2014; Halberstadt et al., 2016). If these features were formed by an extended WAIS, it would imply that ice flowed around Roosevelt Island and Siple Dome and therefore must have been limited in its thickness. These conditions would indicate ice streams were active throughout the last glacial period. Alternatively, these geomorphic features may be the result of ice from a different origin. Price et al. (2007) proposed that during the LGM, an ice dome may have existed on Mary Byrd Land. In this scenario, thick, grounded ice could exist north of Roosevelt Island without flowing over or around the Roosevelt Island sea rise. The RICE records can not distinguish between these scenarios."

28. *Conclusions para 1: The fifth time we read this?*

    We removed this. See response to Technical comment 9.

29. *Many references found in the introduction do not come up again in the discussion. I'd suggest a bit more focus and continuity between the most relevant literature*

*flowing from the intro to the discussion.*

We agree with this comment and will reduce references which are only used in introduction.

30. *As a final point, it is tedious as a reviewer to have to spend so much time commenting on structure, something the author team could have worked on internally prior to submission. The age scale is important and should be presented as accessibly as possible.*

**References**

Frieler, K., Clark, P. U., He, F., Buizert, C., Reese, R., Ligtenberg, S. R. M., Van Den Broeke, M. R., Winkelmann, R., and Levermann, A.: Consistent evidence of increasing Antarctic accumulation with warming, Nat. Clim. Change, 5, 348-352, https://doi.org/10.1038/nclimate2574, 2015.

Fudge, T. J., Markle, B. R., Cuffey, K. M., Buizert, C., Taylor, K. C., Steig, E. J., Waddington, E. D., Conway, H., and Koutnik, M.: Variable relationship between accumulation and temperature in West Antarctica for the past 31,000 years, Geophys. Res. Lett., 43, 3795-3803, https://doi.org/10.1002/2016GL068356, 2016.

Winstrup, M., Vallelonga, P., Kjær, H. A., Fudge, T. J., Lee, J. E., Riis, M. H., Edwards, R., Bertler, N. A. N., Blunier, T., Brook, E. J., Buizert, C., Ciobanu, G., Conway, H., Dahl-Jensen, D., Ellis, A., Emanuelsson, B. D., Keller, E. D., Kurbatov, A., Mayewski, P., Neff, P. D., Pyne, R., Simonsen, M. F., Svensson, A., Tuohy, A., and Wheatley, S.: A 2700-year annual timescale and accumulation history for an ice core from Roosevelt Island, West Antarctica, Clim. Past Discuss, https://doi.org/10.5194/cp-2017-101, in review, 2017.

---

## Author Comment (AC2) · 24 Jan 2019

**Responses to Reviewer 2:**
First, Thank you for reviewing our manuscript. We agree with many of the suggestions made by reviewer 2. A number of these were in regard to improving organization and conciseness of the manuscript. This requires rearranging sections and a significant number of edits at the sentence level. For brevity, we do not include every edit in our response to reviewer comments.

**General Comments:**

1. *This manuscript presents a suite of new gas records from an ice core drilled at Roosevelt Island, an ice rise in the Ross Sea. The primary objective is to*

*establish its chronology by annual layer counting for relatively shallow depths and matching of gas records with existing WAIS Divide and Greenland ice core chronologies. The continuous part of the ice core extends to 65 kyr BP, suggesting that the Roosevelt Island has existed since at least this age. CH4 records show centennial-scale variability throughout the Holocene, with implications on natural vs. anthropogenic CH4 emission in pre-industrial periods. These discussions have some important implications for past climate and ice sheet variations. The dating method developed here is a nice contribution to the ice core community.*

*However, the lack of water isotope records and interpreted temperature records in this manuscript makes it difficult to review the estimated annual layer thickness using a firn densification model and its effects on dating and paleoclimatic implications. I find this study is potentially an important contribution to paleoclimatic communities but do not recommend publication in its current form. The authors would need to decide if they remove some parts of the manuscript regarding annual layer thickness estimates from firn modeling (but it will make the manuscript much less attractive), or they add water isotope data and temperature estimate (I would recommend the latter for publication in CP).*

The depth-age relationship, from which annual layer thickness is derived, is primarily dependent on the gas-based age constraints with only a small correction arising from the climate-dependent $\Delta$-age. In this approach, temperature has only a secondary effect on annual layer thicknesses. One exception to this statement may be during the deglaciation when large changes in $\Delta$-age are implied by rapid changes of $\delta^{15}$N-N$_2$.

We estimate past temperature based on the measurements of $\delta$D. The full high-resolution record of $\delta$D will be made publicly available in a forthcoming RICE project community paper led by project PI Nancy Bertler.

2. *The discussion of anthropogenic and natural CH4 variability needs some quanti-*

*tative analyses (for example comparing frequency and variability after detrending for different time periods). To my eyes, the CH4 records appear to have different centennial-scale variations in earlier and later parts of the Holocene.*

We have decided to remove this section (5.1 New observations of centennial-scale variability in the Holocene methane cycle) from the manuscript in response comments from reviewer 1. Content of this section is not discussed elsewhere in the paper and we hope that its removal will focus the paper on the other chronology-based conclusions and on the chronology development.

**Specific comments:**

1. *P5, L5. Regardless of the careful trimming of the ice in the same shape, the cut-bubble effect should change (generally decreasing) with depth due to the change in bubble sizes. The cut-bubble effect thus needs to be corrected.*

   We agree that the effect described by the reviewer should exist, but as we mentioned in the original text we have not made this correction to our total air content measurements because we do not believe that an accurate correction can be calculated for the RICE samples. This is because many of the RICE samples were fractured. Air intersecting a fracture may escape under vacuum and it is difficult to calculate the surface area of fractures. The air lost through fractures may be significantly greater than that lost due to sample preparation. To avoid samples with obviously large cut-bubble effects we excluded samples based on visual observations: samples with large fractures, with many fractures, which were comprised of multiple pieces of ice, or were an odd shape.

   However, we chose to include samples with small cracks in our data set. Small fractures are inconsistent in allowing air to escape. For this reason, it may not be possible to separate variations in TAC from gas loss through small fractures. In practice, small fractures can be hard to see which makes it possible for air to be lost through a fracture which was not observed.

To clarify this issue we propose the following change in section 3.2 (P5, L3-9).

Original text:

"Air trapped in bubbles, clathrates, or fractures intersecting the surface of the sample is lost, an effect called the cut-bubble effect (Martinerie et al., 1990). The cut-bubble effect is difficult to quantify, especially in ice which contains fractures through which air may be lost. No correction for the cut-bubble effect was applied to the TAC measurements presented here. Samples were cut to uniform shapes whenever possible to ensure that the cut-bubble effect was relatively constant in order to limit the influence it has on the variability of the TAC record. TAC analysis was rejected when the cut-bubble effect was believed to greatly impact the results, such as in samples which fractures could not be excluded or were excluded by cutting the sample into irregular shapes or into multiple pieces."

New Text:

"The cut-bubble effect is difficult to quantify, especially in ice which contains fractures through which air may be lost. Samples were cut to uniform shapes whenever possible to ensure that the cut-bubble effect was relatively constant in order to limit the influence it has on the variability of the TAC record. However, many samples contained fractures through which air may be lost and greatly impact TAC. TAC data were rejected when gas loss was believed to greatly impact the results, such as in samples with fractures or samples which consisted of multiple pieces. However, small fractures were difficult to see and their contribution to gas loss is unknown. For this reason, we choose to not correct TAC measurements for the cut-bubble effect."

2. *P16, L28. I do not understand why the temperature stability of the sample leads to the improvement in S/N of the gas chromatograph.*

Insulation was added to the system as an attempt to minimize the water vapor in the headspace of our sample flasks (by decreasing the head space temperature)

[Figure]

and to minimize variations of water vapor throughout the day. We also have made efforts to regulate the amount of ethanol in the chilled bath for the same purpose.

We adjusted the sentence to clarify our intent and what we did.

Original text (P16, L26-28):

"Since Mitchell et al. (2013), insulation has been added around the ethanol bath and above where the flasks are mounted. The added insulation decreased the temperature variability of the ethanol bath and of the sample flasks throughout the day allowing for better measurement of pressure and improved signal-to-noise for the chromatograph."

Edited text:

"Since Mitchell et al. (2013), insulation has been added around the ethanol bath and above where the flasks are mounted. The added insulation reduced the temperature and water vapor content of gas in the headspace of the flasks and decreased variability throughout the day. Both can affect methane measurements by changing the pressure reading or the retention time of methane in the GC column. Additionally, we have made efforts to more carefully regulate the amount of ethanol in the chilled bath and the temperature of hot water bath during melting. These steps improved stability of measurements and extraction between days."

3. *P17, L30. Please explain why the solubility correction factors are so different for sample and bubble-free ice?*

The solubility corrections are empirically derived, so the difference in solubility between glacial sample ice and bubble-free ice is something we observe. We will add additional explanation to the supplement describing our theory about why our solubility results for bubble-free ice and glacial ice are different.

We believe the difference results from the differences between how blank ice and ice containing air behave during melting.

- Bubble-free ice melts slowly in comparison to glacial ice which sometimes melts rapidly and cracks violently. This, along with bubbles rising and breaching the meltwater surface, cause disturbances in the water-air interface and promotes exchange of $CH_4$ into the meltwater. This should lead to greater mixing and homogenization of air and water.

- Bubbles released into the meltwater will be at higher pressures than the overlaying air because of surface tension. The higher partial pressure of $CH_4$ in those bubbles, in comparison to the standard gas added over the bubble-free ice, will cause air to go in to solution faster.

- Because glacial ice tends to be melted sooner than blank ice, a longer time period for liquid-gas exchange is available.

4. *Fig. 2c and i. The scales of the axes should be the same for the left and right panels.*

    Done. Thank you for pointing this out.

5. *Fig. 5d. Why is the vertical line drawn at about 9000 yr BP and not near 9200 yr BP (highest occurrences)?*

    There is a difference between what we considered the "best" chronology and the most frequently occurring age of a specific depth. If we were to accept the most frequently occurring age of each sample depth in our Monte Carlo analysis as our final chronology, there is no guarantee that the age of ice increases with depth. Instead, we chose the age-scale with the best "goodness-of-fit." In this routine, goodness-of-fit is a single statistical value describing how similar both the $CH_4$ and $\delta^{18}O_{atm}$ records look like their corresponding records from WAIS Divide.

    Added text to be inserted before P8,L18:

    "The best age estimate of a sample depth (single point on the depth-age scale) is not necessarily the same as the most frequent age estimate for that depth.

Fig. 5d shows in example from sample depth 621.28 m where there is a large difference between these two age estimates. In the case of sample depth 621.28 m, most realizations resulted in an age estimate of this sample of 9200 yr bP, similar to its prior age estimate of 9,240 yr bP, but the best realization estimated the age to be 9012 yr BP. The difference in estimates could be random because no significant improvement in the goodness-of-fit was found by adjusting the age of this depth or because shifting this age tended to worsen the fit of adjacent depths."

6. *Supplementary file "RICE17_Interpolated_Ages_20180530.txt" appears to contain two units for the ice age (probably C.E. and yr BP are switched at 343.5 m).*

Corrected.

---

## Author Response (AR2)

**Responses to Reviewer 2:**

1. *P12, L25-31. Also, response to Reviewer 1's comment 21: Please give values of the difference between the shifts in d15N and CH4 in terms of depth and gas age. You also mention to add a new plot of d15N and CH4 against depth, but I do not see a plot that visually indicates the depth difference in the revised manuscript.*

   The figure, which was intended as an attachment to the author's comment, is now included in the paper as Figure 9.

   We have also revised the text on P12 L25-31, to include the depths and gas ages of shifts in d15N and CH4.

   *"Analysis of the lead-lag relationship between methane and the thermal-signal from d15N-N2 has been used to infer a closely coupled climate throughout the tropics and northern hemisphere (Rosen et al., 2014)." This is about thermal diffusion signal in d15N in Greenland ice cores, and it probably confuses readers because you give the impression that you are going to interpret the RICE d15N in the same way. Please clarify here or just below, that Antarctic d15N changes are mostly associated with the changes in gravitational separation (firn thickness).*

   We have adjusted the text to comment on thermal fractionation of $\delta^{15}$N-N$_2$ in Antarctic ice cores. The text now reads:

   "Where as Rosen et al. (2014) specifically considered the thermal component of $\delta^{15}$N-N$_2$, the thermal component is considered to be negligible in Antarctic cores because air temperature only changes gradually. At Roosevelt Island, this period of low $\delta^{15}$N-N$_2$ values is interpreted to represent shallow firn thickness which in the firn model is caused by a large reduction of snow accumulation (less than 10 cm/yr, Fig. 8e)."

2. *The references at L28 could include Buizert et al., 2018.*

   A reference to "Buizert et al., 2018" has been added.

   *"Curiously, the abrupt decrease in 15N-N2 at 12.38 ka" at L29. Isn't it at 14.71 ka?*

   The age has been corrected.

3. *Response to Reviewer 2's general comment 1. Your reply only gives reasons to show the dD in this manuscript (especially, "One exception to this statement may be during the deglaciation when large changes in Delta-age are implied by rapid changes of d15N-N2."). The issue*

*here is not whether you think the data are significant or not for your conclusion (or whether the correction associated with temperature is small or large), but it is the invisibility of and inaccessibility to the essential data to obtain the main result of this manuscript (moreover, it is impossible for the reviewers to see the unimportance unless the data is shown). Thus I keep my position in the original review as unanswered.*

The dD data will be part of a forthcoming manuscript. Arrangements have been made with the editor to withhold publication of this manuscript until the data is made available.

4. *Response to Review 2's specific comment 1 (cut-bubble effect). I understand that you may have larger uncertainty (potential gas loss through fractures) than the simple effect of the cut bubble at the surface of a sample. However, it cannot be the reason for no correction at all. You should apply the Martinerie's correction procedure for the cut-bubble effect (so you can partly correct for the lost volume, which is far better than no correction) and then should discuss the importance (or not) of the TAC underestimation and its variability by fractures for the sake of glaciological discussion using TAC. If you are going to omit the cut-bubble correction, you should at least provide the convincing discussion that the cut-bubble effect and gas loss through fractures did not induce artifactual TAC changes during the termination.*

We have now included a cut-bubble correction for the TAC measurements. Although the application of this correction is intended for interpretation of the absolute values of TAC (in contrast to relative changes between samples), we advise against interpreting the RICE TAC record in this way. We have edited the section describing TAC measurements in accordance.

New Text:

Preparation of ice samples causes loss of air from bubbles which intersect the surface of the sample, an effect known as the cut-bubble effect (Martinerie et al., 1990). The amount of air lost due to the cut-bubble effect depends on the geometry of the sample and the bubble size. Ice samples were cut to uniform shapes to limit variability of TAC related to the cut-bubble effect so that TAC can be directly compared between samples. However, inclusion of some fractures was often unavoidable and their contribution to gas loss is potentially large. TAC data were rejected when gas loss was believed to greatly impact the results, such

as in samples with visible fractures or samples which consisted of multiple pieces. Of the 706 samples measured at OSU for TAC, 165 results were rejected based upon visual inspection of the sample. Many of these came from the 670-752.95 m (11.7-83 ka) interval where only 58 of 177 TAC measurements are considered reliable. Reproducibility of replicate TAC measurements is 0.7% after application of a correction for the cut-bubble effect (reproducibility of 0.6% without correction). Estimate of uncertainty of TAC measurements is 28% with the largest contributions to the uncertainty from the shape of the sample and the assumed bubble diameter used for the cut-bubble correction.

The TAC record from the RICE ice core without the cut-bubble correction appears remarkably consistent (age weighted mean=0.1182 cm$^3$ air-STP/g ice, age weighted standard deviation=0.0023 cm$^3$ air-STP/g ice, n=410). After application of the cut-bubble correction, the TAC record shows a trend to higher values at more recent ages (Fig. 1d). This trend reflects the decrease of bubble size with depth and therefore a larger cut-bubble correction for shallower samples. Although this trend has a physical explanation, it is not statistically significant due to the uncertainty of the cut-bubble correction. We advise caution when interpreting the absolute values of TAC from the RICE ice core due to the potential artifacts caused by gas loss through fractures and the large uncertainty in the cut-bubble correction.

5. *Response to Reviewer 2's specific comment 3. Thank you for your explanation, and please include what you wrote with the three bullet points in the manuscript.*

We have added the suggested text to the supplemental information regarding calculation of CH4 concentration (page 16, line 29). This text now reads:

"We believe the difference in $S$ for the different ice samples results from the differences between how blank ice and ice containing air behave during melting. Specifically,

- Bubble-free ice melts slowly in comparison to glacial ice which sometimes melts rapidly and cracks violently. This, along with bubbles rising and breaching the melt water surface, cause disturbances in the water-air interface and promotes exchange of $CH_4$ into the melt water. This should lead to greater mixing and homogenization of air and water.

- Bubbles released into the melt water will be at higher pressures than the overlaying air because of surface tension. The higher partial pressure of $CH_4$ in those bubbles, in comparison to the standard gas added over the bubble-free ice, will cause air to go in to solution faster.

- Because glacial ice tends to be melted sooner than blank ice, a longer time period for liquid-gas exchange is available."

6. *P2L4-5: "Visual stratigraphy" should be added to the list of parameters used for layer counting.*

   "Visual stratigraphy" has been added to the list of ways which annual layers can be identified in ice cores.

7. *P10L6: Add, Kawamura et al., 2007 (Nature) for a d18Oatm record.*

   This citation has been added.

8. *Supplementary data needs corrections. TAC: the name for TAC column reads, "mean CH4" Interpolated age: Ice age unit is year CE from the surface to 343.50 m. (it seemed uncorrected in the folder I downloaded "cp-2018-68-supplement-version2")*

   The column headers have been corrected for the file: "RICE_Final_TAC.txt".

   Interpolated ages in "RICE17_Interpolated_Ages.txt" has been updated.

[revised manuscript text omitted]

---

## Author Response (AR3)

**Responses to Reviewer 2:**

1. *In several part of the manuscript you state that the ice core length is 763 m except at page 2 line 2 in the Introduction. Please, may you check?*

   There was a typo on the depth which has been corrected. The correct depth is 753.75 m, which is the deepest/oldest age constraint for the RICE ice core. The bottom ∼10 m of core (753.75-764 m) are not included in the RICE17 age scale.

2. *Page 2, lines 3-4: absolute age markers .... Are you referring only to volcanic ash layers? I do not think so ..... They should be sulphate volcanic peaks and in case also ash layer .... Please modify.*

   Volcanic ash layers was meant as an example of an absolute age marker, not a comprehensive list. We have modified the text to clarify this point.

3. *Page 6, line 3: I would add also here that the Antarctic dN changes are mostly associated with changes in gravitational separation (see comment from Referee 2).*

   We have added the following sentence to page 6, line 3: "In Antarctica, temperature changes gradually and the effect of thermal separation is minimal."

4. *Page 6, line 13. This figure 3 rather than 5. Please modify.*

   The markers representing age control points can be seen in both figures 3 and 5. A reference to figure 3 has been included here.

5. *Page 13, line 5: it seems that at the end you did not modify this (see referee comment): the date is not 12.38 ka but 14.71 ka. Please change.*

   The date has now been changed.

6. *Page 13, lines 20-22: the sentence starting with "The timing ... " is not clear. You are referring to the dust record of Taylor Dome ... describe what happened in the dust record of Taylor... Otherwise it is not clear.*

   We have now included a description of the changes in dust composition observed in the Taylor Dome ice core by Aarons et al. (2016) and their interpretation of the dust record.

7. Data availability: when the paper will be finally accepted please add a link where the data will be archived.

   The methane, d15N-N2, d18O-atm, and TAC data are all included as supplementary text files. We will also archive these data sets and include a link to them.

8. *Page 37: the caption of figure 8 is not correct: b) is methane, c) is Annual Layer Thickness, d) is Lock-in-Depth and e) is accumulation rate ... Please modify.*

   The caption has been corrected.

---

## Author Response (AR4)

Thank you very much for your reviews of our manuscript. Your feedback has greatly improved this manuscript.

We have submitted a slightly modified manuscript to comply with your requests for including the dD record from which we base our temperature history. Edits include:

- showing the RICE dD record to figures 2 and 3,
- a subsection (Section 3.4) describing measurement of dD and relevant references,
- new text in Section 4.4 (describing the Delta-age model) which refers to the dD record shown in Figure 3 and citing the minimal impact that the chosen smoothed-length of the dD record has on determining Delta-age, and
- submission of the RICE dD datafile as SoM will be made available through the USAP data center.

In these figures we show a smoothed version of the record, produced using a LOESS filter with characteristic length of 500 years.

Cheers,
James Lee

[revised manuscript text omitted]

---

## Author Response (AR5)

Dear Editors of Climate of the Past,

Thank you for the last round of technical edits. We have made the following changes in accordance to your suggestions.

-Page 2, line 27: *change "an" before Monte Carlo with "a".*
The word has been changed to "a"

-Page 4, line 31: *may you check the citation Edwards et al. (in prep), also in the Reference. If not published, may you use another?*
We have removed this reference. This reference supported another, published, article ("Mitchell et al. 2015 JGR-Atmospheres")

-Page 11, line 12: *is there any alternative to the citation Clemens-Sewall? in case, also in the References. Moreover, perhaps it should be moved to the line 19. May you check?*
We have removed this reference from the reference list. It had previously been removed from the body of the text but was accidentally left in the reference list.

-Page 16, line 16: *add also dD data in "Data availability section".*
dD has been added to the "Data Availability" section.

-Page 32: *please update the caption of figure 2!*
We have updated the caption to figure 2 and in-text references to panels of figure 2 to reflect the addition of the dD record to this figure.

Page 33: *please update the caption of figure 3!*
We have updated the caption to figure 3 and in-text references to panels of figure 3 to reflect the addition of the dD record to this figure.

[revised manuscript text omitted]